# Regulatory controls of duplicated gene expression during fiber development in allotetraploid cotton

Jiaqi You[1,5], Zhenping Liu[1,5], Zhengyang Qi[1,5], Yizan Ma [1,5], Mengling Sun[1], Ling Su[1], Hao Niu[1], Yabing Peng[1], Xuanxuan Luo[1], Mengmeng Zhu[1], Yuefan Huang[1], Xing Chang[1], Xiubao Hu[1], Yuqi Zhang[1], Ruizhen Pi[1], Yuqi Liu[1], Qingying Meng[1], Jianying Li [1], Qinghua Zhang[1], Longfu Zhu [1], Zhongxu Lin [1], Ling Min [1], Daojun Yuan[1], Corrinne E. Grover [2], David D. Fang[3], Keith Lindsey [4], Jonathan F. Wendel [2], Lili Tu [1]✉, Xianlong Zhang [1]✉ & Maojun Wang [1]✉

Polyploidy complicates transcriptional regulation and increases phenotypic diversity in organisms. The dynamics of genetic regulation of gene expression between coresident subgenomes in polyploids remains to be understood. Here we document the genetic regulation of fiber development in allotetraploid cotton *Gossypium hirsutum* by sequencing 376 genomes and 2,215 time-series transcriptomes. We characterize 1,258 genes comprising 36 genetic modules that control staged fiber development and uncover genetic components governing their partitioned expression relative to subgenomic duplicated genes (homoeologs). Only about 30% of fiber quality-related homoeologs show phenotypically favorable allele aggregation in cultivars, highlighting the potential for subgenome additivity in fiber improvement. We envision a genome-enabled breeding strategy, with particular attention to 48 favorable alleles related to fiber phenotypes that have been subjected to purifying selection during domestication. Our work delineates the dynamics of gene regulation during fiber development and highlights the potential of subgenomic coordination underpinning phenotypes in polyploid plants.

The significance of polyploidy in plants has long been recognized, with respect to both species diversification and implications for genetic improvement during plant breeding[1–5]. The genome-wide redundancy conferred by whole genome duplication increases genetic diversity and provides additional avenues for evolving functionality, thereby increasing biological complexity[6,7]. Duplicated genes in polyploid organisms, or homoeologs, are coordinated in several ways, mediating gene dosage

effects, gene balance, interaction between *cis*- and *trans*-acting factors and rewiring of gene expression networks[8–13]. Unequal contribution of the expression of each homoeologous gene to the total expression level (that is, biased homoeolog expression) has been observed in many allopolyploid plants[14–17]. Homoeologous expression bias reflects one aspect of duplicate gene coordination, which is thought to be associated with increased variation and hence adaptive potential. The many

[1]National Key Laboratory of Crop Genetic Improvement, Hubei Hongshan Laboratory, Huazhong Agricultural University, Wuhan, China. [2]Department of Ecology, Evolution, and Organismal Biology, Iowa State University, Ames, IA, USA. [3]Cotton Fiber Bioscience Research Unit, USDA-ARS, Southern Regional Research Center, New Orleans, LA, USA. [4]Department of Biosciences, Durham University, Durham, UK. [5]These authors contributed equally: Jiaqi You, Zhenping Liu, Zhengyang Qi, Yizan Ma. ✉e-mail: lilitu@mail.hzau.edu.cn; xlzhang@mail.hzau.edu.cn; mjwang@mail.hzau.edu.cn

**Fig. 1 | Gene expression atlas and genetic regulation during fiber development. a**, Samples used for RNA-seq in fiber development. The samples include ovules at 0 DPA and fibers at 4 DPA, 8 DPA, 12 DPA, 16 DPA and 20 DPA. The sample number of each stage is shown in parentheses. **b**, Principal component analysis (PCA) plots of the first two components for 2,215 RNA-seq samples. **c**, The number of genes expressed (dark gray outside line) or not expressed (lighter gray outside line) in all RNA-seq samples. **d**, The number of expressed homoeologous genes in each timepoint. Red, homoeologous genes with expression bias toward the At (BiasA); blue, homoeologous genes with expression bias toward the Dt (BiasD); gray, homoeologous genes without expression bias (BiasN); purple, homoeologous genes with expression bias toward the At and Dt (bidirect bias). **e**, The number of eQTLs and distribution of eGenes that had *cis*-eQTL, *trans*-eQTL or both. At, the At subgenome; Dt, the Dt subgenome; Sca, genes in unanchored scaffolds.

complexities of duplicate gene *cis*- and *trans*-interactions in biological networks, however, have only recently begun to be elucidated[8,10,12,13,17]. A particularly promising research avenue is the mapping of expression quantitative trait loci (eQTL) for genome-wide discovery of genetic regulatory variants that influence gene expression.

Allotetraploid 'upland' cotton, *Gossypium hirsutum* L., which originated following an interspecific hybridization event between two diploid ancestors (genome type AA and DD) approximately 1–2 million years ago[18], is the dominant source of natural renewable fiber for textiles. Cotton 'fibers' are single-celled epidermal ovular trichomes with modular expression and phenotypic stages encompassing initiation, primary wall synthesis, secondary wall synthesis and maturation[19]. As a vital economic commodity, cotton fiber development has been extensively studied, and many genetic loci and functional genes responsible for fiber development have been discovered[20–26]. However, our understanding of how the co-existing At and Dt subgenomes genetically coordinate the dynamic development of the fiber is limited. Accordingly, few breeding practices consider the interactions of genetic effects due to the two subgenomes, which have unique transcriptional and biochemical suites of interactions[8,10,12,13]. Here we present a genetic regulation analysis of dynamic gene expression in developing fibers across a suite of highly diversified *G. hirsutum* accessions and uncover the genetic components that may optimize homoeologous gene expression for unlocking the potential for fiber improvement.

## Results

### Gene expression atlas in fiber development

To uncover the genetic regulation of gene expression in fiber development, we collected 376 diverse *G. hirsutum* accessions for genome and transcriptome analysis. A total of 13.5 Tb of genome resequencing data were generated, with an average depth of 15.6× (Supplementary Table 1). Accessions were sampled at different developmental stages, including ovules on the day the flower opens (0 days postanthesis (DPA)) and fibers at five timepoints spanning elongation to secondary cell wall synthesis (4, 8, 12, 16 and 20 DPA). A total of 2,215 RNA-sequencing (RNA-seq) data samples were generated (with 41 failing to sample), with an average of 40 million read pairs for each sample (Fig. 1a and Supplementary Table 2). Principal component analysis showed that samples from the same timepoint clustered together, and samples from adjacent development stages were closely associated, indicating a continuous developmental trajectory (Fig. 1b and Extended Data Fig. 1a,b).

A total of 49,860 genes were transcribed (fragments per kilobase of transcript per million mapped reads (FPKM) > 0.1 in at least 5% accessions) during cotton fiber development, including 24,486 in the At subgenome and 25,238 in the Dt subgenome (Fig. 1c, Extended Data Fig. 1c and Supplementary Table 3), of which 12,875 were expressed at one to five timepoints (Extended Data Fig. 1d). Of note is the observation that 20,189 homoeologous gene pairs (2*n* = 40,378) were expressed (Extended Data Fig. 1e and Supplementary Tables 3 and 4), of which 79.6% (*n* = 16,081) showed expression bias (expression level fold change ≥2 between two homoeologs in ≥ 5% accessions) toward the At (BiasA) or Dt (BiasD) subgenome in at least one timepoint (Fig. 1d), including 3,256 pairs with stable direction of expression bias at all timepoints (Extended Data Fig. 1f).

### Genetic regulation of dynamic gene expression

Using both RNA-seq data in fiber development and genome resequencing data for each accession, we considered the impact of genetic variants on gene expression. We used eQTL mapping to identify *cis*- (within 1 megabase (Mb) of each gene on either side) and *trans*- (>1 Mb apart or on a different chromosome) regulatory variants (eVariants) that are

associated with differences in gene expression. We leveraged approximately 2.7 million SNPs with a minor allele frequency (MAF) > 0.05 in conjunction with the 45,545 genes exhibiting expression variation across all stages (64.8% of the predicted transcriptome) in the eQTL mapping (Supplementary Figs. 1 and 2). In total, 53,854 *cis*-eQTLs were identified for 18,637 genes (23.8–28.4% of *cis*-eQTLs overlapped with open chromatin[20]) and 23,811 *trans*-eQTLs were identified for 10,391 genes (eGenes, that is, genes whose expression is associated with one or more eQTL), with the largest number of eGenes (12,674, or 55.1%) identified at 12 DPA, because of the presence of a few eQTL hotspots (that is, local chromosomal regions that were associated with transcriptional regulation of more than three genes) at this timepoint (Fig. 1e and Supplementary Table 5).

To compare the sharing of eQTL and further understand the differences in genetic regulation between stages, we collated eQTLs from all stages and distinguished regulatory mechanisms that were found in only one stage (stage-specific) or at least two stages (stage-shared; Supplementary Figs. 3 and 4). We found that stage-shared eQTLs showed larger effects than stage-specific eQTLs, and genes with stage-shared eQTLs had a higher proportion of *cis*-eQTLs than stage-specific eGenes (Extended Data Fig. 2a,b). For each *cis*-eQTL, we assessed sharing among stages by comparing local false sign rate (LFSR) and magnitude, which represented the metrics for eQTL significance and effect estimates[27]. In this analysis, 27,102 (50.2%) *cis*-eQTLs that were shared among six timepoints had significant signals in the comparison of LFSR, of which only 11,072 had detected effects (fold change of magnitude ≤2 between different timepoints), suggesting many stage-shared eQTLs (LFSR ≤ 0.05) showed variable effect magnitude during fiber development (Extended Data Fig. 2c). In terms of the effect magnitude estimates for eQTLs that are shared between timepoints, the vast majority (91.1%) showed consistent effect direction (Extended Data Fig. 2d).

## Fine-mapping of fiber quality associations

We decoded the genetics of fiber quality-related traits by integration of genome-wide association study (GWAS) and eQTL data. A total of 18 QTLs were identified, including five for fiber length (FL), six for fiber strength (FS), four for fiber elongation (FE) rate and three for fiber uniformity (FU; Fig. 2a and Supplementary Fig. 5), of which nine were previously uncharacterized (Supplementary Table 6). This result was partially verified by an $F_2$ population (Supplementary Fig. 6). Then, two complementary methods, including a transcriptome-wide association study (TWAS) and a colocalization analysis[28,29], were used to prioritize causal genes for fiber quality-related traits. Using TWAS, 1,255 genes (false discovery rate (FDR) < 0.05) were identified across the whole genome (311 for FL, 655 for FS, 877 for FE and 308 for FU; Fig. 2a, Supplementary Fig. 7 and Supplementary Table 7). Specifically, 43 genes were prioritized as candidate genes for 17 GWAS QTLs using TWAS (Supplementary Table 8). As proof, genetic knockout of a TWAS gene (Ghir_D10G004160) showed that FL became significantly shorter (Extended Data Fig. 3a,b and Supplementary Figs. 8 and 9). Using two

colocalization strategies, summary-data-based Mendelian randomization (SMR) and coloc[29,30], we also identified 14 fiber quality-related genes (3 for FL, 4 for FS, 6 for FE and 4 for FU) in nine GWAS QTLs (Fig. 2b, Extended Data Fig. 3c, Supplementary Figs. 10 and 11 and Supplementary Table 8), 11 of which overlapped with genes in the TWAS analysis. Of note is the observation that 1,243 of the 1,258 genes from TWAS and colocalization analysis maintained the same effect on fiber quality traits in fiber development.

On chromosome D05, we identified a QTL that is significantly related to FL, for which *BB2* (Ghir_D05G007220) is characterized as a causal gene (Fig. 2c). *BB2* encodes an E3 ubiquitin ligase, which was found to be related to cell proliferation and elongation in *Arabidopsis*[31], and positively regulates FE after 12 DPA. Based on the expression pattern of *BB2*, all cotton accessions were divided into eight different groups, with groups 2, 3, 4 and 5 including 94.4% of accessions (Fig. 2d). From group 2 to group 5, the expression pattern of *BB2* was gradually delayed, and the median FL value of the corresponding accessions gradually increased (Fig. 2e). This result indicates that the expression pattern of group 5 (highly expressed from 8 DPA to 16 DPA) is conducive to FE, probably coinciding with the extended FE period. Similar to the analysis for *BB2*, we investigated the expression patterns for all 1,258 genes from TWAS and colocalization analysis in all accessions and defined 'favorable' expression patterns as those in accessions with favorable fiber quality traits, such as longer or stronger fiber. We found 158 FL, 196 FS, 349 FE and 148 FU-related genes exhibited favorable expression patterns in the accessions with favorable traits (Extended Data Figs. 3d and 4a). From short-fiber to long-fiber accessions, the number of genes with favorable expression patterns tended to increase in accessions with longer fiber (Pearson coefficient: 0.621, $P < 2.2 \times 10^{-16}$; Extended Data Fig. 4b,c). Similar observations were found for other traits (Extended Data Fig. 4d–f).

We next explored the genetic effect of regulatory variants associated with genes in TWAS and colocalization analysis. Most of the loci showed moderate effects with a median FL of 0.48 mm, FS of 1.14 cN tex⁻¹, FE of 0.04% and FU of 0.36% (Fig. 2f,g). We found that 534 genes showed pleiotropic effects (Extended Data Fig. 4g), such as the regulatory variant of Ghir_D06G018130 that contributed to both FS and FE with very large effects (FS: 2.30 cN tex⁻¹; FE: 0.07%). We delineated the growth trend of heritability by considering different numbers of trait-related loci through random sampling. The estimated heritability grew smaller as the number of loci increased, in a logarithmic manner (Fig. 2h). The phenotypic variance explained by the integration of TWAS and colocalization genes was much more than could be explained by just considering GWAS loci (Fig. 2i).

## Regulatory modules underpinning fiber quality

Because one eQTL may regulate multiple eGenes and one gene may also be regulated by multiple eQTLs, this relationship becomes more complicated when considering the presence of eQTL hotspots. We identified 406 eQTL hotspots that regulated 4,689 genes across five fiber

**Fig. 2 | Fine-mapping of fiber quality associations. a**, Manhattan plot of the genome-wide association study (top panel) and transcriptome-wide association study (bottom panel) for fiber quality. Significant QTLs are labeled. Significance thresholds of $P = 3.76 \times 10^{-7}$ (one-sided $F$ test) and FDR = 0.05 ($P$ value of two-sided Student's $t$ test corrected by FDR) were used, respectively. **b**, Phenotypic effects (TWAS $z$ score or correlation between expression and phenotype) of FL/FS-related candidate genes. Strategies are shown at the left, with illustrative color ranges. Positive $z$ scores or correlations are shown in orange and negative values in purple. Significant genes are marked with 'check mark.' **c**, Regional association plots for FL (top row) and eQTLs for gene Ghir_D05G007220 (*BB2*) at 12 DPA and 20 DPA. Chromosomal location and gene position were labeled at the bottom. The lead SNPs are highlighted with a purple diamond. Boxplots on the right panel show fiber length and *BB2* expression for accessions with different genotypes ($n = 154$ versus 215; two-sided Wilcoxon rank-sum test; centerline,

median; box limits, first and third quartiles; whisker, 1.5× interquartile range). **d**, Expression profiles of Ghir_D05G007220 (*BB2*) in 340 accessions were divided into eight expression groups. The figure shows the four larger expression groups. Heatmaps showing normalized FPKM in each accession at each timepoint. Line charts showing the mean expression of accessions at each timepoint. The arrow points to the timepoint with the highest mean expression. **e**, Boxplot for fiber length of accessions with different expression patterns ($n = 16$ versus 88 versus 108 versus 109; two-sided Student's $t$ test; centerline, median; box limits, first and third quartiles; whisker, 1.5× interquartile range). **f,g**, Dot plot for the genetic effect of variants associated with FL/FS-related candidate genes. $X$ axis indicates the variants sorted by genetic effect. $Y$ axis indicates the genetic effect of variants. The representative variants and regulated genes are labeled. **h**, The heritability accumulation accompanying the increase of genetic variants. **i**, The heritability is explained by only GWAS loci and both GWAS loci and eQTLs.

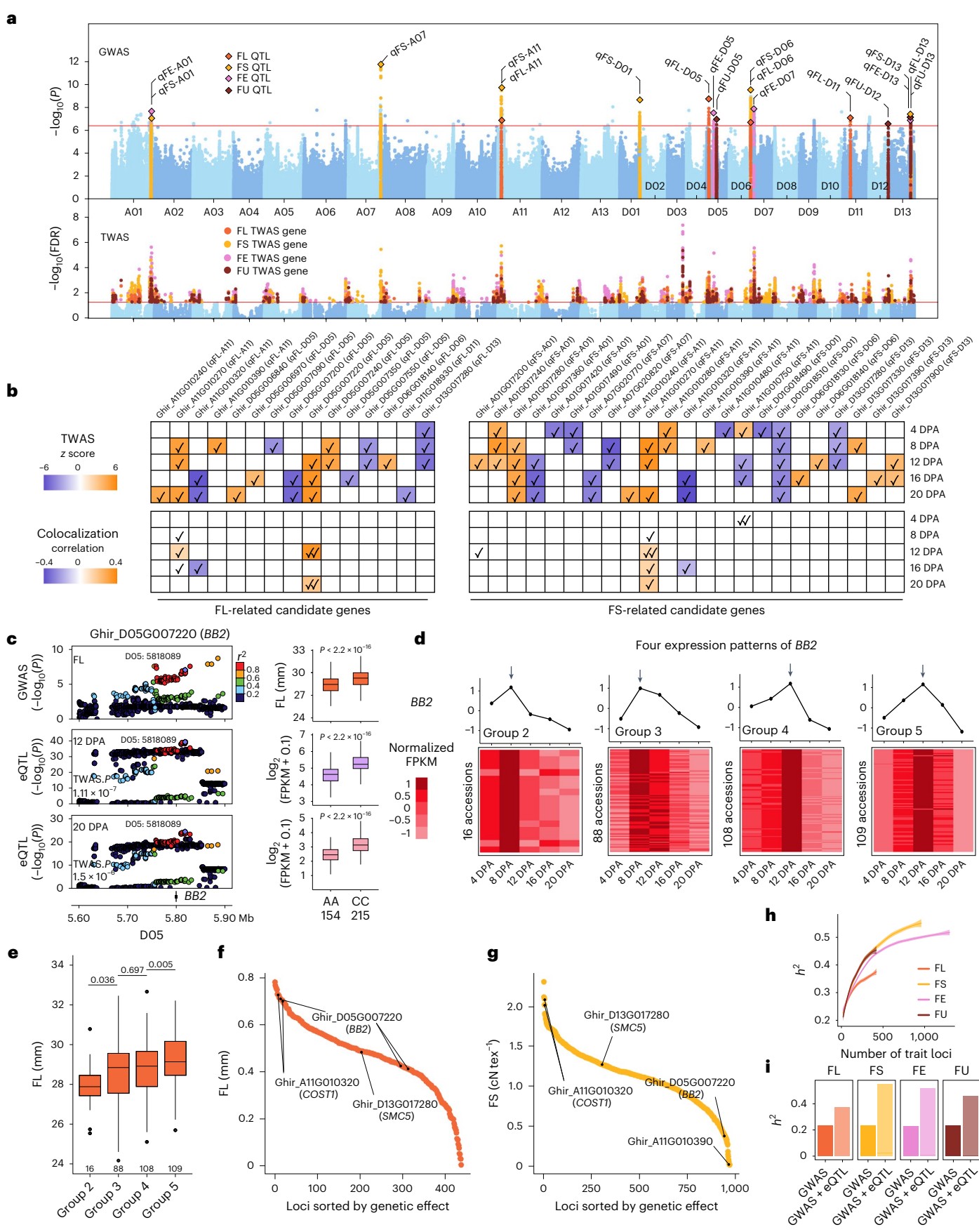

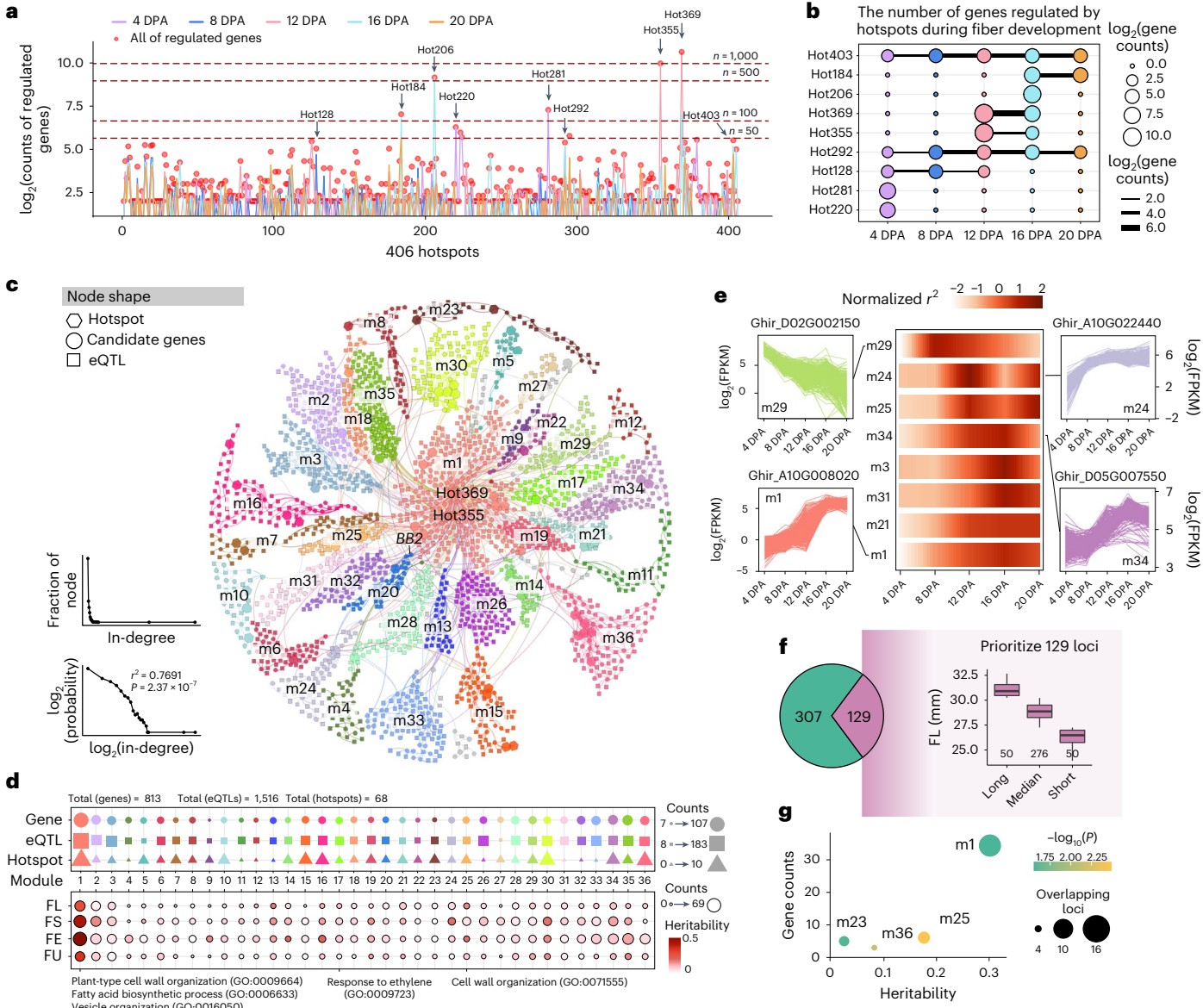

**Fig. 3 | eQTL hotspots and genetic network of genes associated with fiber quality. a**, Line chart showing the number of genes regulated by eQTL hotspots at each timepoint. Red point, the total number of regulated genes across five timepoints. **b**, Bubble plot showing the variable number of genes regulated by nine hotspots in fiber development. Point size, the scaled number of regulated genes in each timepoint; line size, the scaled number of genes that were coregulated between adjacent timepoints. **c**, Genetic network of fiber-related genes. The node color represents the module to which it belongs. Hexagon node, eQTL hotspot; circular, candidate genes; square, eQTL. The edge is used to connect hotspots/eQTLs with genes. Line plot, distribution of node in-degree values (one-sided $F$ test). **d**, The counts of genes, eQTLs and hotspots (top) in 36 modules and module heritability of four fiber quality-related traits (bottom). **e**, Analysis of dynamic interpretation (normalized $r^2$) of fiber length in eight modules at five timepoints. The expression patterns of four representative genes are shown with line plots. **f**, A total of 129 loci with higher favorable allele frequency (>0.2) in the accessions with the longest fiber ($n = 50$) compared with the accessions with the shortest fiber ($n = 50$). Centerline, median; box limits, first and third quartiles; whisker, 1.5× interquartile range. **g**, Module enrichment analysis of the 129 loci in **f** (one-sided Fisher's exact test).

developmental timepoints (Fig. 3a), including 283 timepoint-specific hotspots and 123 timepoint-shared hotspots, of which nine representative large hotspots were found to regulate a variable number of genes in fiber development ranging from 4 (Hot292 at 4 DPA) to 1,546 genes (Hot369 at 12 DPA; Fig. 3b). The eQTL/hotspot–eGene relationships constituted a comprehensive genetic network (Fig. 3c and Supplementary Tables 9 and 10j). We found that the in-degree distribution of this network followed a linear trend in the log scale ($r^2 = 0.7691$, $P = 2.37 \times 10^{-7}$), which is a landmark of scale-free networks (Fig. 3c and Extended Data Fig. 4h)[32]. In this network, all nodes (eQTLs/hotspots or eGenes) were clustered into 36 modules according to the connectivity between nodes.

Each module shows diverse heritability for four traits (Fig. 3d). We identified 23 FL, 25 FS, 26 FE and 25 FU-related modules with heritability ≥0.05 for each trait (Supplementary Table 10). For example, module 25 shows the highest heritability for FL and FE and module 24 shows the highest heritability for FS. This indicates that modules may differ with respect to phenotypic effects, possibly because different modules are controlled by different regulatory factors such as transcriptional factors (Extended Data Fig. 4i)[33]. In all modules, 35.3–57.4% of eQTLs regulated candidate genes at a specific timepoint, 11.8–38.2% of eQTLs had steady regulatory effects on candidate genes at all timepoints and the other eQTLs (16.8–37.3%) showed variable regulation at two to four timepoints (Extended Data Fig. 4j).

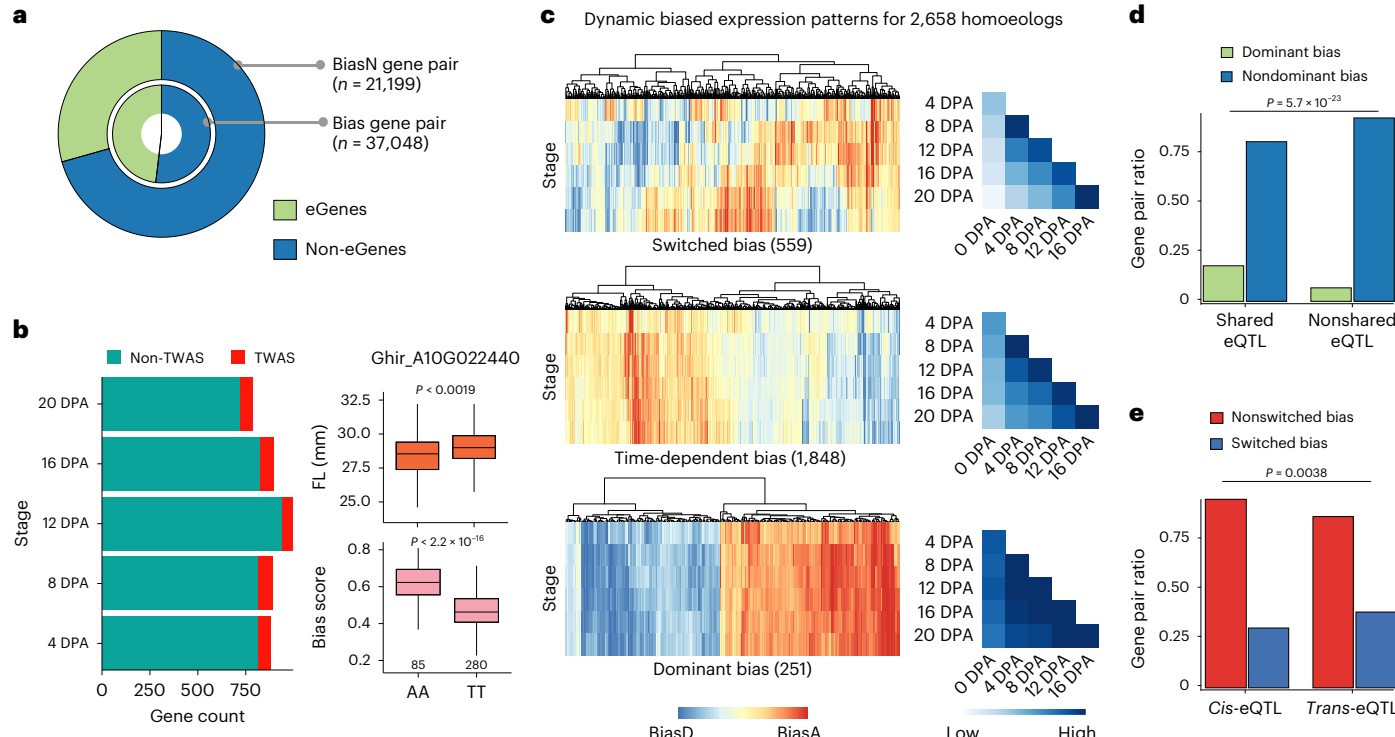

**Fig. 4 | Genetic regulation of homoeologous expression bias. a**, The circular diagram shows the proportion of homoeologous gene pairs that were characterized as eGenes and non-eGenes. The inner circle shows homoeologous pairs with expression bias, and the outer circle indicates gene pairs without expression bias. **b**, The number of TWAS genes present in the bias–eQTL gene pairs. Boxplots on the right panel show fiber length and homoeologous expression bias level at 20 DPA for the two haplotypes (n = 85 versus 280; two-sided Wilcoxon rank-sum test; centerline, median; box limits, first and third quartiles; whisker, 1.5× interquartile range). **c**, The heatmap on the left shows

switched, time-dependent and dominant bias patterns for 2,658 gene pairs across stages. The heatmap on the right indicates the correlation (Spearman coefficient) of the eQTL effect for 2,658 gene pairs between stages. **d**, Histogram shows a significant difference (based on one-sided Fisher's exact test) in the proportion of gene pairs showing dominant expression bias with shared eQTLs or without shared (nonshared) eQTLs. **e**, Histogram indicates a significant difference (based on one-sided Fisher's exact test) in the proportion of gene pairs with switched expression bias in specific cis-eQTLs and trans-eQTLs.

To examine the dynamic effect of modules on FL, the eight modules (heritability >0.05 and FL-related gene counts ≥5) with higher interpretation (normalized $r^2$) to FL were tabulated (Fig. 3e). For example, genes in module 29 are mainly involved in regulation at the early elongation stage (4–8 DPA), such as Ghir_D02G002150 that encodes a xyloglucan endotransglucosylase/hydrase involved in primary cell wall extension[34]. The genes in module 1 are mainly involved in regulation at a later elongation stage (16–20 DPA), including Ghir_A10G008020, which encodes a plant glycogen-like starch initiation protein that has a role in secondary cell wall biosynthesis (Fig. 3e)[35].

To investigate the extent of favorable allele aggregation in each module, we divided cotton accessions into three categories by their FL, that is, a long-fiber group (31.08 ± 0.68 mm), an intermediate-fiber group (28.85 ± 0.80 mm) and a short-fiber group (26.31 ± 0.82 mm). Among the 436 loci associated with FL, 129 show a higher favorable allele frequency (>0.2) in the long-fiber group compared with the short-fiber group, and these loci are enriched in modules 1, 23, 25 and 36 (Fisher's exact test, P < 0.05; Fig. 3f). Accordingly, modules 1 and 25 have a very high heritability to FL, representing strong candidates for prioritization in cotton breeding (Fig. 3g).

**Genetic effects on homoeologous expression bias**

In polyploid cotton, subgenomic expression bias occurs for a number of homoeologous genes; however, the cis- or trans-transcriptional regulatory controls of this bias are not well understood. To investigate the genetic basis of expression bias of homoeologous genes, we performed

GWAS using bias fraction score as a phenotype (Supplementary Fig. 12). A total of 14,133 significant associations (bias-eQTLs) were detected for 4,026 homoeologous gene pairs during fiber development.

To explore the possibility that bias-eQTLs contribute to homoeologous expression bias, we compared the genetic effects of eQTLs in one subgenome with those of bias-eQTLs for homoeologous genes, after discovering that eQTLs were significantly enriched for homoeologous pairs exhibiting expression bias (17,828, 48.1% versus 6,225, 29.3%; Fig. 4a and Supplementary Fig. 13). We observed that 2,658 gene pairs with 5,350 significant signals that were detected in both the eQTL and bias–eQTL analyses tended to have a wider pattern of expression bias in different accessions and their cis-regulatory regions had a higher proportion of variance for expression bias (Extended Data Fig. 5a,b). Of the 5,350 bias-eQTLs, 4,846 (90.5%) were colocalized with eQTLs for one of their homoeologous genes, and the number of variants in genic regions showed noticeable differences (Extended Data Fig. 5c). This suggests that variants in the transcriptional regulation region for one copy of homoeologous gene may result in dysregulation of expression and lead to expression bias. Of note is the observation that for gene pairs with bias-eQTLs identified at each stage, 8% (6.2–9.3%) of them are TWAS genes. There is an association between the variation of their homoeologous expression and FL change (Fig. 4b), suggesting that some homoeologous pairs with biased expression were implicated in the regulation of traits.

To further interrogate the impact of genetic variants on the dynamics of expression bias during fiber development, we clustered the 2,658 gene pairs into three groups (switched, time-dependent and dominant)

according to the number of accessions belonging to the patterns of expression bias (Extended Data Fig. 5d; Methods). Gene pairs belonging to the switched group exhibit changes in the direction of expression bias at different stages, which had relatively low correlation of genetic effects of eQTLs across stages (Fig. 4c). The dominant group shows a higher proportion of shared eQTLs, indicating that eQTLs often contribute to this shared expression pattern (Fig. 4d). Interestingly, compared with gene pairs with nonswitched bias of direction, more gene pairs in the switched group had *trans*-eQTLs instead of *cis*-eQTLs (Fig. 4e). This may suggest that *trans*-eQTLs were prone to mediate the dynamics of expression bias direction and *cis*-eQTLs were more likely to contribute to stable direction of expression bias during fiber development. In addition, by constructing a co-expression network of 16,081 homoeologous genes with expression bias, we found that homoeologous genes in the switched group have lower network connectivity that probably represents simpler regulatory relationships (Extended Data Fig. 5e–j and Supplementary Table 11). These findings may facilitate further understanding of the genetic regulatory dynamics underlying the expression bias of homoeologous genes.

### Subgenomic coordination of genetic effect on fiber quality

Fiber quality-related genes were identified in both subgenomes; however, the genetic contribution of their homoeologous copies to fiber quality is not well understood. In this study, we found that a very small proportion of both homoeologous copies (0.8%, 2 of 241) were identified as candidate genes for FL in the TWAS and colocalization analysis. Similar results were obtained for FS (2.7%, 14 of 518), FE (3.7%, 26 of 700) and FU (0.8%, 2 of 240). We identified pseudoregulatory sites by mapping the significant SNPs of each candidate gene to the sequences flanking the homoeologous gene (2 Mb upstream and downstream) in the other subgenome. This analysis showed that few pseudoregulatory sites (2 of 2,442) for the four fiber-quality traits are mutated in the population (Extended Data Fig. 6a–d).

Due to the relative paucity of regulatory variants, we evaluated the contribution of the homoeologous gene of each candidate gene by comparing their expression levels. For each gene that positively regulates fiber quality, we assume the candidate gene and its homoeologous gene have similar regulatory effects when the candidate gene is associated with a trait-beneficial genotype and the homoeologous gene has higher or no differences in expression level (favorable expression). Based on this, the contribution of each homoeologous gene pair to fiber quality was grouped into the following four models: (1) favorable homoeologous pairs, with both favorable genotype and expression; (2) only favorable genotype with unfavorable expression of its homoeologous gene; (3) only favorable expression that has unfavorable genotype of candidate gene and (4) unfavorable homoeologous pairs (Methods). We found that 29.5% of the 133 homoeologous pairs that positively regulate FL have favorable expression, 34.9% do not, 3.1% lack a favorable genotype and 32.6% lack both favorable genotype and expression (Fig. 5a). Among the 106 homoeologous pairs that negatively regulate FL, 27.5% have the favorable state, 19.6% show decreased expression of homoeologous genes, 8.8% do not have a favorable genotype and 44.1% lack both favorable genotypes and expression levels (Fig. 5b). These data show that the majority (67.5–75.8%) of homoeologous genes might be further optimized for fiber improvement (Extended Data Fig. 6e–g). At the module level, module 31 appears to be relatively optimized with respect to homoeologous gene expression, indicating that most modules might be promising targets for improvement through optimizing both homoeologs (Fig. 5c and Supplementary Figs. 14–16).

To assess the impact of aggregating homoeologous pairs with favorable genotype or expression on fiber quality traits, we counted the number of gene pairs represented by the four models above in each accession. For FL, we observed that the number of gene pairs categorized as favorable homoeologous pairs in long-fiber accessions became generally larger than in short-fiber accessions (Fig. 5d).

We also found that 81.6% of the TWAS signals (308) corresponding to 231 homoeologous pairs showed changes of expression bias fraction score between accessions with favorable and unfavorable genotypes (Fig. 5e). Interestingly, by counting the bias levels of homoeologous pairs in each accession, we found that with the aggregation of favorable genotypes in long-fiber accessions, the expression bias exhibited the following two patterns: either fewer gene pairs (Fig. 5f) or more gene pairs (Fig. 5g) showed biased expression. The former implicates that only one subgenome has favorable genotypes and the latter might result from neither of subgenomes having favorable genotypes in short-fiber accessions. Similar patterns were observed for the other three traits (Extended Data Fig. 7a–c). These results suggest that the targeted utilization of both subgenomes may enhance the potential for fiber improvement and that aggregation of favorable genotypes has led to changes in the transcriptional regulation of homoeologous genes.

### Genomic design for fiber quality improvement

To expand insight into the aggregation of favorable alleles for fiber improvement, we evaluated the effect of domestication selection on fiber quality-associated loci. In this analysis, 3,552 cotton accessions, including 332 landraces and 3,220 cultivars, were collected (Supplementary Table 12)[20,23,36,37]. A library of favorable alleles was constructed using the fiber quality-associated loci and these 3,552 accessions (Fig. 6a). Overall, cultivars aggregated more favorable alleles compared with landraces (Extended Data Fig. 8a,b and Supplementary Table 13), providing indirect ex post facto confirmation of the utility of focusing on the regulation of homoeologs related to fiber traits.

We calculated the sharing ratio of favorable alleles in 3,220 modern cultivars and grouped the trait-associated loci into four categories, C1–C4. From C1 to C4, the utilization level of favorable alleles in modern cultivars increased (Fig. 6b–d and Extended Data Fig. 8c,d). We also compared the differences of the four categories between cultivars and landraces (Fig. 6e and Extended Data Fig. 8e). Based on the differences in allele frequency (>0.6), we identified 91 selectively favorable alleles in cultivars (Supplementary Table 14). We also identified 48 loci for which the favorable alleles were under purifying selection, indicating that human selection did not always lead to the aggregation of favorable alleles (Fig. 6f). We observed no linkage drag effects for the loci under purifying selection, and the effect of these loci was no different to that of preferentially aggregated loci in cultivars (Supplementary Fig. 17). Nevertheless, we found that many of the genes corresponding to the loci under purifying selection showed pleiotropic effects (27/48; Supplementary Table 14). For example, Ghir_A01G013620, encoding a nicotinamide adenosine dinucleotide (phosphate) (NAD(P))-linked oxidoreductase superfamily protein, was characterized as a representative candidate for fiber development. The expression of this gene at 4 and 8 DPA was positively correlated with FL, FS and FE (Fig. 6f,g). This type of functional implication suggests that future fiber improvement efforts should consider enabling the use of these alleles.

A strong linear relationship was observed between the total number of favorable alleles and the phenotype, indicating that combining favorable alleles in elite lines is an effective way to design favorable cotton cultivars (Extended Data Fig. 9a). Ridge regression was used to estimate the overall effect of the trait-associated loci (Extended Data Fig. 9b,c). The correlation coefficient between predicted and observed values for FL reached 0.77, indicating these loci are predictive (Fig. 6h). The accuracy of the model was evaluated using an external GWAS cohort ($n = 1,040$)[37], and the correlation coefficient reached 0.47, which was similar to the accuracy of ridge regression best linear unbiased prediction (rrBLUP) by using the SNP set filtered through linkage disequilibrium (LD) clumping method (GWAS *P* value threshold = 0.001; Fig. 6i and Supplementary Fig. 18)[38]. We also predicted the best value of FL, which was 35.55 mm, much longer than the normally field-produced 28 mm fibers. The distance of the 376 accessions to the predicted best FL ranged from 2.88 mm to 11.64 mm (Fig. 6j).

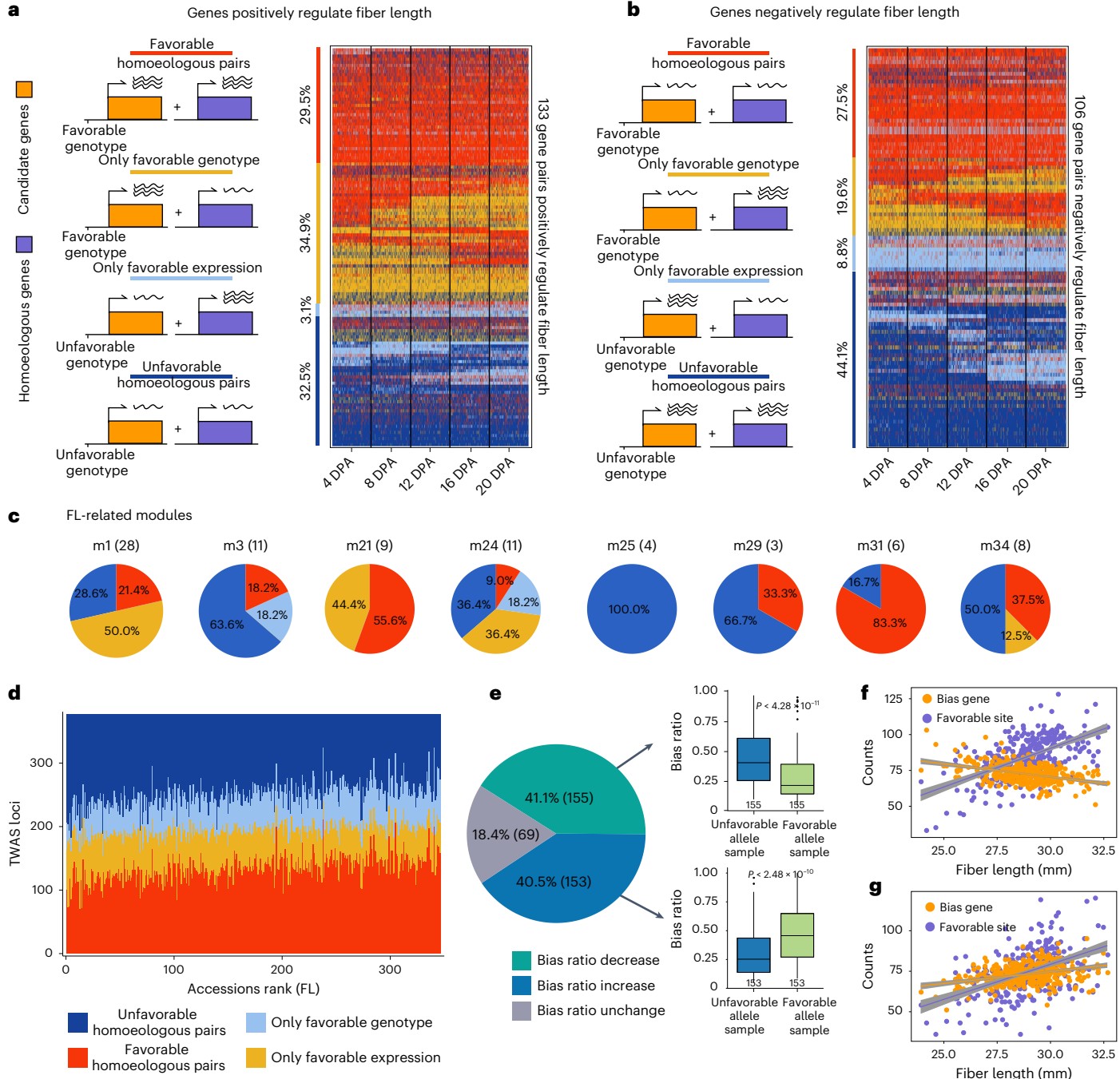

**Fig. 5 | Relationships between homoeologous expression and fiber quality.**
**a**, A total of 133 homoeologous gene pairs positively regulate fiber length. Red, favorable homoeologous pairs (homoeologous gene pairs with both favorable genotype and favorable expression). Yellow, homoeologous gene pairs with only favorable genotype of the candidate gene. Light blue, homoeologous gene pairs with only favorable expression. Dark blue, unfavorable homoeologous pairs (homoeologous gene pairs without either favorable genotype or favorable expression). Heatmap shows the classification of 133 gene pairs in 340 accessions at five timepoints. **b**, A total of 106 homoeologous genes negatively regulate fiber length. The classification is the same as in **a**. **c**, Pie charts of the states of homoeologous genes in eight FL-related modules. The number of FL-related genes is shown in the parentheses for each module. **d**, Proportional distribution of four types of homoeologous genes in accessions that were sorted by FL from short to long. **e**, Comparison of bias score of gene pairs in samples with favorable and unfavorable alleles. The pie chart shows the ratio of gene pairs with different patterns of bias score change. The boxplots indicate gene pairs that show a significant difference in bias score. Centerline, median; box limits, first and third quartiles; whisker, 1.5× interquartile range. **f,g**, Correlation between the number of gene pairs with biased expression, the number of favorable loci and fiber length. Panel **f** shows the accumulation in a single accession of 155 gene pairs that show a decrease in bias score in **e**. Panel **g** shows the accumulation in a single accession of 153 gene pairs that show an increase in bias score in **e**. The gray band represents 95% confidence interval for the fitted regression.

Because the expression of the homoeologous copies of trait-associated candidate genes may affect the phenotypes, we hold the view that fiber quality-related traits can be further improved by modifying the expression of homoeologous genes. To estimate the overall contribution of homoeologous gene expression, we integrated the expression of the homoeologous genes that were correlated with fiber quality-related traits in at least one developmental stage into the predictive model. This gave an increased prediction accuracy of

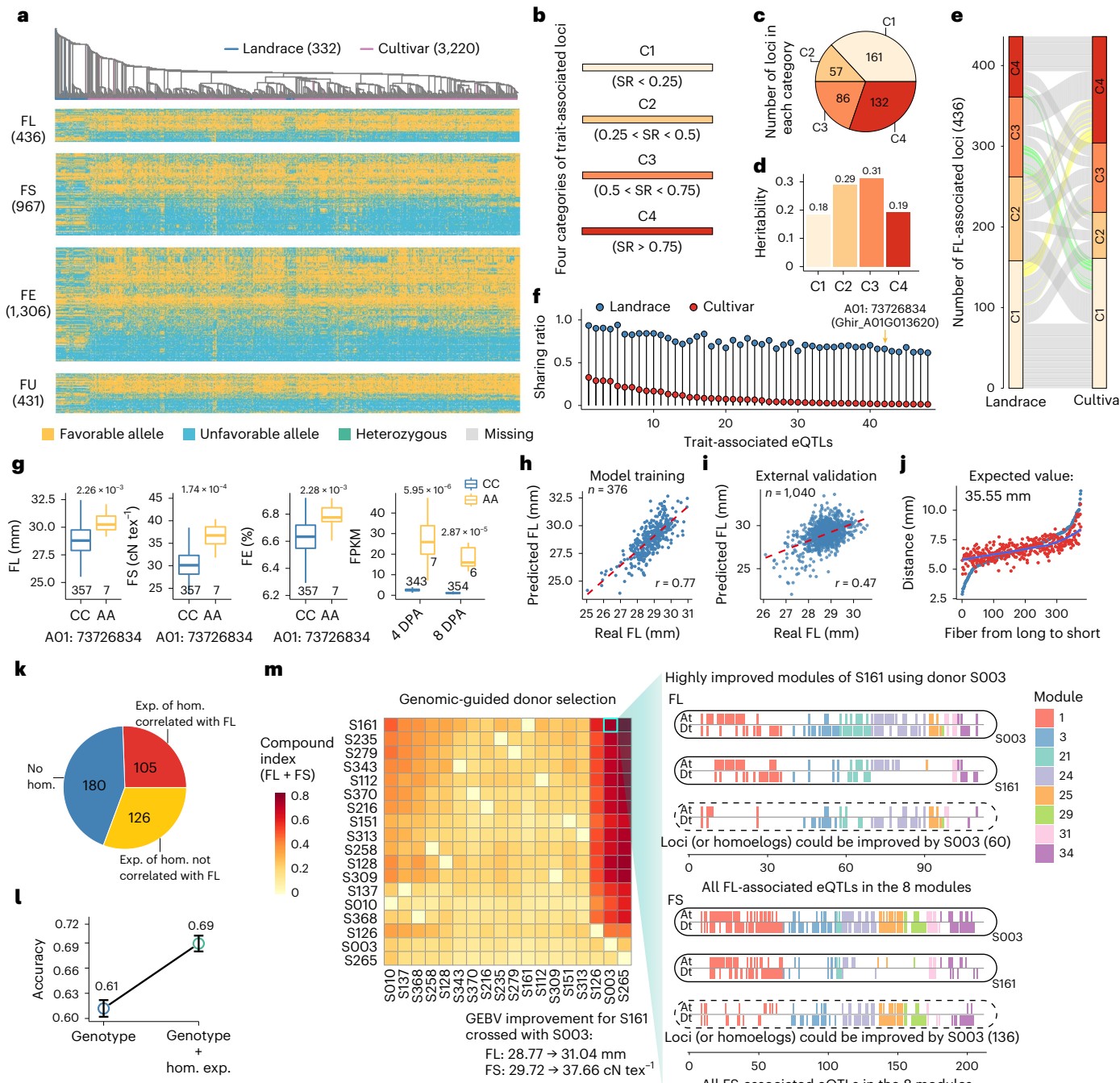

**Fig. 6 | Genomic design for fiber quality improvement. a**, Phylogenetic tree and genotypes of fiber-related loci in 332 cotton landraces and 3,220 cultivars. **b**, Trait-associated loci were grouped into four categories according to their favorable allele frequency in the modern cultivar population. **c**, Number of loci in four categories with different sharing ratios of favorable allele for FL. **d**, Heritability of the SNPs in each of the four categories for FL. **e**, Comparison of favorable allele frequency in the four categories between landrace and cultivar population. The frequency difference above 0.6 was highlighted in green and yellow. **f**, Favorable allele frequency of loci that were highlighted in green color in **e**. **g**, Comparison of fiber traits (FL, FS and FE) and expression levels of Ghir_A01G013620 in accessions with different genotypes (CC and AA). Two-sided Wilcoxon rank-sum test. Centerline, median; box limits, first and third quartiles; whiskers, 1.5× interquartile range. **h**, Correlation between real FL values and FL breeding value (BV) estimates calculated by ridge regression. **i**, Model predictability tested by applying the training model in **a** to an external dataset

(1,040 accessions) for FL. **j**, Distance of real FL (blue dots) and FL BV estimates (red dots) of the accessions to the estimated best value of FL. **k**, The number of FL-associated genes and the number of the homoeologous genes whose expression was correlated or not correlated with FL. **l**, Fivefold cross-validated prediction accuracy using ridge regression model based on two predictive sets. One contains genotype and the other contains both genotype and the expression of corresponding homoeologous genes. Hundred replications were run. Circle, mean; error bar, mean ± s.e. **m**, Genomic-guided donor parent selection. The heatmap (left) shows the level of genetic improvement. Each row represents the accession to be improved and each column represents the donor. Right, the utilization level of FL/FS-associated loci in the eight genetic modules or the homoeologous genes corresponding to these loci of S161 and S003. The loci or the homoeologous genes of S161 that could be improved by S003 were listed at the bottom.

8% for FL, 10% for FS, 5% for FE and 6% for FU (Fig. 6k,l and Extended Data Fig. 9d). These data pinpoint the importance of the involvement of homoeologous genes and expression levels in genomic breeding.

To enable genomic design for FL and FS improvement, we evaluated the trait-associated loci, as well as the state of utilization of the corresponding homoeologous genes that could be improved by other accessions (Fig. 6m and Supplementary Tables 15–18). As a proof of concept, we present the improvement degree matrix of 18 cotton accessions to clarify the donor parent selection process. The accessions S003 and S265 would be the most commonly used donor parents in the first crossing. S161 could be improved at up to 165 loci by S003 for FL and 381 loci for FS, and as a consequence, the genomic-estimated breeding value could be increased by 7.9% for FL and 26.7% for FS. We also present the degree of utilization of S161 and S003, considering both genetic modules and homoeologous genes. A total of 80 and 197 loci (or the homoeologous genes) of S161 could be improved by S003 in the eight genetic modules for FL and FS, respectively (Fig. 6m).

## Discussion

Precise spatiotemporal regulation of gene expression by both *cis*-regulatory sequences and *trans*-acting factors is required for developmental programs in higher organisms[39]. In this study, we characterized potential *cis*- and *trans*-regulatory variants of gene expression across different stages of cotton fiber development by eQTL mapping, which provides a rich resource for the community to identify genetic regulatory components associated with fiber quality. We show that a large proportion of eQTLs showed stage-dependent regulatory effects, similar to the increasing number of observations that many regulatory variants are not associated with gene expression at a steady state[21,40–42]. This finding suggests that the genetic effect of genes on fiber quality should be evaluated at a specific developmental stage. Future studies might explore the functional implications of *cis*-eQTLs represented by variants in transcription factor binding sites, and also ascertain whether key transcriptional factors are mutated in *trans*-eQTLs, the latter requiring the implementation of a cotton Encyclopedia of DNA Elements project[43].

In polyploid cotton, the effect of *cis*- and *trans*-interactions that lead to both intrasubgenomic and intersubgenomic interactions on the expression of duplicated (homoeologous) genes is largely unresolved[8,13]. We show that the transcriptional dysregulation mediated by *cis*-regulatory variants in a certain subgenome may give rise to homoeologous expression bias. This phenomenon may be addressed further from an evolutionary viewpoint to understand the phylogenetic timing and extent of homoeologous gene expression divergence relative to both polyploidization and cotton domestication. We further show that widespread partitioned expression of homoeologous genes during fiber development is accompanied by only 30.2% of fiber quality-related homoeologous copies with both favorable aggregations during breeding. This suggests the exciting prospect that fiber quality may be improved through subgenome optimization, harnessing genomic modifications of genetic regulatory loci associated with homoeologous expression partitioning. This study highlights that the dissection of the genetic basis of agronomic traits or identification of functional genes should consider subgenomic counterparts in polyploid crops. We note that although we do not provide experimental evidence demonstrating that subgenome optimization in breeding programs actually has improved fiber quality, we point to this important consideration here. In summary, genetic dissection of the regulation of homoeologous expression partitioning may lead to both fundamental and applied advances with respect to polyploidy and crop improvement.

## Online content

Any methods, additional references, Nature Portfolio reporting summaries, source data, extended data, supplementary information,

acknowledgements, peer review information; details of author contributions and competing interests; and statements of data and code availability are available at https://doi.org/10.1038/s41588-023-01530-8.

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

## Methods

### Plant materials

A population of upland cotton (*G. hirsutum*) containing 376 diverse accessions was cultivated in the field at Huanggang (114°55′22″E, 30°34′19″N), Hubei, China in 2019. Three replicates of each accession were cultivated on three separate plots. Cotton leaf samples of each accession from one plant were collected for genomic DNA extraction using the cetyltrimethylammonium bromide (CTAB) method[44]. Cotton bolls were marked on the day of flowering (0 DPA) for transcriptomic analysis. Ovules at 0 DPA and developing fibers at 4, 8, 12, 16 and 20 DPA were collected from at least ten cotton bolls of different plants in two rows and stored in liquid nitrogen until RNA extraction (see below). Mature cotton fibers were collected from each plant for fiber quality measurement.

### DNA sequencing and SNP calling

Purified genomic DNA was sonicated to about 250–300 bp, followed by pair-end repairing. Resequencing libraries were constructed following manufacturer protocols for the MGISEQ-T7 platform (Pair-end 150 bp reads; including 373 new DNA sequencing data and three published DNA sequencing data[20]: accession S112, SRR4018975; S123, SRR4018977; S280, SRR4018974). Fastp software (v0.23.0) was used to trim the adapter sequence and remove low-quality reads[45]. Variant calling was accomplished following the Sentieon pipeline (License 201808.07)[46]. In brief, Sentieon invoked BWA to align the clean sequencing reads to TM-1 reference genome[47] following customized parameter (mem -M -K 10000000)[48]. SAMtools (v1.9) was used to eliminate duplicated and low-quality reads alignment[49]. We performed SNP calling in two different ways. First, Sentieon Haplotyper algorithm (--algo Haplotyper --genotype_model multinomial --emit_conf 30 --call_conf 30) was used to process variant calling for each sample. The global variation file (GVCF) for each sample was generated by GVCFtyper algorithm (--algo GVCFtyper --emit_mode gvcf) in Sentieon, followed by joint calling to merge all variations into an integrated VCF file. Only the variations identified by both algorithms and covered by more than five sequencing reads were regarded as high confident and were retained.

### RNA sequencing and data analysis

For the cotton ovule and fiber samples, RNA was extracted using the RNAprep Pure Plant Kit (polysaccharides and polyphenolics-rich). In total, 2 μg RNA was used to construct sequencing libraries using VAHTS Universal V6 RNA-seq Library Prep Kit (Vazyme, NRM604-02) that were sequenced on an MGISEQ-2000 platform (pair-end 150 bp reads). After removing adaptors and clipping low-quality bases with Trimmomatic (v0.36), clean reads were mapped to the reference genome of TM-1 using HISAT2 (v2.1.0)[50,51]. SAMtools (v1.9) was used to remove PCR duplicates and reads with mapping quality less than 20 (ref. 49). The remaining reads were used to calculate the expression level (FPKM) of genes with StringTie (v2.1.4)[52].

### Expression bias analysis of homoeologous genes

For each accession at each stage, the expression change of homoeologous gene pairs was defined if at least one gene was expressed (FPKM > 0.1) and both genes exhibited an expression difference of at least twofold, which was tested under different conditions (Supplementary Table 3). For each expressed homoeologous gene pair in all accessions, if the number of accessions with expression bias (at the single accession level) was greater than 5% of all accessions, this gene pair was identified as biased homoeologous gene pair. For each homoeologous gene pair with biased expression, we compared the expression level at each timepoint between the At and the Dt in all accessions using a two-sided Wilcoxon rank-sum test and corrected the *P* value using the Benjamini–Hochberg method. Homoeologous gene pairs with expression bias in at least 5% accessions and FDR ≤ 0.05 were considered biased homoeologous gene pairs at the population

level. In this analysis, we classified the expression bias of gene pairs into two kinds, that is, bias-At and bias-Dt, with observed expression bias toward the At or Dt subgenome. For gene pairs showing expression bias toward the At or Dt subgenome in less than 5% of samples or with two bias directions, they were defined as bias-N or bidirectional-bias, respectively. In total, we identified 29,690 gene pairs categorized as bias-At, 31,604 as bias-Dt and 1,667 as bidirectional-bias in all six timepoints of fiber development.

### Peer factors for eQTL mapping

To account for the hidden batch effect and other global confounders, we used the probabilistic estimation of expression residuals (PEER, v1.3) method to estimate hidden covariates for gene expression levels for each stage[53]. This method can also increase the detection power of eQTL mapping by accounting for these covariates in the analysis. A set of PEER factors were tested, and the number of PEER factors was selected to maximize the number of mapped eGenes. Due to the similar number of samples at each timepoint, 20 PEER factors were chosen in six timepoints for the correction of hidden covariates (Supplementary Fig. 1).

### *Cis*-eQTL mapping

To identify eQTL for genes in fiber development, genes with low expression levels (FPKM < 0.1) in more than 95% of samples were filtered, and the expression levels of remaining genes of all samples from a given stage were normalized using an inverse normal transformation. *Cis*-eQTL mapping was performed using FastQTL (v7)[54], and the top three genetic principal components and 20 PEER factors were used as covariates[52]. By testing the association with variants within ±1 Mb of the transcription start site (TSS) for each gene, an adaptive permutation was used with the setting --permute 1000 10000. The significant variant–gene associations were identified by applying gene level nominal *P* value thresholds corresponding to FDR < 0.05. The significant SNP with the strongest association signal was defined as the lead SNP for each association. We subsequently performed a forward–backward stepwise regression analysis to identify multiple independent *cis*-eQTL signals for a given expression phenotype. The stepwise regression procedure was implemented in the conditionally independent QTL module of the software tensorQTL (v1.0.5), as described previously[55].

### *Trans*-eQTL mapping

For *trans*-eQTL mapping, we used the same covariates as for *cis*-eQTL analysis. The FAST-LMM (v0.2.32) program was used to perform GWAS of each gene and the whole genomic SNPs (MAF > 0.05)[56]. Variant–gene pairs with *P* value less than $3.76 \times 10^{-7}$ were considered significant, and the significant variants for each eGene were grouped into clusters with a maximum distance of 10 kb between two consecutive SNPs, and only those clusters with more than three SNPs were considered as a putative eQTL. We identified a total of 46,749 significant variant–gene associations in six stages, of which 22,938 had target eGenes with variants within ±1 Mb of the genic region. Because the corresponding variants were highly correlated with variants identified in FastQTL, we only kept the variant–gene associations where the variants were found out of the 1 Mb region of the eGene.

### Comparison of eQTL effect between stages

To estimate the cross-stage activity of *cis*-eQTLs, we used Multivariate Adaptive SHrinkage (as implemented in the R package mashr (v0.2.45)) for all *cis*-eVariant–gene pairs across stages[27]. This analysis was performed using the β coefficients and s.e. of each eQTL as input, and nine SNPs were randomly selected from the ±1 Mb region of each eGene. A total of 575,820 variant–gene pairs were used to fit the MASHR model. Effect size estimates and LFSR (that is, LFSR) outputted by mash were used as metrics of *cis*-eQTL magnitude and significance. An LFSR ≤ 0.05 was used as the threshold of significant *cis*-eQTL activity. To estimate

the activity of *trans*-eQTL across stages and test the robustness of mash, we used a meta-analysis approach as implemented in MetaSoft (v2.0.1) to calculate the posterior probability and *m* value that the eQTL effect exists in each stage[57]. An *m* value ≥ 0.9 was used as the threshold of significant eQTL activity. To assess the stage specificity of eQTL, we paid attention to significant eQTL identified at different stages that may be in the same LD genomic interval. To reduce duplicate statistics, significant eQTLs for each gene were merged based on the lead SNPs with $r^2 ≥ 0.6$ and distance ≤100 kb. For each eQTL, we focused on its lead SNP, and the effect of stage-shared eQTL was defined as the average effect of lead SNP in all stages.

### Detection of bias–eQTL

To identify significant variants associated with the expression bias of homoeologous gene pairs, the expression bias level in each accession of each stage was quantified into a biased score as shown below. The association between SNPs and bias score was performed using the factored spectrally transformed linear mixed models (FAST-LMM) program (Supplementary Fig. 12)[56].

$$\text{Bias score} = \frac{\text{At} - \text{Dt}}{\text{At} + \text{Dt}}$$

We applied a strict filter of homoeologous genes with expression bias–biased expression was present in at least 5% but no more than 95% of all samples, and the expression of the homoeologous genes differed by twofold change of FPKM. After filtering, an average of 5,579-6,563 homoeologous gene pairs in each stage were used for GWAS analysis. We clustered the significant SNPs according to the method described previously[58]. SNP clusters were further merged based on LD ($r^2 ≥ 0.6$) and distance (≤100 kb) in the genome. Across six timepoints, a total of 14,133 bias-eQTLs were identified for 4,026 homoeologous gene pairs. Genes regulated by eQTLs were significantly enriched in homoeologous genes with expression bias, so we compared the effect of all bias-eQTLs to their effect in eQTL analysis. The effect size was estimated by the β coefficient and s.e. of lead SNP in each bias–eQTL. When calculating the effect of bias–eQTL in regulating the expression of Dt-subgenomic genes, the effect value is correspondingly multiplied by −1.

### Analysis of heritability of gene expression

To estimate the variance of expression bias level explained by different genomic regions, genome-wide SNPs were divided into LD-friend SNPs (that is, target SNPs that are in significant LD) of *cis*-SNPs and *trans*-SNPs according to the eQTLs of each homologous gene pair. For LD-friend SNPs, GCTA-LDF (v1.94.0) was performed with parameters --ld-wind 100 and --ld-sig 0.05 to search for SNPs that are in LD with the lead SNP[59,60]. *Cis*-SNPs and *trans*-SNPs are distinguished according to whether the distance between SNP and TSS is more than 1 Mb. Each set was used to build a kinship matrix using the direct method and --power −0.25 in LDAK (v5.2)[61]. Considering the fact that different genome regions have different LD levels, the LDAK weightings model was used to equalize the tagging of SNPs in the genome. The genetic variance was then calculated for these SNPs using the restricted maximum likelihood (REML) model and −mgrm[61].

### Genome-wide association analysis

For GWAS, we used a total of 2,658,921 high-quality SNPs (MAF > 0.05) from the 376 accessions. Association analysis was carried out using a linear mixed model implemented in FAST-LMM for four fiber quality-related traits, including FL, strength, elongation and uniformity. The population structure was inferred using STRUCTURE software ($k = 3$, SNP number = 5,000) and included in the model as covariates. The kinship matrix was calculated based on all SNPs using FAST-LMM[56]. The threshold for genome-wide significance was set as $3.76 × 10^{-7}$ ($1/n$, where $n$ is the total number of genomic SNPs). The significant SNPs were

first grouped into one locus if two adjacent SNPs were within a 20-kb interval. Consecutive loci were further merged into a single locus if SNPs with the lowest *P* values in each of the adjacent loci were located in LD regions ($r^2 ≥ 0.6$).

### TWAS and colocalization analysis

To identify associations between gene expression and fiber quality-related traits, we conducted TWAS using the FUSION package[28]. Briefly, we constructed a standard binary PLINK format file for each gene using SNPs within 500 kb on either side of the gene boundary with the integration of log-transformed expression data. Then, the script FUSIONcompute_weights.R (https://github.com/gusevlab/fusion_twas) in the FUSION package[28] was used to compute the expression weights for each gene, taking the binary PLINK format file as input. Five models (BLUP, BSLMM, LASSO, Elastic Net and top SNPs) were used in this step and the effect sizes from these models acted as weights. The script FUSION.test.R (https://github.com/gusevlab/fusion_twas) was run to perform the typical TWAS analysis, with the computed gene expression weights, GWAS summary statistics and an LD reference panel that was constructed using the SNPs matching the GWAS SNPs. We considered genes with corrected *P* values < 0.05 as significant associations.

To detect the shared causal variants between eQTL and GWAS signals, we performed a colocalization analysis of *cis*-eQTLs from five fiber developmental stages and GWAS loci of four fiber quality traits using coloc[29]. The *cis*-eGenes within 1 Mb of the GWAS loci were extracted for colocalization analysis. The function coloc.abf was used to calculate posterior probabilities for PP.H4 that shows both traits (gene expression and GWAS phenotype) share a single causal variant. A *cis*-eGene was defined as having evidence of colocalization when the posterior probability of colocalization (PP.H4) was higher than 0.8. We implemented SMR approach to test if an eGene is associated with a trait through eQTL[30]. SMR (v1.03) implements the heterogeneity in dependent instruments (HEIDI)-outlier test to distinguish causality or pleiotropy from linkage. SMR associations were declared significant if the Bonferroni-corrected SMR *P* < 0.05 and HEIDI-outlier test *P* > 0.05.

### Definition of biased expression pattern in fiber development

To analyze the dynamic expression bias of homoeologous genes, we explored whether the direction (the At or Dt subgenome) of expression bias in each accession was consistent over six timepoints. For each gene pair, all cotton accessions were classified into the following eight classes according to the biased expression patterns in fiber development: class 1 indicates accession with the same bias direction across six timepoints; class 2 indicates accession with the same bias direction across five timepoints while having no bias in the remaining timepoint; class 3 indicates accession with the same bias direction across four timepoints while having no bias in the remaining timepoints; class 4 indicates accession with the same bias direction across three timepoints while having no bias in the remaining timepoints; class 5 indicates accession with the same bias direction across two timepoints while having no bias in the remaining timepoints; class 6 indicates accession with bias at a specific timepoint while having no bias in the remaining timepoints; class 7 indicates homoeologous genes had balanced expression in all six timepoints (without expression bias) or class 8 indicates the direction of expression bias is different in at least two timepoints. For each gene pair, the number of accessions in each classification was counted. According to the number of accessions in each classification, 2,658 gene pairs were sorted using unsupervised hierarchical clustering (Extended Data Fig. 5d). For the gene pairs of the switched cluster, most of their accessions are presented in the bias expression of class 8. For the gene pairs of the dominant cluster, most of their accessions show the same bias direction across six timepoints (class 1). For the gene pairs of time-dependent cluster, most of their accessions have no bias expression in at least one timepoint (classes 2–7).

## Genetic network construction

To explore the regulatory landscape of candidate genes during fiber development, we used eQTLs to constitute a genetic network that integrated regulatory relationships from all timepoints. This network was composed of eQTLs, eQTL hotspots and candidate genes identified in TWAS or colocalization analysis. To describe the dynamic eQTL regulation, eQTLs in all timepoints were merged when the distance between lead SNPs was less than 50 kb and $r^2$ was greater than 0.6. We obtained 16,914 merged eQTLs from 53,854 $cis$-eQTLs and 23,811 $trans$-eQTLs from six timepoints. eQTL hotspots were identified using the HOT_SCAN program at six timepoints[62] and then merged when the distance between adjacent hotspots was less than 20 kb. We identified a total of 463 hotspots across six timepoints and 406 hotspots across five fiber developmental timepoints. The visualization and module division of the network was accomplished by the built-in program in Gephi (v0.9.5)[63]. The principle of modularization was to maximize the connectivity between nodes in the same module and minimize the connectivity between nodes in the different modules. Each module consists of a collection of genes related to fiber quality (TWAS$_{FDR}$ < 0.05 or COLOC$_{PP,H4}$ > 0.8). Modules with heritability ≥0.05 and gene counts ≥5 were used for multiple regression of gene expression to phenotypic values across timepoints. For each module, the degree of correlation between gene expression and phenotype at a given timepoint was indicated by the magnitude of the normalized $r^2$ value. We identified 129 loci with higher favorable allele frequency (>0.2) in the top 50 accessions with the longest fiber. To determine whether the modules were enriched for the 129 loci, one-sided Fisher's exact test was performed in R.

## Conception of subgenome coordination model

In this study, favorable genotypes refer to those identified by TWAS or colocalization analysis and corresponding to higher trait values. For candidate genes that positively regulate phenotype, favorable expressions refer to cases where homoeologous genes have higher expression than candidate genes (with twofold expression change) or no differences. For candidate genes that negatively regulate phenotype, favorable expressions refer to cases where homoeologous genes have lower expression than candidate genes (with twofold expression change) or no differences. We classified the status of a pair of homoeologous genes in all accessions into the following four models: favorable homoeologous pairs, only favorable genotype, only favorable expression and unfavorable homoeologous pairs. Based on the above classification, a pair of homoeologous genes in accessions at all timepoints was classified—(1) favorable homoeologous pair, judged as favorable homoeologous pair in ≥50% accessions at all timepoints; (2) only favorable genotype, judged as favorable homoeologous pair or only favorable genotype in ≥50% accessions at all timepoints but did not meet the first condition; (3) only favorable expression, judged as favorable homoeologous pair or only favorable expression in ≥50% accessions at all timepoints but did not meet the first condition; (4) unfavorable homoeologous pairs, the remaining gene pairs that did not meet the above three conditions.

## Heritability estimation

To evaluate how the heritability changes when considering more trait-related SNPs, we first took subsets from all SNPs at different sizes by using the sample() function in R and then used the LDAK-Thin model to estimate the heritability contributed by these SNP subsets[61]. The LDAK-Thin assumes the expected heritability contributed by an SNP is higher for SNPs in regions with lower levels of LD and for those with higher MAF. The process of thinning the genetic variants cameters '--window-kb 100 and --window-prune 0.98.' Kinship matrix was calculated for each SNP subset using the main argument '--calc-kins-direct' by setting the power equal to 0.25. Heritability contributed by each kinship matrix was estimated separately using a generalized REML solver that was included in LDAK[61].

## Genomic prediction using ridge regression

In this study, ridge regression, a linear regression with penalty, was used to reduce overfitting[64]. The predictor variables were the number of superior alleles at each locus, encoded as 0, 1 or 2. The responding variable was the real phenotype. The function cv.glmnet() in glmnet package in R was used to fit the ridge regression model by setting α equal to 1 and the cross-validation will be automatically performed. The value of $\lambda$ (regularization parameter) that gives cross-validated minimum mean error was selected to obtain the model coefficients[65].

## Reporting summary

Further information on research design is available in the Nature Portfolio Reporting Summary linked to this article.

## Data availability

All raw sequencing data generated in this paper have been deposited into the National Center for Biotechnology Information database (BioProject ID: PRJNA917453 for DNA-resequencing data and PRJNA891378 for RNA-seq data).

## Code availability

All software used in the study are publicly available on the Internet as described in the Methods and Reporting Summary.

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

## Acknowledgements

This study was supported by the National Key Research and Development Program of China (2021YFF1000900 to M.W. and 2021YFF1000100 to X.Z.) and the National Natural Science Foundation of China (32170645 and 31922069 to M.W. and 31830062 to X.Z.). This study was also supported by the National Key Laboratory of Crop Genetic Improvement Self-research Program (ZW22B0204) and the Foundation of Hubei Hongshan Laboratory (2021hszd014). We thank the high-performance computing platform at the National Key Laboratory of Crop Genetic Improvement in Huazhong Agricultural University.

## Author contributions

M.W., X.Z. and L.T. designed the experiments and managed the project. L.T., J.Y. and Z. Liu conducted the cultivation of cotton plants. J.Y., Z. Liu, M.S., L.S., H.N., Y.P., X.L., M.Z., Y.H., X.C., X.H., Y.Z., R.P., Y.L., Q.M. and Q.Z. collected fiber samples and performed RNA sequencing. Y.M. and Z.Q. performed genome resequencing and SNP calling. M.W., J.Y. and Z. Liu performed GWAS, eQTL mapping, TWAS and colocalization analysis. Z. Liu and Z.Q. performed bias–eQTL analysis and genomic prediction analysis. Z. Lin, L.M. and Y.M. collected cotton accessions. J.L., D.Y., L.Z., D.D.F. and C.E.G. contributed to project discussion. J.Y., Z. Liu and Z.Q. wrote the manuscript draft and M.W., X.Z., J.F.W. and K.L. revised it.

## Competing interests

The authors declare no competing interests.

## Additional information

**Extended data** is available for this paper at https://doi.org/10.1038/s41588-023-01530-8.

**Correspondence and requests for materials** should be addressed to Lili Tu, Xianlong Zhang or Maojun Wang.

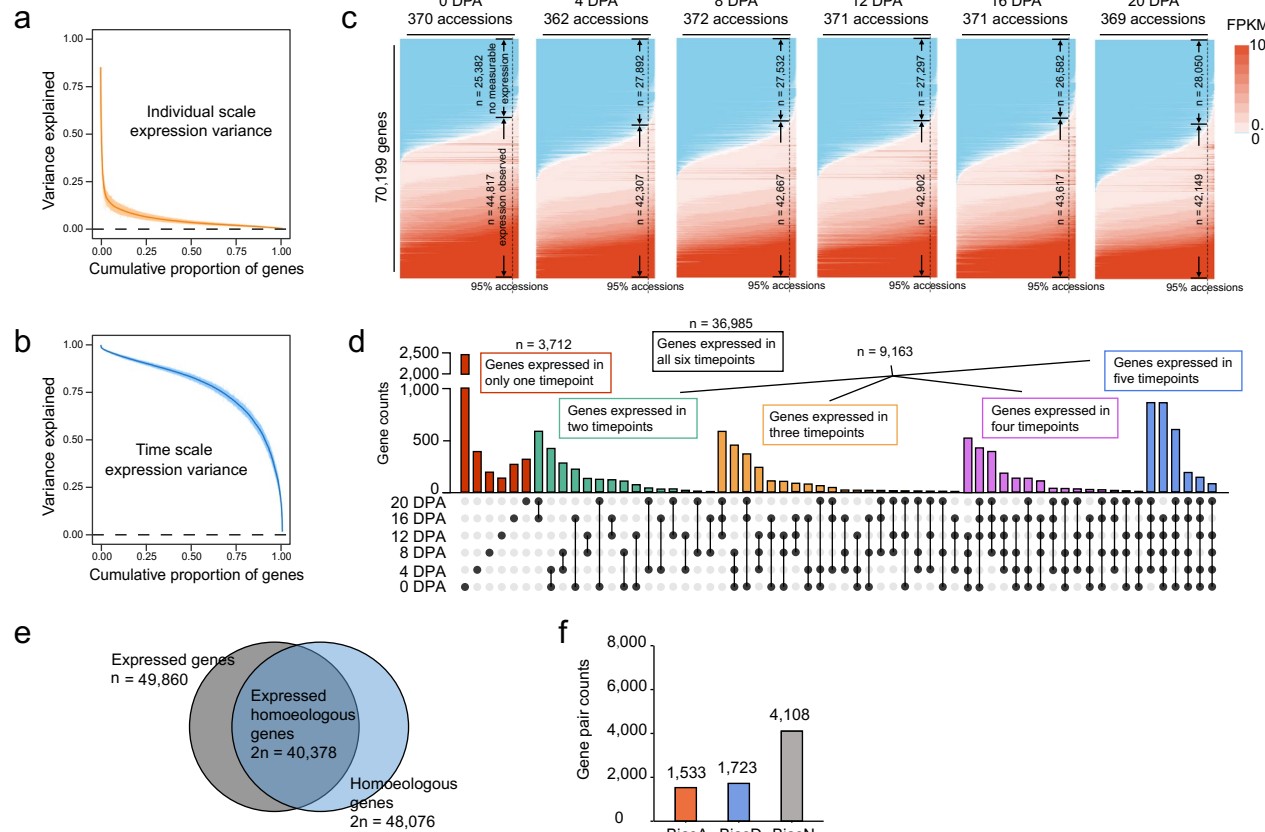

**Extended Data Fig. 1 | Gene expression atlas in fiber development.**
**a**, Proportion of gene expression variance explained by differences among accessions. Orange band represents 95% confidence interval for the fitted regression. **b**, Proportion of gene expression variance explained by differences among developmental timepoints. Blue band represents 95% confidence interval for the fitted regression. **c**, Heatmaps of gene expression in all accessions in ovules (0 DPA) and fibers (4 DPA, 8 DPA, 12 DPA, 16 DPA, 20 DPA). Genes were sorted by proportion of accessions with observed expression. **d**, Venn diagram showing the number of genes expressed in one to six timepoints. **e**, The number of expressed homoeologous gene pairs. **f**, The number of expressed homoeologous gene pairs with steady expression bias. Brown, expression bias towards the At (BiasA); blue, expression bias towards the Dt (BiasD); gray, without expression bias (BiasN).

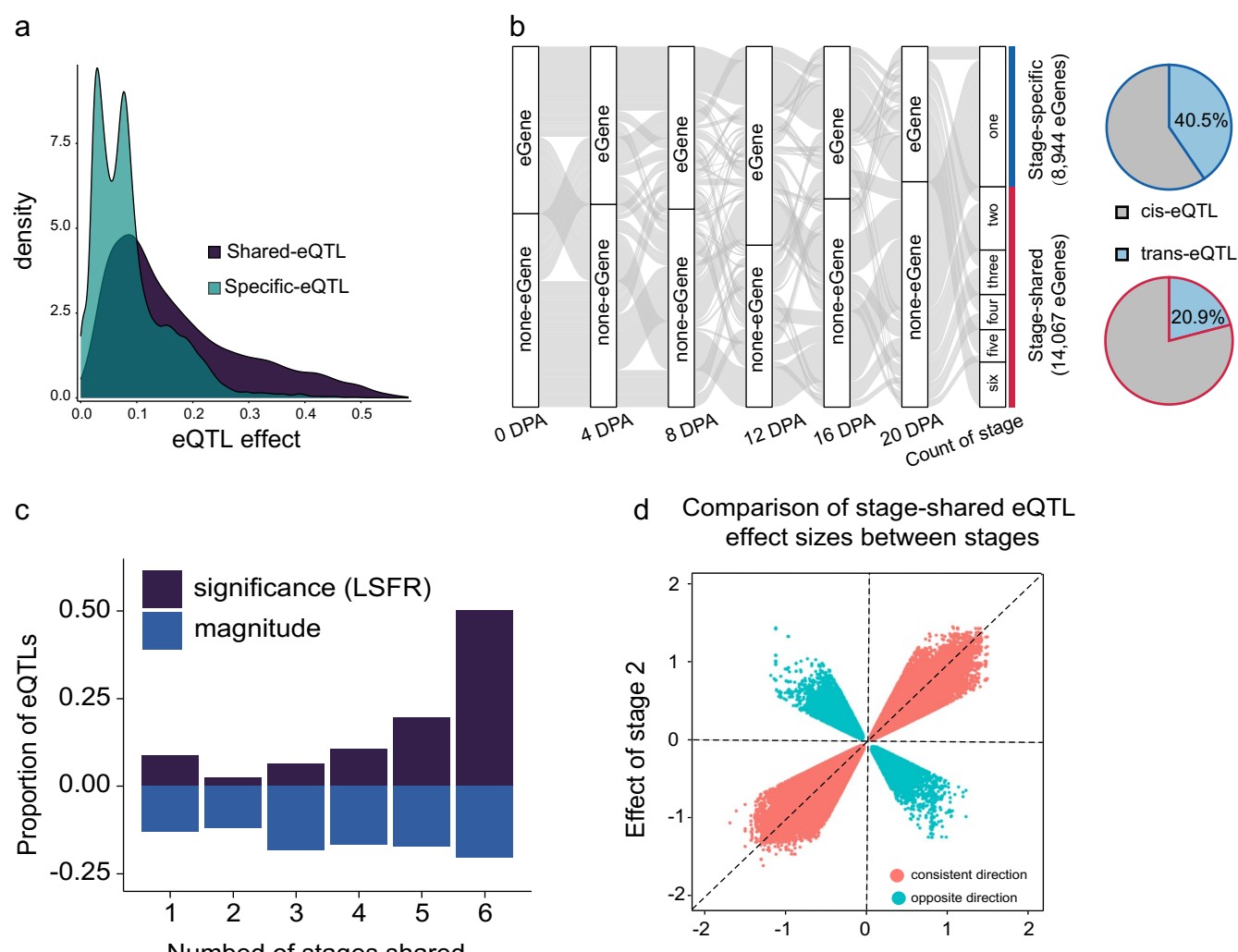

**Extended Data Fig. 2 | Characterization of eQTLs across stages. a**, Compared with specific eQTLs, shared eQTLs have markedly higher effects ($P < 2.2 \times 10^{-16}$, two-sided Wilcoxon rank sum test). **b**, Genes with stage-shared eQTL were linked to the number of shared stages. The pie diagram indicates the proportion of gene pairs that had different types of eQTLs. **c**, Distribution of the number of stages in which *cis*-eQTLs were shared by significance (up) or magnitude (down). *cis*-eQTL magnitude was defined to be shared when the effect estimate outputted by mash was within 2 folds. *cis*-eQTL was defined significant with a mash local false sign rate (LFSR < 0.05). **d**, Comparison of effect sizes of stage-shared eQTLs between stages. The effect size was estimated by mash.

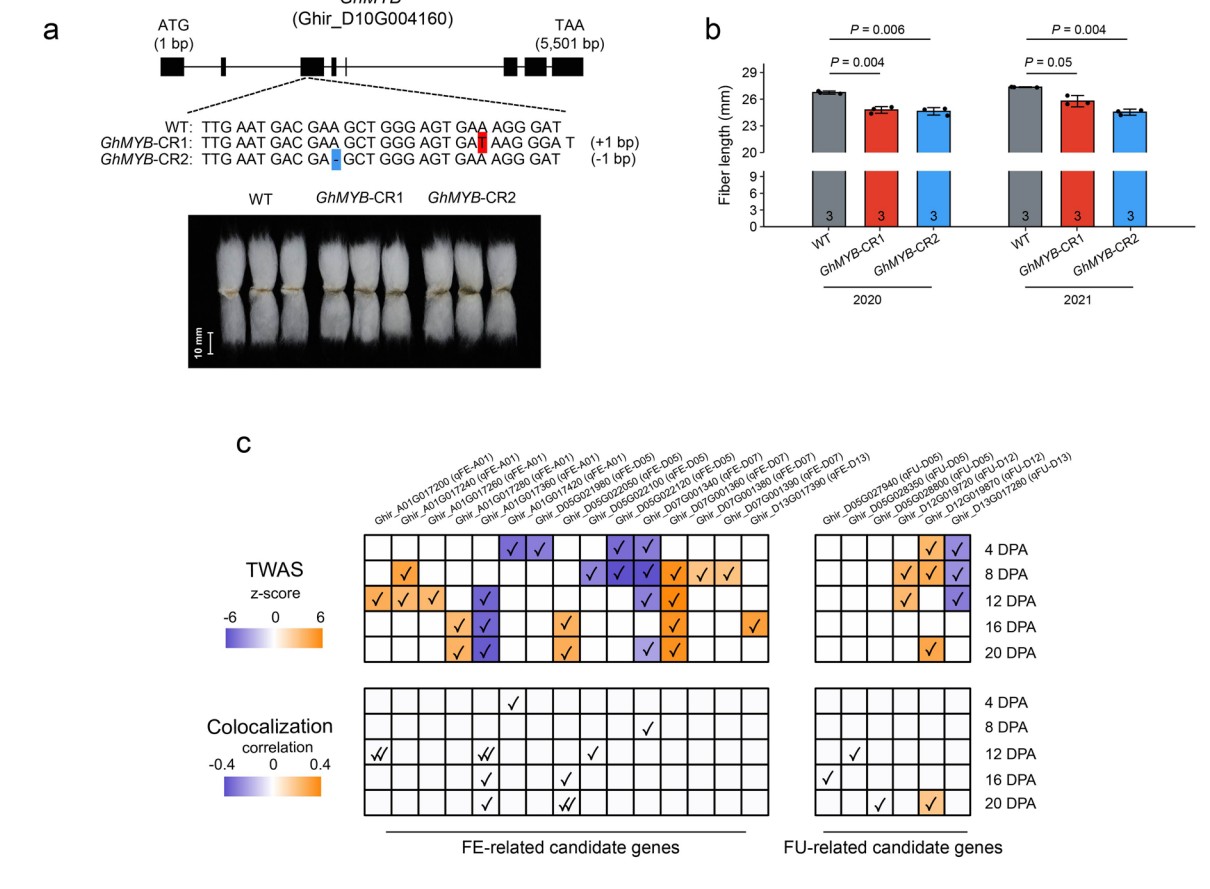

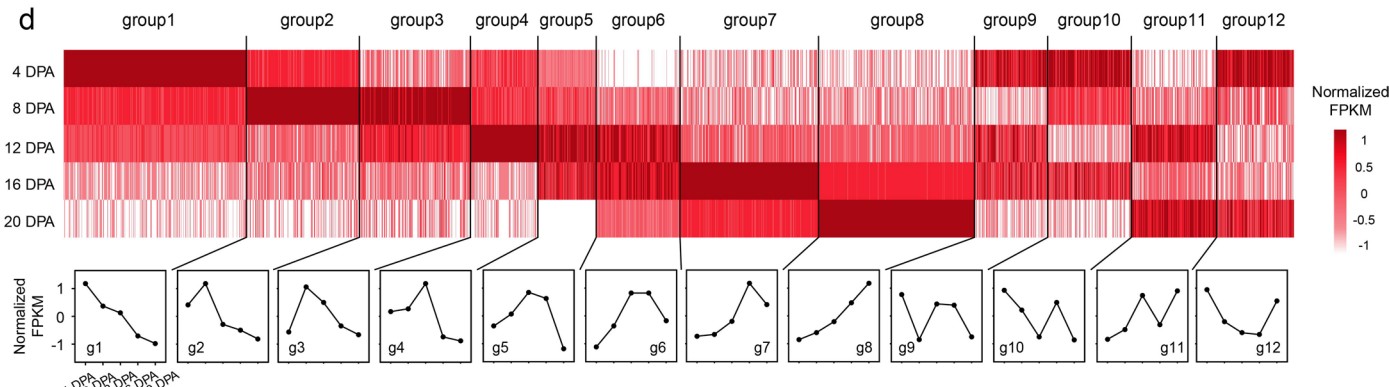

**Extended Data Fig. 3 | Candidate genes identified by TWAS and colocalization analysis. a**, Exon-intron structure and mutant sites of Ghir_D10G004160. *GhMYB*-CR1 and *GhMYB*-CR2 represent two different mutant types. The red shadow represents insertions; blue shadow represents deletions. WT, wild type. **b**, Comparison of fiber length in two years (2020 and 2021) between WT and mutants (*GhMYB*-CR1 and *GhMYB*-CR2) (n = 3 in each year; two-sided Student's *t*-test; error bar, mean ± SD). **c**, Phenotypic effects (TWAS z-score or correlation between expression and phenotype) of FE/FU-related candidate genes. Strategies are shown at the left, with illustrative color ranges. Positive z-score or correlation are shown in orange and negative values in purple. Significant betas are marked with 'check mark.' **d**, Expression patterns of 1,258 candidate genes in 340 accessions were divided into 12 groups. Heatmap showing normalized FPKM in each accession at each timepoint. Line charts showing mean values of accessions at each timepoint.

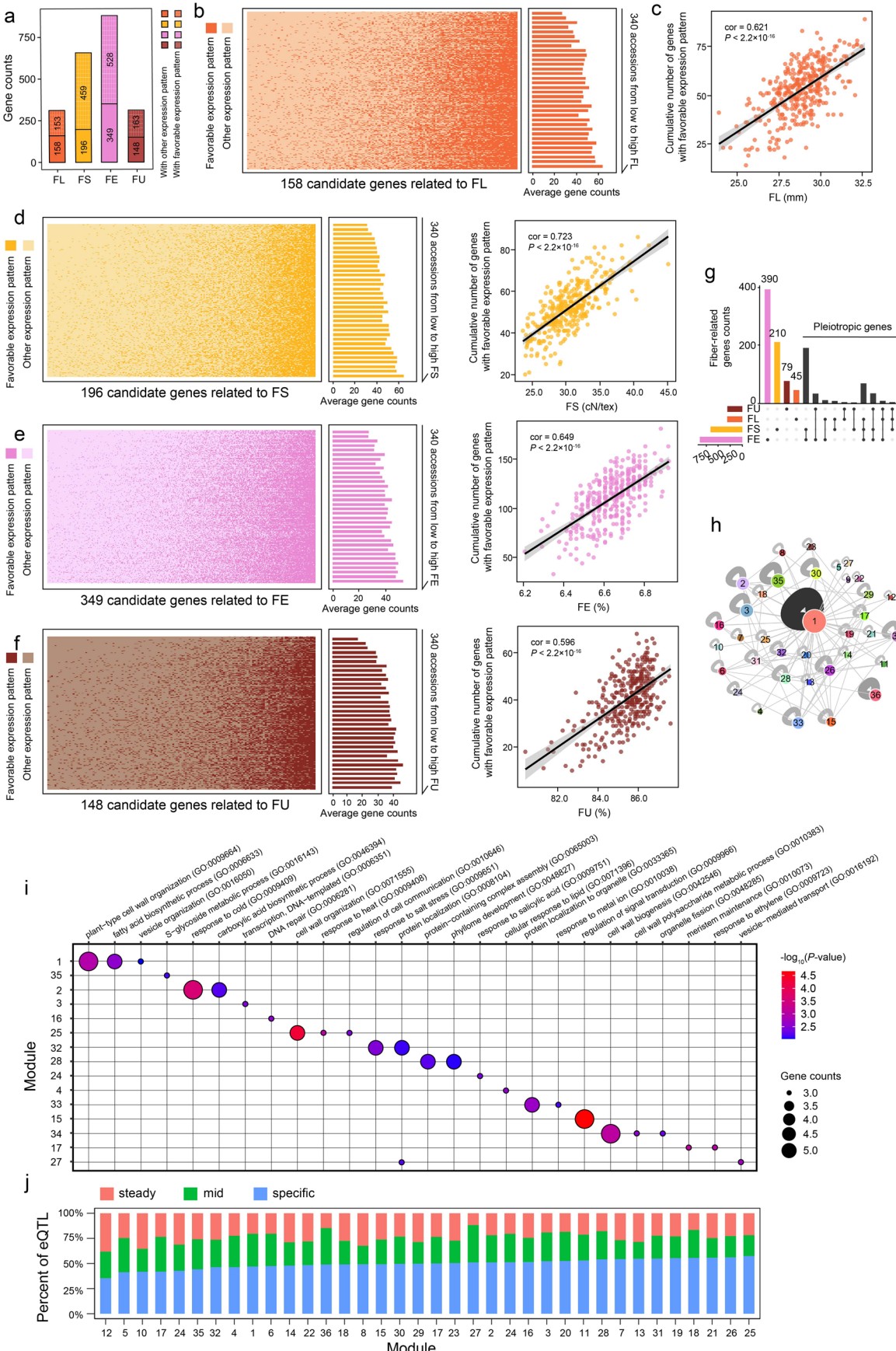

**Extended Data Fig. 4 | See next page for caption.**

**Extended Data Fig. 4 | Correlations of expression pattern with fiber quality and characterization of eQTL hotspots and genetic modules. a**, The counts of candidate genes with/without favorable expression patterns for four fiber quality-related traits. **b**, Heatmap showing FL-related genes with (lighter orange) or without (dark orange) favorable expression patterns in 340 cotton accessions. The 340 accessions are sorted by fiber length. Bar plot indicates mean counts of adjacent accessions. **c**, Correlation (cor) of fiber length (mm) and the cumulative number of genes with favorable expression patterns. Two-sided Student's *t*-test. The gray band represent 95% confidence interval for the fitted regression. **d**–**f**, Heatmap showing FS/FE/FU-related genes with (lighter color) or without (dark color) favorable expression pattern in 340 accessions. Cotton accessions are sorted by values of FS/FE/FU. The right panel shows correlation between values of FS/FE/FU and cumulative number of genes with favorable expression pattern. Two-sided Student's *t*-test. The gray band represent 95% confidence interval for the fitted regression. **g**, Venn diagram showing the number of genes identified for four fiber quality traits. **h**, Graph diagram showing eQTL-eGene connections within and between modules. Each module is represented as a circle with size proportional to the number of nodes. **i**, Gene ontology enrichment (biological process) of candidate genes from 15 modules (one-sided Fisher's exact test). **j**, Stack bar plot of eQTL distribution in 36 modules.

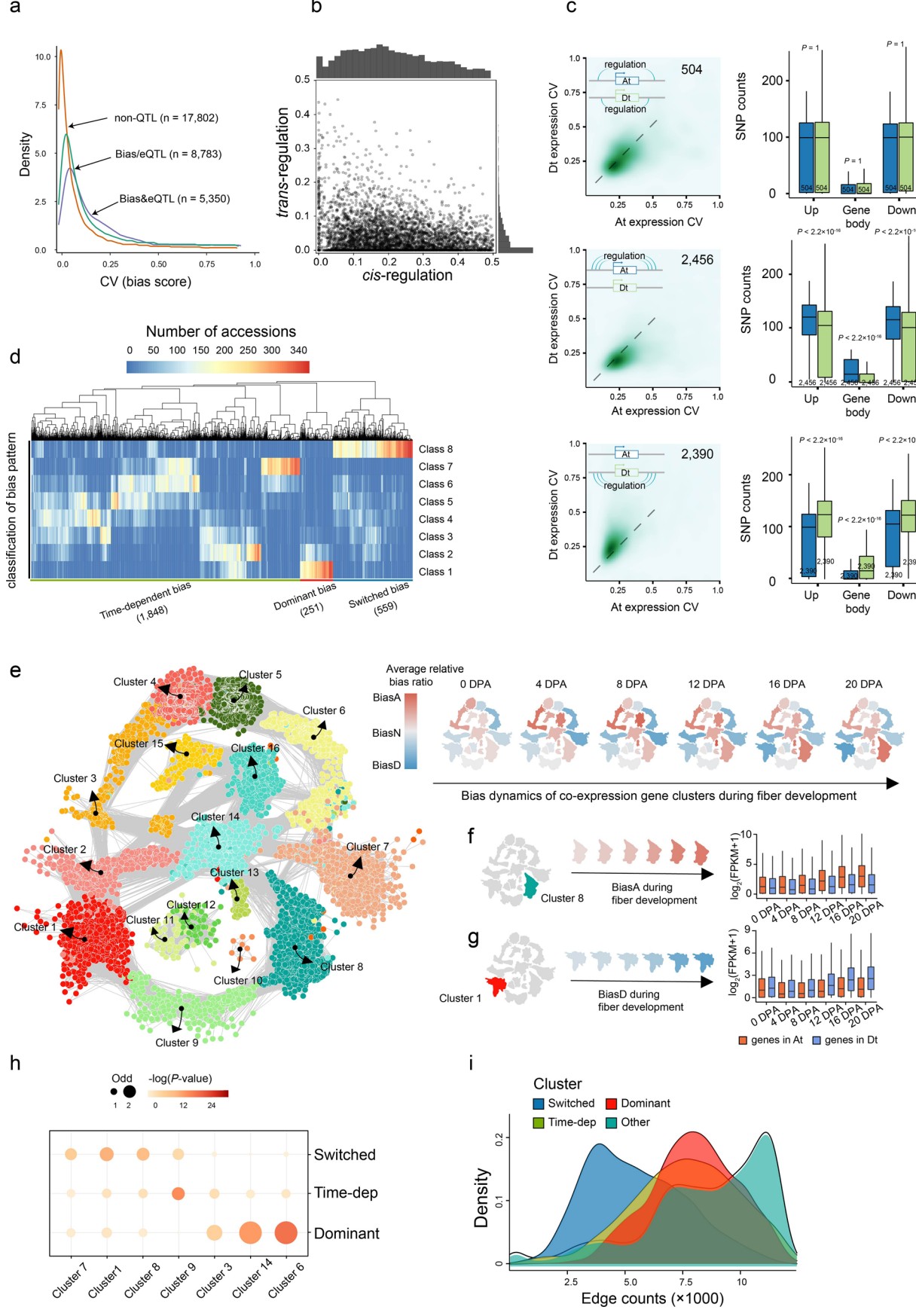

**Extended Data Fig. 5 | See next page for caption.**

**Extended Data Fig. 5 | Characterization of homoeologous expression bias and its dynamics across stages. a**, Comparison of the coefficient of variation for bias score in different expression bias patterns. 'non-QTL' indicates gene pairs without significant eQTL and bias-eQTL. 'Bias/eQTL' shows gene pairs with one eQTL or bias-eQTL. 'Bias&eQTL' indicates gene pairs with both eQTL and bias-eQTL. **b**, Genetic variance partitioning for a gene pair using intra- and inter-subgenomic SNPs (within 1 Mb up- or downstream of TSS). **c**, Density map showing the coefficient of variation (CV) of gene expression for gene pairs with different genetic regulatory patterns. The boxplot shows the count of SNPs in gene body and flanking regions (1 kb up- and downstream) (two-sided Wilcoxon rank sum test; center line, median; box limits, first and third quartiles; whisker, 1.5×interquartile range). **d**, Distribution of the number of accessions belonging to each dynamic bias pattern. Rows contain 8 categories of dynamic bias patterns (Methods) and columns are gene pairs with bias-eQTL. **e**, The dynamics

of homoeologous gene expression patterns was shown in a co-expression network. Nodes were colored for 16 co-expression clusters (left panel). The co-expression network was colored for expression bias (colored by average relative bias ratio; right panel). **f**, Dynamic expression bias and expression level (FPKM) of co-expression cluster 8 (n = 915 genes in different subgenomes at 6 timepoints; center line, median; box limits, first and third quartiles; whisker, 1.5× interquartile range). **g**, Dynamic expression bias and expression level (FPKM) of co-expression cluster 1 (n = 646 genes in different subgenomes at 6 timepoints; center line, median; box limits, first and third quartiles; whisker, 1.5× interquartile range). **h**, Enrichment of gene pairs with three bias patterns in co-expression clusters. The odd ratio and *P*-values for enrichment of clusters were calculated based on a one-sided Fisher's exact test. **i**, Distribution of edge numbers in co-expression clusters that were enriched by switched, time-dependent, and dominant gene pairs.

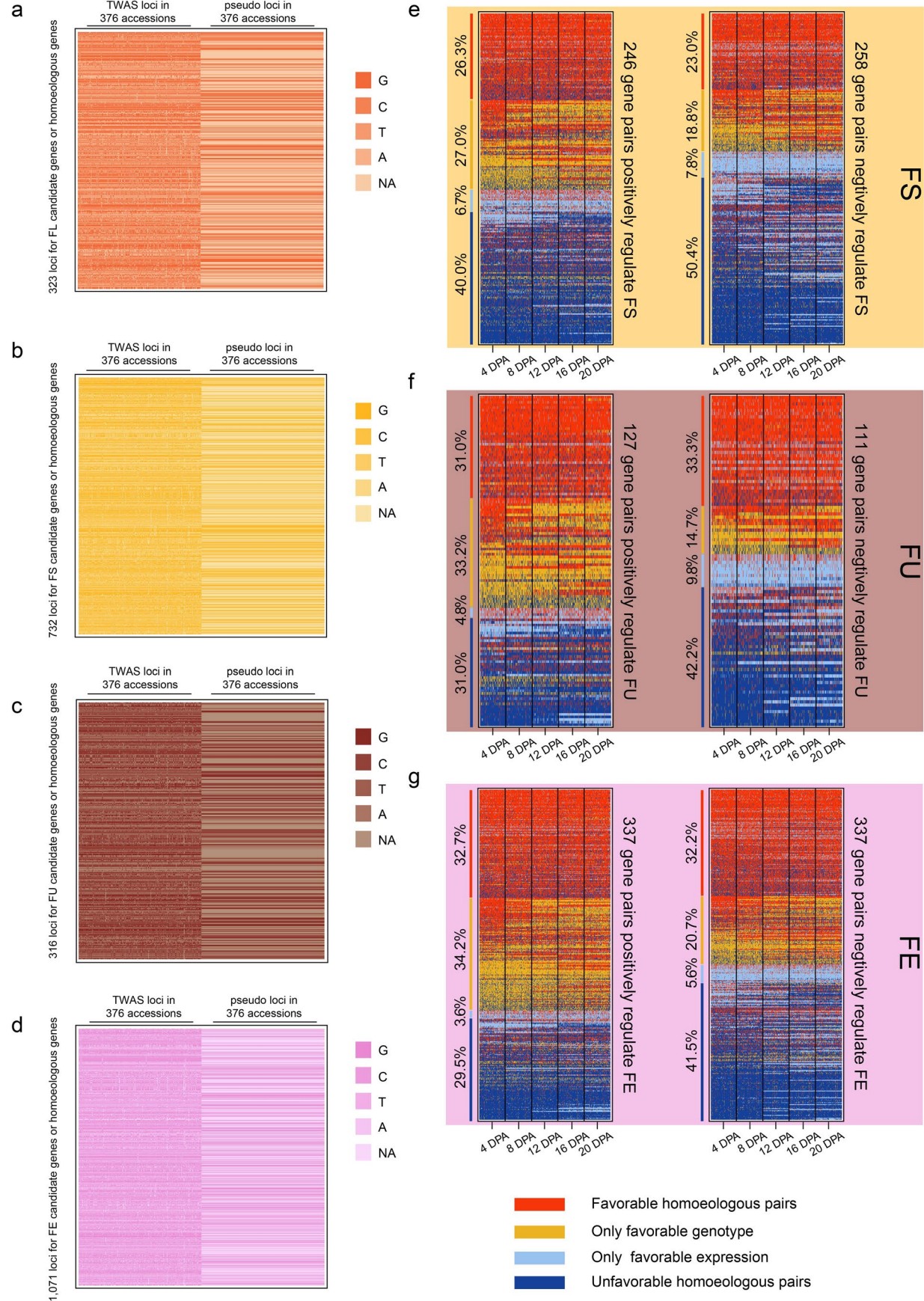

**Extended Data Fig. 6 | See next page for caption.**

**Extended Data Fig. 6 | Characterization of subgenomic coordination on FS, FE, and FU. a–d**, FL/FS/FU/FE-associated SNPs and genotypes of pseudo-regulatory sites in the other subgenome. **e**, 246 homoeologous gene pairs and 258 pairs with positive and negative regulation on fiber strength (FS), respectively. Heatmap shows the classification of 246 positive and 258 negative gene pairs in 340 accessions at five timepoints. **f**, 127 homoeologous gene pairs and 111 pairs with positive and negative regulation on fiber uniformity (FU), respectively. **g**, 337 positive homoeologous gene pairs and 337 pairs with positive and negative regulation on fiber elongation rate (FE), respectively.

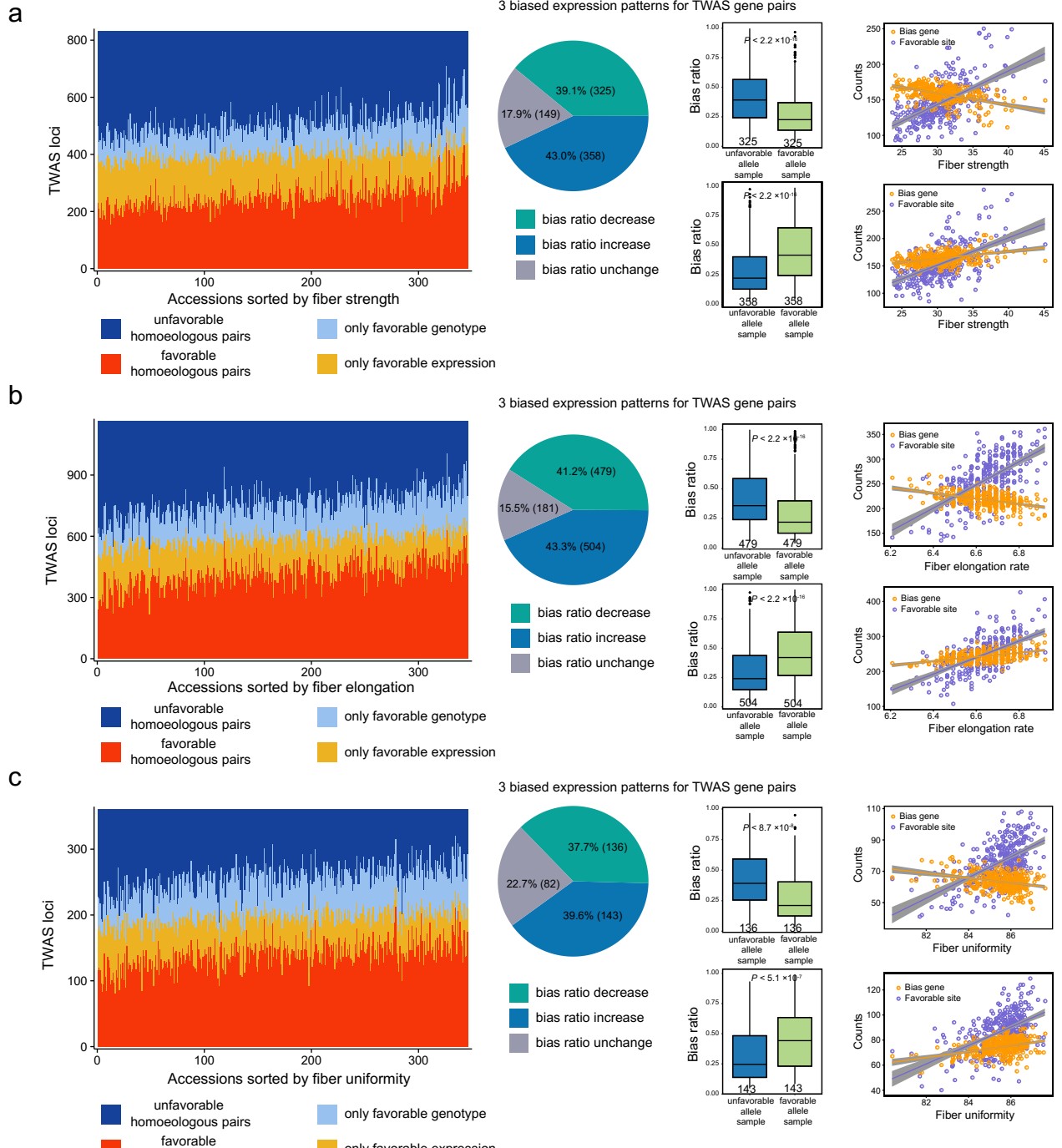

**Extended Data Fig. 7 | Relationships between homoeologous expression and three fiber quality-related traits. a**, Proportional distribution of four types of homoeologous genes in accessions that were sorted by fiber strength. The pie chart shows the ratio of gene pairs with different patterns of bias score change. The boxplots indicate gene pairs that show a significant difference in bias score (two-sided Wilcoxon rank sum test; center line, median; box limits, first and third quartiles; whisker, 1.5× interquartile range). The right panel shows correlation between the number of gene pairs with biased expression, the number of favorable loci, and fiber strength, the gray band represent 95% confidence interval for the fitted regression. **b**, Proportional distribution of four types of homoeologous genes in accessions that were sorted by FE. **c**, Proportional distribution of four types of homoeologous genes in accessions that were sorted by FU.

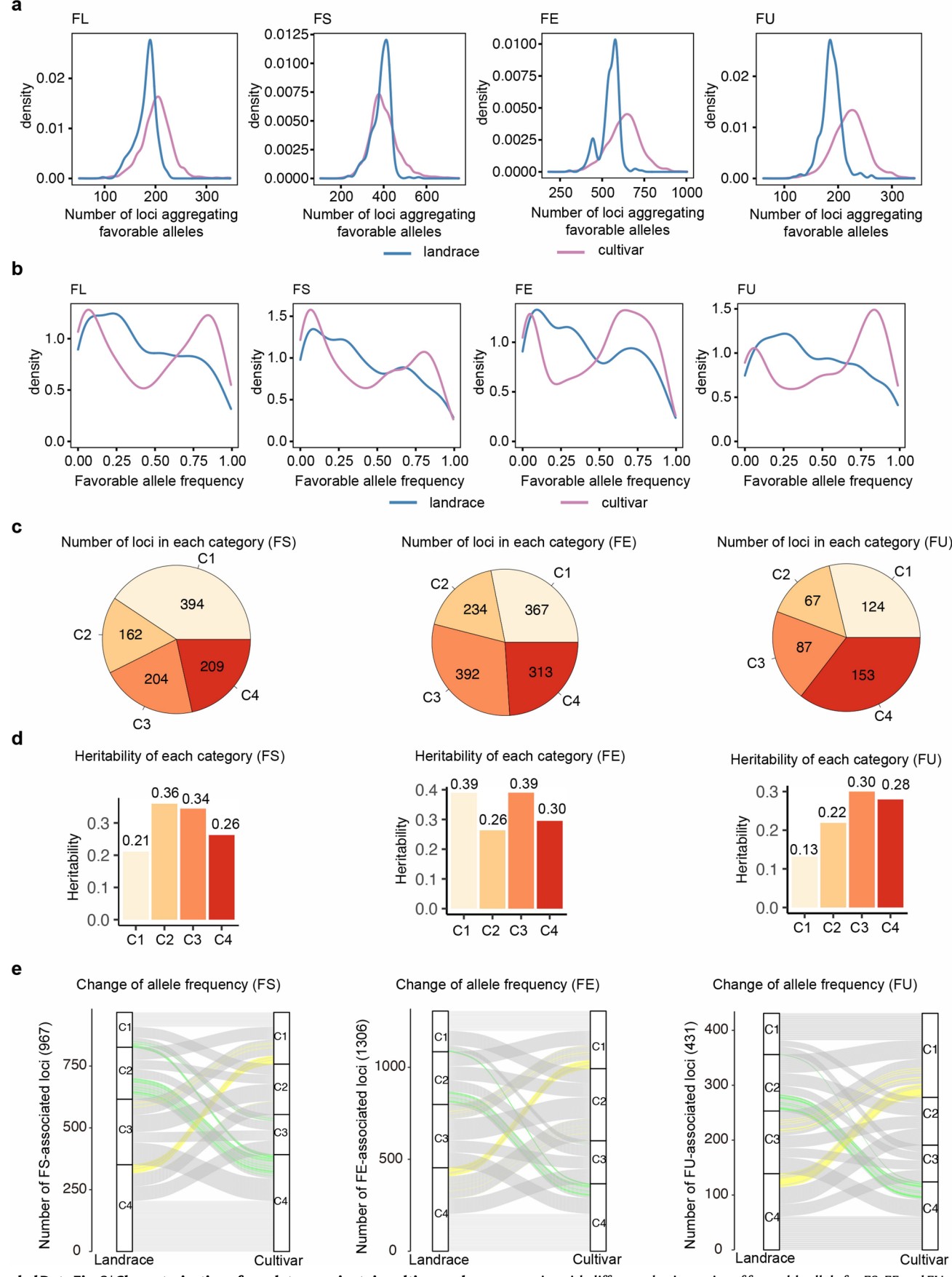

**Extended Data Fig. 8 | Characterization of regulatory variants in cultivar and landrace populations. a**, Distribution of the number of loci which aggregated favorable alleles in landrace and cultivar populations. **b**, Distribution of favorable allele frequency in landrace and cultivar populations. **c**, Number of variants in 4 categories with different sharing ratios of favorable allele for FS, FE and FU. **d**, Heritability of the variants in each of the 4 categories for FS, FE and FU. **e**, Comparison of the 4 categories between landrace and cultivar populations. The frequency differences above 0.6 were highlighted in yellow and green.

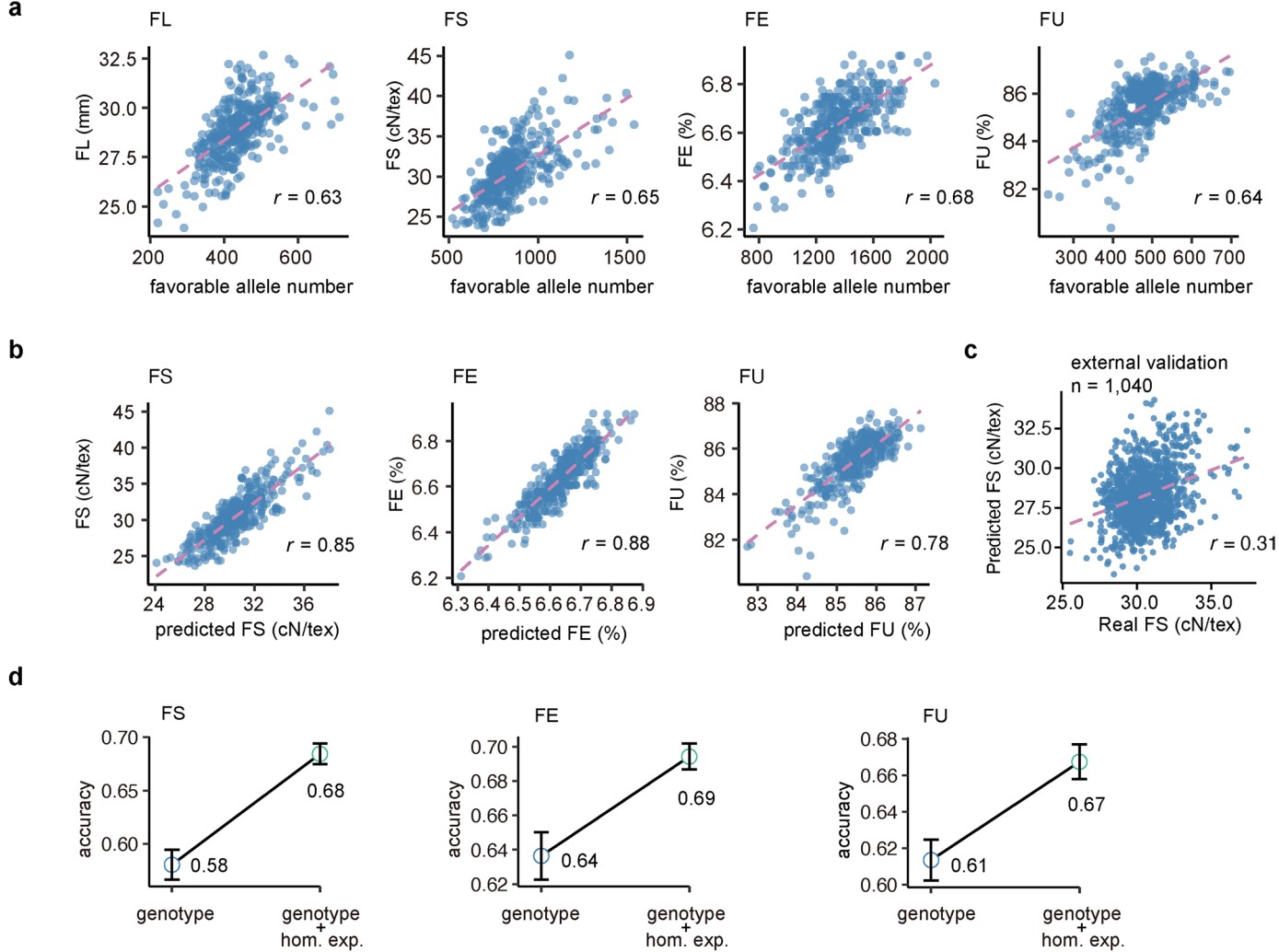

**Extended Data Fig. 9 | Predictive model and accuracy assessment for fiber-quality traits. a**, Correlation between the favorable allele number and fiber-quality traits. **b**, Correlation between the observed and predicted values for FL, FE and FU, respectively. Predictions were performed using ridge regression based on the inferred regulatory variants. **c**, Model predictability tested by applying the trained model in an external dataset (with 1,040 accessions) for FS. **d**, Fivefold cross-validated prediction accuracy using ridge regression model based on two predictive sets for 3 fiber quality-related traits (FS, FE and FU). One only contains genotype of trait-associated loci and the other contains both genotype and the expression of corresponding homoeologous gene copy. 100 replications were run. Circle, mean; error bar, mean ± SE.

|---|---|

# Reporting Summary

## Statistics

For all statistical analyses, confirm that the following items are present in the figure legend, table legend, main text, or Methods section.

| n/a | Confirmed | |
|---|---|---|
| ☐ | ☒ | The exact sample size (*n*) for each experimental group/condition, given as a discrete number and unit of measurement |
| ☐ | ☒ | A statement on whether measurements were taken from distinct samples or whether the same sample was measured repeatedly |
| ☐ | ☒ | The statistical test(s) used AND whether they are one- or two-sided *Only common tests should be described solely by name; describe more complex techniques in the Methods section.* |
| ☐ | ☒ | A description of all covariates tested |
| ☐ | ☒ | A description of any assumptions or corrections, such as tests of normality and adjustment for multiple comparisons |
| ☐ | ☒ | A full description of the statistical parameters including central tendency (e.g. means) or other basic estimates (e.g. regression coefficient) AND variation (e.g. standard deviation) or associated estimates of uncertainty (e.g. confidence intervals) |
| ☐ | ☒ | For null hypothesis testing, the test statistic (e.g. *F*, *t*, *r*) with confidence intervals, effect sizes, degrees of freedom and *P* value noted *Give P values as exact values whenever suitable.* |
| ☐ | ☒ | For Bayesian analysis, information on the choice of priors and Markov chain Monte Carlo settings |
| ☐ | ☒ | For hierarchical and complex designs, identification of the appropriate level for tests and full reporting of outcomes |
| ☐ | ☒ | Estimates of effect sizes (e.g. Cohen's *d*, Pearson's *r*), indicating how they were calculated |

*Our web collection on statistics for biologists contains articles on many of the points above.*

## Software and code

Policy information about availability of computer code

| Data collection | MGISEQ-T7 |
|---|---|
| Data analysis | All open source softwares have been described in Methods. Fastp (v0.23.0), Sentieon (201808.07), SAMtools (v1.9), Trimmomatic (v0.36), HISAT2 (v2.1.0), StringTie (v2.1.4), PEER (v1.3), FastQTL (v7), tensorQTL (v1.0.5), FAST-LMM (v0.2.32), mashr (v0.2.45), MetaSoft (v2.0.1), GCTA-LDF(v1.94.0), LDAK (v5.2), FUSION, pyliftover (v0.4), LASTZ (v1.04.03), STRUCTURE (v2.3.4), PLINK (v1.9), coloc (v5.1.0.1), SMR (v1.03), HOT_SCAN, WGCNA (v1.70.3), Gephi (v0.9.5), FastTree (v2.1.11), ggtree (v2.4.2), glmnet (v4.1.3), DESeq2 (v1.24.0), ANNOVAR, beagle (v4.1), pysam (v0.16.0.1) |

For manuscripts utilizing custom algorithms or software that are central to the research but not yet described in published literature, software must be made available to editors and reviewers. We strongly encourage code deposition in a community repository (e.g. GitHub). See the Nature Portfolio guidelines for submitting code & software for further information.

## Data

Policy information about availability of data

All manuscripts must include a data availability statement. This statement should provide the following information, where applicable:
- Accession codes, unique identifiers, or web links for publicly available datasets
- A description of any restrictions on data availability
- For clinical datasets or third party data, please ensure that the statement adheres to our policy

> All raw sequencing data generated in this paper have been deposited into the National Center for Biotechnology Information database (BioProject ID: PRJNA917453 for DNA resequencing data and PRJNA891378 for RNA-Seq data).

## Human research participants

Policy information about studies involving human research participants and Sex and Gender in Research.

| | |
|---|---|
| Reporting on sex and gender | NA |
| Population characteristics | NA |
| Recruitment | NA |
| Ethics oversight | NA |

Note that full information on the approval of the study protocol must also be provided in the manuscript.

# Field-specific reporting

Please select the one below that is the best fit for your research. If you are not sure, read the appropriate sections before making your selection.

☒ Life sciences    ☐ Behavioural & social sciences    ☐ Ecological, evolutionary & environmental sciences

For a reference copy of the document with all sections, see nature.com/documents/nr-reporting-summary-flat.pdf

# Life sciences study design

All studies must disclose on these points even when the disclosure is negative.

| | |
|---|---|
| Sample size | A total of 376 G. hirsutum accessions, including 2,215 ovule or fiber samples at 0 DPA, 4 DPA, 8 DPA, 12 DPA, 16 DPA and 20 DPA were collected. In this study, we generated DNA re-sequencing data for 373 accessions and RNA sequencing data for 2,215 samples. The accessions used for population genetic analysis represent core germplasm of this species, and the sample sizes are sufficient for genetic analysis. |
| Data exclusions | NA |
| Replication | All attempts at replication for RNA-Seq experiment of GhMYB mutants were successful. All the experiments for replication were performed independently. |
| Randomization | This is not relevant to our study. |
| Blinding | This is not relevant to our study. |

# Behavioural & social sciences study design

All studies must disclose on these points even when the disclosure is negative.

| | |
|---|---|
| Study description | *Briefly describe the study type including whether data are quantitative, qualitative, or mixed-methods (e.g. qualitative cross-sectional, quantitative experimental, mixed-methods case study).* |
| Research sample | *State the research sample (e.g. Harvard university undergraduates, villagers in rural India) and provide relevant demographic information (e.g. age, sex) and indicate whether the sample is representative. Provide a rationale for the study sample chosen. For studies involving existing datasets, please describe the dataset and source.* |
| Sampling strategy | *Describe the sampling procedure (e.g. random, snowball, stratified, convenience). Describe the statistical methods that were used to* |

| Sampling strategy | predetermine sample size OR if no sample-size calculation was performed, describe how sample sizes were chosen and provide a rationale for why these sample sizes are sufficient. For qualitative data, please indicate whether data saturation was considered, and what criteria were used to decide that no further sampling was needed. |
| --- | --- |
| Data collection | Provide details about the data collection procedure, including the instruments or devices used to record the data (e.g. pen and paper, computer, eye tracker, video or audio equipment) whether anyone was present besides the participant(s) and the researcher, and whether the researcher was blind to experimental condition and/or the study hypothesis during data collection. |
| Timing | Indicate the start and stop dates of data collection. If there is a gap between collection periods, state the dates for each sample cohort. |
| Data exclusions | If no data were excluded from the analyses, state so OR if data were excluded, provide the exact number of exclusions and the rationale behind them, indicating whether exclusion criteria were pre-established. |
| Non-participation | State how many participants dropped out/declined participation and the reason(s) given OR provide response rate OR state that no participants dropped out/declined participation. |
| Randomization | If participants were not allocated into experimental groups, state so OR describe how participants were allocated to groups, and if allocation was not random, describe how covariates were controlled. |

# Ecological, evolutionary & environmental sciences study design

All studies must disclose on these points even when the disclosure is negative.

| Study description | Briefly describe the study. For quantitative data include treatment factors and interactions, design structure (e.g. factorial, nested, hierarchical), nature and number of experimental units and replicates. |
| --- | --- |
| Research sample | Describe the research sample (e.g. a group of tagged Passer domesticus, all Stenocereus thurberi within Organ Pipe Cactus National Monument), and provide a rationale for the sample choice. When relevant, describe the organism taxa, source, sex, age range and any manipulations. State what population the sample is meant to represent when applicable. For studies involving existing datasets, describe the data and its source. |
| Sampling strategy | Note the sampling procedure. Describe the statistical methods that were used to predetermine sample size OR if no sample-size calculation was performed, describe how sample sizes were chosen and provide a rationale for why these sample sizes are sufficient. |
| Data collection | Describe the data collection procedure, including who recorded the data and how. |
| Timing and spatial scale | Indicate the start and stop dates of data collection, noting the frequency and periodicity of sampling and providing a rationale for these choices. If there is a gap between collection periods, state the dates for each sample cohort. Specify the spatial scale from which the data are taken |
| Data exclusions | If no data were excluded from the analyses, state so OR if data were excluded, describe the exclusions and the rationale behind them, indicating whether exclusion criteria were pre-established. |
| Reproducibility | Describe the measures taken to verify the reproducibility of experimental findings. For each experiment, note whether any attempts to repeat the experiment failed OR state that all attempts to repeat the experiment were successful. |
| Randomization | Describe how samples/organisms/participants were allocated into groups. If allocation was not random, describe how covariates were controlled. If this is not relevant to your study, explain why. |
| Blinding | Describe the extent of blinding used during data acquisition and analysis. If blinding was not possible, describe why OR explain why blinding was not relevant to your study. |

Did the study involve field work? ☐ Yes ☐ No

# Field work, collection and transport

| Field conditions | Describe the study conditions for field work, providing relevant parameters (e.g. temperature, rainfall). |
| --- | --- |
| Location | State the location of the sampling or experiment, providing relevant parameters (e.g. latitude and longitude, elevation, water depth). |
| Access & import/export | Describe the efforts you have made to access habitats and to collect and import/export your samples in a responsible manner and in compliance with local, national and international laws, noting any permits that were obtained (give the name of the issuing authority, the date of issue, and any identifying information). |
| Disturbance | Describe any disturbance caused by the study and how it was minimized. |

# Reporting for specific materials, systems and methods

We require information from authors about some types of materials, experimental systems and methods used in many studies. Here, indicate whether each material, system or method listed is relevant to your study. If you are not sure if a list item applies to your research, read the appropriate section before selecting a response.

## Materials & experimental systems

| n/a | Involved in the study |
|-----|------------------------|
| ☒ | ☐ Antibodies |
| ☒ | ☐ Eukaryotic cell lines |
| ☒ | ☐ Palaeontology and archaeology |
| ☒ | ☐ Animals and other organisms |
| ☒ | ☐ Clinical data |
| ☒ | ☐ Dual use research of concern |

## Methods

| n/a | Involved in the study |
|-----|------------------------|
| ☒ | ☐ ChIP-seq |
| ☒ | ☐ Flow cytometry |
| ☒ | ☐ MRI-based neuroimaging |

## Antibodies

| | |
|---|---|
| Antibodies used | *Describe all antibodies used in the study; as applicable, provide supplier name, catalog number, clone name, and lot number.* |
| Validation | *Describe the validation of each primary antibody for the species and application, noting any validation statements on the manufacturer's website, relevant citations, antibody profiles in online databases, or data provided in the manuscript.* |

## Eukaryotic cell lines

Policy information about cell lines and Sex and Gender in Research

| | |
|---|---|
| Cell line source(s) | *State the source of each cell line used and the sex of all primary cell lines and cells derived from human participants or vertebrate models.* |
| Authentication | *Describe the authentication procedures for each cell line used OR declare that none of the cell lines used were authenticated.* |
| Mycoplasma contamination | *Confirm that all cell lines tested negative for mycoplasma contamination OR describe the results of the testing for mycoplasma contamination OR declare that the cell lines were not tested for mycoplasma contamination.* |
| Commonly misidentified lines (See ICLAC register) | *Name any commonly misidentified cell lines used in the study and provide a rationale for their use.* |

## Palaeontology and Archaeology

| | |
|---|---|
| Specimen provenance | *Provide provenance information for specimens and describe permits that were obtained for the work (including the name of the issuing authority, the date of issue, and any identifying information). Permits should encompass collection and, where applicable, export.* |
| Specimen deposition | *Indicate where the specimens have been deposited to permit free access by other researchers.* |
| Dating methods | *If new dates are provided, describe how they were obtained (e.g. collection, storage, sample pretreatment and measurement), where they were obtained (i.e. lab name), the calibration program and the protocol for quality assurance OR state that no new dates are provided.* |

☐ Tick this box to confirm that the raw and calibrated dates are available in the paper or in Supplementary Information.

| | |
|---|---|
| Ethics oversight | *Identify the organization(s) that approved or provided guidance on the study protocol, OR state that no ethical approval or guidance was required and explain why not.* |

Note that full information on the approval of the study protocol must also be provided in the manuscript.

## Animals and other research organisms

Policy information about studies involving animals; ARRIVE guidelines recommended for reporting animal research, and Sex and Gender in Research

| | |
|---|---|
| Laboratory animals | *For laboratory animals, report species, strain and age OR state that the study did not involve laboratory animals.* |

| Wild animals | *Provide details on animals observed in or captured in the field; report species and age where possible. Describe how animals were caught and transported and what happened to captive animals after the study (if killed, explain why and describe method; if released, say where and when) OR state that the study did not involve wild animals.* |
| Reporting on sex | *Indicate if findings apply to only one sex; describe whether sex was considered in study design, methods used for assigning sex. Provide data disaggregated for sex where this information has been collected in the source data as appropriate; provide overall numbers in this Reporting Summary. Please state if this information has not been collected. Report sex-based analyses where performed, justify reasons for lack of sex-based analysis.* |
| Field-collected samples | *For laboratory work with field-collected samples, describe all relevant parameters such as housing, maintenance, temperature, photoperiod and end-of-experiment protocol OR state that the study did not involve samples collected from the field.* |
| Ethics oversight | *Identify the organization(s) that approved or provided guidance on the study protocol, OR state that no ethical approval or guidance was required and explain why not.* |

Note that full information on the approval of the study protocol must also be provided in the manuscript.

# Clinical data

Policy information about clinical studies
All manuscripts should comply with the ICMJE guidelines for publication of clinical research and a completed CONSORT checklist must be included with all submissions.

| Clinical trial registration | *Provide the trial registration number from ClinicalTrials.gov or an equivalent agency.* |
| Study protocol | *Note where the full trial protocol can be accessed OR if not available, explain why.* |
| Data collection | *Describe the settings and locales of data collection, noting the time periods of recruitment and data collection.* |
| Outcomes | *Describe how you pre-defined primary and secondary outcome measures and how you assessed these measures.* |

# Dual use research of concern

Policy information about dual use research of concern

## Hazards

Could the accidental, deliberate or reckless misuse of agents or technologies generated in the work, or the application of information presented in the manuscript, pose a threat to:

No | Yes
☐ ☐ Public health
☐ ☐ National security
☐ ☐ Crops and/or livestock
☐ ☐ Ecosystems
☐ ☐ Any other significant area

## Experiments of concern

Does the work involve any of these experiments of concern:

No | Yes
☐ ☐ Demonstrate how to render a vaccine ineffective
☐ ☐ Confer resistance to therapeutically useful antibiotics or antiviral agents
☐ ☐ Enhance the virulence of a pathogen or render a nonpathogen virulent
☐ ☐ Increase transmissibility of a pathogen
☐ ☐ Alter the host range of a pathogen
☐ ☐ Enable evasion of diagnostic/detection modalities
☐ ☐ Enable the weaponization of a biological agent or toxin
☐ ☐ Any other potentially harmful combination of experiments and agents

# ChIP-seq

## Data deposition

☐ Confirm that both raw and final processed data have been deposited in a public database such as GEO.

☐ Confirm that you have deposited or provided access to graph files (e.g. BED files) for the called peaks.

| | |
|---|---|
| **Data access links**<br>*May remain private before publication.* | *For "Initial submission" or "Revised version" documents, provide reviewer access links. For your "Final submission" document, provide a link to the deposited data.* |
| **Files in database submission** | *Provide a list of all files available in the database submission.* |
| **Genome browser session**<br>(e.g. UCSC) | *Provide a link to an anonymized genome browser session for "Initial submission" and "Revised version" documents only, to enable peer review. Write "no longer applicable" for "Final submission" documents.* |

## Methodology

| | |
|---|---|
| **Replicates** | *Describe the experimental replicates, specifying number, type and replicate agreement.* |
| **Sequencing depth** | *Describe the sequencing depth for each experiment, providing the total number of reads, uniquely mapped reads, length of reads and whether they were paired- or single-end.* |
| **Antibodies** | *Describe the antibodies used for the ChIP-seq experiments; as applicable, provide supplier name, catalog number, clone name, and lot number.* |
| **Peak calling parameters** | *Specify the command line program and parameters used for read mapping and peak calling, including the ChIP, control and index files used.* |
| **Data quality** | *Describe the methods used to ensure data quality in full detail, including how many peaks are at FDR 5% and above 5-fold enrichment.* |
| **Software** | *Describe the software used to collect and analyze the ChIP-seq data. For custom code that has been deposited into a community repository, provide accession details.* |

# Flow Cytometry

## Plots

Confirm that:

☐ The axis labels state the marker and fluorochrome used (e.g. CD4-FITC).

☐ The axis scales are clearly visible. Include numbers along axes only for bottom left plot of group (a 'group' is an analysis of identical markers).

☐ All plots are contour plots with outliers or pseudocolor plots.

☐ A numerical value for number of cells or percentage (with statistics) is provided.

## Methodology

| | |
|---|---|
| **Sample preparation** | *Describe the sample preparation, detailing the biological source of the cells and any tissue processing steps used.* |
| **Instrument** | *Identify the instrument used for data collection, specifying make and model number.* |
| **Software** | *Describe the software used to collect and analyze the flow cytometry data. For custom code that has been deposited into a community repository, provide accession details.* |
| **Cell population abundance** | *Describe the abundance of the relevant cell populations within post-sort fractions, providing details on the purity of the samples and how it was determined.* |
| **Gating strategy** | *Describe the gating strategy used for all relevant experiments, specifying the preliminary FSC/SSC gates of the starting cell population, indicating where boundaries between "positive" and "negative" staining cell populations are defined.* |

☐ Tick this box to confirm that a figure exemplifying the gating strategy is provided in the Supplementary Information.

# Magnetic resonance imaging

## Experimental design

| | |
|---|---|
| **Design type** | *Indicate task or resting state; event-related or block design.* |

| Design specifications | *Specify the number of blocks, trials or experimental units per session and/or subject, and specify the length of each trial or block (if trials are blocked) and interval between trials.* |
|---|---|
| Behavioral performance measures | *State number and/or type of variables recorded (e.g. correct button press, response time) and what statistics were used to establish that the subjects were performing the task as expected (e.g. mean, range, and/or standard deviation across subjects).* |

## Acquisition

| Imaging type(s) | *Specify: functional, structural, diffusion, perfusion.* |
|---|---|
| Field strength | *Specify in Tesla* |
| Sequence & imaging parameters | *Specify the pulse sequence type (gradient echo, spin echo, etc.), imaging type (EPI, spiral, etc.), field of view, matrix size, slice thickness, orientation and TE/TR/flip angle.* |
| Area of acquisition | *State whether a whole brain scan was used OR define the area of acquisition, describing how the region was determined.* |

Diffusion MRI    ☐ Used    ☐ Not used

## Preprocessing

| Preprocessing software | *Provide detail on software version and revision number and on specific parameters (model/functions, brain extraction, segmentation, smoothing kernel size, etc.).* |
|---|---|
| Normalization | *If data were normalized/standardized, describe the approach(es): specify linear or non-linear and define image types used for transformation OR indicate that data were not normalized and explain rationale for lack of normalization.* |
| Normalization template | *Describe the template used for normalization/transformation, specifying subject space or group standardized space (e.g. original Talairach, MNI305, ICBM152) OR indicate that the data were not normalized.* |
| Noise and artifact removal | *Describe your procedure(s) for artifact and structured noise removal, specifying motion parameters, tissue signals and physiological signals (heart rate, respiration).* |
| Volume censoring | *Define your software and/or method and criteria for volume censoring, and state the extent of such censoring.* |

## Statistical modeling & inference

| Model type and settings | *Specify type (mass univariate, multivariate, RSA, predictive, etc.) and describe essential details of the model at the first and second levels (e.g. fixed, random or mixed effects; drift or auto-correlation).* |
|---|---|
| Effect(s) tested | *Define precise effect in terms of the task or stimulus conditions instead of psychological concepts and indicate whether ANOVA or factorial designs were used.* |

Specify type of analysis:    ☐ Whole brain    ☐ ROI-based    ☐ Both

| Statistic type for inference (See Eklund et al. 2016) | *Specify voxel-wise or cluster-wise and report all relevant parameters for cluster-wise methods.* |
|---|---|
| Correction | *Describe the type of correction and how it is obtained for multiple comparisons (e.g. FWE, FDR, permutation or Monte Carlo).* |

## Models & analysis

| n/a | Involved in the study |
|---|---|
| ☐ | ☐ Functional and/or effective connectivity |
| ☐ | ☐ Graph analysis |
| ☐ | ☐ Multivariate modeling or predictive analysis |

| Functional and/or effective connectivity | *Report the measures of dependence used and the model details (e.g. Pearson correlation, partial correlation, mutual information).* |
|---|---|
| Graph analysis | *Report the dependent variable and connectivity measure, specifying weighted graph or binarized graph, subject- or group-level, and the global and/or node summaries used (e.g. clustering coefficient, efficiency, etc.).* |
| Multivariate modeling and predictive analysis | *Specify independent variables, features extraction and dimension reduction, model, training and evaluation metrics.* |

