## [Peer Review File · Nature Genetics]

Peer Review Information

Manuscript Title: Regulatory controls of duplicated gene expression during fiber development in allotetraploid cotton

Corresponding author name(s): Professor Lili Tu, Professor Xianlong Zhang, Professor Maojun Wang

Editorial Notes:

Transferred manuscripts This document only contains reviewer comments, rebuttal and decision letters for versions considered at Nature Genetics.

Reviewer Comments & Decisions:

Decision Letter, initial version:

27th Jan 2023

Dear Professor Wang,

Your Article, "Genetic regulome of dynamic gene expression in allotetraploid cotton enables fiber improvement through subgenome coordination" has now been seen by 3 referees. You will see from their comments below that while they find your work of interest, some important points are raised. We are interested in the possibility of publishing your study in Nature Genetics, but would like to consider your response to these concerns in the form of a suitably revised manuscript before we make a final decision on publication.

To guide the scope of the revisions, the editors discuss the referee reports in detail within the team with a view to identifying key priorities that should be addressed in revision. In this case, we think all three referees have provided constructive reviews aimed at strengthening the analyses and improving the presentation, and we particularly ask that you address their comments as thoroughly as possible with appropriate revisions. We hope that you will find the prioritized set of referee points to be useful when revising your study. Please do not hesitate to get in touch if you would like to discuss these issues further.

We therefore invite you to revise your manuscript taking into account all reviewer and editor comments. Please highlight all changes in the manuscript text file. At this stage we will need you to upload a copy of the manuscript in MS Word .docx or similar editable format.

*2) If you have not done so already please begin to revise your manuscript so that it conforms to our Article format instructions, available [here](http://www.nature.com/ng/authors/article_types/index.html). Refer also to any guidelines provided in this letter.

[redacted]

We hope to receive your revised manuscript within 3 to 6 months. If you cannot send it within this time, please let us know.

We look forward to seeing the revised manuscript and thank you for the opportunity to review your

work.

Sincerely,
Wei

Wei Li, PhD
Senior Editor
Nature Genetics
New York, NY 10004, USA
www.nature.com/ng

Reviewers' Comments:

Reviewer #1:

Remarks to the Author:

In this study, to dissect the genetic regulome of dynamic gene expression in allotetraploid cotton enables fiber improvement through subgenome coordination, the authors first construct a genetic regulome of gene expression in 376 diverse *G. hirsutum* accessions using transcriptome from 6 fiber development timepoints, then they performed comprehensive genetic regulome analysis, including eQTL, TWAS, bias eQTL, and multiple combination analyses, which led them to uncover the genetic components that may optimize homoeologous gene expression for unlocking the potential for fiber improvement. I would say this work is one of the most comprehensive genetic regulome analysis, at least in plants, and provides valuable information and highlight the significance of subgenomic coordination in fiber improvement in allotetraploid cotton.

Here are some comments that may help to further improve the manuscript:

1. I believe there must be some artificial error or noise for the eQTLs. The fact "the authors found that stage-shared eQTLs showed larger effects than stage-specific eQTLs, and stage-shared eGenes had a higher proportion of cis-eQTLs than stage-specific eGenes" may be related to the artificial error or noise. The authors may consider to do some further analyses to check if there are some artificial errors. This may help to remove some artificial errors or noise eQTLs.

Another solution might be to focus on the leading eQTLs to perform the important analysis.

2. The authors found that "Most GWAS QTLs overlapped with TWAS peaks, indicating that our data have good repeatability across different methods". This part is very interesting. In fact, there are some specific QTLs for each method. For example, for the TWAS, there is a significant TWAS QTL in D03. These kinds of specific TWAS QTLs may need to pay more attention, which also closely match to the character of the datasets from this study.

In addition, I am not quite understand why the TWAS show such high overlap pattern with GWAS: the QTLs are overlapped; the TWAS QTLs also showed association peak. Does that mean, most of the genetic variations of this region result in gene expression changes, and also related to the phenotype? Or one genetic variation resulted in the expression change of multiple genes? This is very interesting to me. The authors may try more detailed analyses for this part.

3. For the "genetic regulatory network" part, it is very interesting and provides a lot of information. However, it sounds too complicated to me. If it can be simplified, it may give clearer clues. For example, for the 36 modules, most of them show tiny or even no contribution for the four traits. If these components were reduced, the major modules that dominantly regulate the fiber traits may be

left, and may give more clear information.

Minor comments:

1. Line 98: in the text, it is described that "Among these, 74.2% were expressed at all timepoints". However, in the Extended Data Fig. 1d, no data shows the genes expressed at all timepoints.
2. The authors found that "23,811 trans-eQTLs for 23,012 genes", and then "a higher proportion of trans-eGenes were detected with multiple eQTL signals". The two datasets are little confused. May need to further check or to revise the text a little bit to make it clear.
3. It is better to give some explanation of the abbreviations in the text. For example, for Fig 4b, it is better to term the bias and biasN in the text. It is hard to get the meaning of "BiasN" without careful consider.
4. For the example of BB2, it is found that, from group 2 to group 5, the expression pattern of BB2 was gradually delayed, and the average FL of the corresponding accessions gradually increased. I am not sure if this is related to the maturity/flowering time which may affect the development of fiber development. If so, it might be better to give a discussion.
5. Line 253-254: It was found that expression variance of genes in each subgenome was mainly explained by intra-subgenomic variants. Based on Extended Fig 5a, the At and Dt for At eGene are close although the statistic has some difference.
6. For the part of "Definition of biased expression pattern in fiber development", all cotton accessions were classified into 8 classes according to the biased expression patterns. There should be more complicated classes, such as 3 stage showed the one direction and the other 3 stages showed another direction. Are they belong to the 'bias direction change'? Whereas, the definition of it looks like is not the case: direction of expression bias is different in any two timepoints. In addition, what is the relationship between the 8 classes and the 3 groups (switched, time-dependent and dominant)?
7. As to the comprehensive analyses, the discussion is too simple. The authors may add more discussion, for example for the subgenome coordination, the tris/trans eQTL, and the candidate genes for the fiber development.

Reviewer #2:

Remarks to the Author:

The work described in this paper uses extensive genotyping and transcriptome data from different stages of cotton ovule and fibre development in many diverse cotton accessions to identify eQTL that are associated with variation in gene expression. They identified 18 QTL for different fibre traits, of which 9 were new.

The main strengths of the work are the large datasets that permit identification of new genetic variants that can be used in breeding of this key crop. The current study aims to be comprehensive, such as defining the roles of genetic variation in homoeologous genes that influence gene expression in developing fibres and their influence on traits. The Introduction and Methods are succinctly and clearly laid out, giving hope that the general reader will be able to follow the work. However, the Discussion does little to relate the current work to the field and is a mere brief recitation of the Results.

The major problem is the Results section, which is densely opaque and extremely difficult to follow and to interpret. I found the rationale for some approaches and analyses was not clearly explained,

and the results obtained were sometimes over-interpreted and at times left quite vague. This is not a problem of the written English, as there are several native speakers as authors- some with the specific role of editing the manuscript. I think the problem lies in lack of experience in reducing complex observations into clear and easily understood messages. To do this the authors need, for each set of Results, to first define the objective and why it's important, then explain the choice of methods, and explain how significant the results are. As currently written, the reader (at least me) has no clear idea about why the set of analyses were carried out and could not work out its significance.

It may be that some experts in the field can follow the work clearly, but for publication in a general journal, the Results and Discussion sections need to be comprehensively re-written with the objective of helping the reader.

Reviewer #3:

Remarks to the Author:

You et al. have generated an impressive collection of cotton genome data. I can summary their key results/findings as below:

- Re-sequence 376 cotton accessions. RNA-seq for 6 tissues for each accession.
- Compute eQTL separately for each tissue.
- Performed a fiber quality GWAS and found 18 QTL that overlaps with eQTL. Also did a TWAS and found 1255 genes associated with fiber quality.
- Combine eQTL derived from 5 fiber tissues to create a "regulatory network"
- They found gene expression differences of homologous genes in At and Dt subgenome.
- They also examined the expression pattern of some homologous gene pairs that are associated with fiber quality. Then they proposed that one can further improve fiber quality by targeting one of the "less favourably expressed" gene in the homologous gene pairs.

To me, this is an important genomic resource for cotton and did deserve a spot in a top journal. But there are lots of small problems in writing, analysis, missing statistics, and how to properly interpret the result. I list them below and hope they can correct those in the revised manuscript.

L101. If there is no replicate for the RNA-seq, how to call expression bias for the homologous gene pairs? And how to correct for multiple test? It appears to me that they only use FPKM fold change. At least one can use the raw count for a binominal test and then correct for a 5% FDR.

L103, a proper statistical test must be performed to call the At and Dt bias pairs in all tissue (time point), and also need multiple test correction. We can't just use FPKM fold change to infer differential expression.

L119. Explain (or hypothesize) why the 12 DPA sample has the largest number of eGene.

L121, which the distance cutoff used for cis-eQTL? 1Mb? Better write it in front at L112.

L126. What is the overlap of cis-eQTL of open chromatin regions?

L133 "state-shared eGene had a higher proportion of cis-eQTL than stage-specific eGenes" Why? And How come? State-share or -specific apply to eQTL, not the gene. Do you mean stage-shared eQTLs

have a higher proportion of cis-eQTL than stage-specific eQTLs?

L150, whenever we say "significantly higher" there should be a statistic test to support that claim.

L151. This is such a meaningless conclusion. Isn't it common sense that genes regulated by the same mechanism have the same expression pattern.

L165. "Most GWAS QTLs overlap". Please be precise, how many GWAS QTL are there? And how many of them overlap with TWAS. Also please explain how the GWAS QTL (eg how big is the region?) can overlap with TWAS peaks (that is individual gene's transcribed region?).

L207-209. Define what is favorable expression before using this term. If it is based on association of traits, then what is the cutoff and what association (pearson?), please write with clarity.

L211. "Aggregation of gene with favorable expression pattern was helpful to improve fiber quality" We shouldn't draw a conclusion like this. You define favorable expression pattern based on whether the expression correlate with fiber quality. It is like you first define rich people as the one has more money and then conclude that having more money can make people rich.

L216. "construct a regulatory network"? This is wrong. It is not a regulatory network. It is just merging eQTL identified from different tissues. I remember the Genotype-Tissue Expression Consortium (GTEx) would refrain from calling their multi-tissue eQTL data a network. They wont even say it is a "regulome". I suggest the authors to be more modest in writing and there is no need to use those fancy and not realistic words. The data is good enough, just describe it as it is.

L201-247. I personally found this whole result section "genetic regulatory modules underpinning fiber quality" rather empty, vague and meaningless. What did we actually learn from this analysis? If it achieved anything, haven't we already known it from the eQTL analysis? The most funny part is this. When the module show enrichment for particular trait, they say the module is "relatively functionally differentiated with respect to phenotypic effects (L226). When they didn't see a clear enrichment, they said "both modules may play a role in ... (L246). Then what is the point of having a fake network and analyze its module? If they really want to keep this section, they can use some softwares that can handle multiple tissue RNA-seq data for eQTL calling. I remember the GTEx consortium has done something like that.

L250 "the genetics of gene expression maybe regulated by variants.." this doesn't make sense.

L253. poor interpretation. Firstly, shouldn't most gene expression variance are mainly explained by cis-eQTL? Then why single out sub-genome genes? If they want to test something meaningful, they can look at the trans-eQTL, and ask whether the trans- and their target eGene are located in the same subgenome.

L260. Again, a lazy/meaningless conclusion. When we don't know how to explain, we can't just say "indicative of the complex genetic control". I think they should first get a reliable list of gene pairs with expression bias. Also can check the direction of the bias, and the expression level of these gene groups. Or check their expression profile to come up with a proper conclusion or hypothesis.

L262, no such thing as "genetic architecture"

L263. Need to call the bias expression gene with proper statistical test.

L268. Define clearly how "highly correlated", and what test?

L277, we don't know how the variants are gained or lost. So how can we know it is "unequal accumulation of variants .. lead to asymmetric regulation..". in maize, subgenome experienced unequal gene loss and have biased expression pattern. Is it possible what they saw in cotton is just like the maize? One of the duplicated genes is no longer needed or the plant can't tolerate a doubled expression level. So it either turned it off or express it in another tissue for another biological function.

L296. Define the 3 terms before using them.

L300. Define "highly correlate"

L301. If the dominant group is defined as pairs with consistent expression bias in all 5 tissues' RNA-seq data, isn't it common sense that we can call the same eQTL using these 5 RNA-seq data separately?

L304. This is common sense again. Trans-regulator = transcription factors. expression change of TF during plant development of coz will correlate with their target genes.

L314. How to test this is not random? Even if it is not random, how can we tell that these genes are just clustered together because they have lower connectivity? Then the whole argument following it will collapse.

L327. Define "flanking" clearly. How many bp up and downstream?

L328. Define "rarely mutated"

L330. Better provide some evidence showing that it is not due to alignment artefacts. Or the homologous gene are "dead" (no promoter open chromatin? Not expressed?)

L348. L367 plus the whole final result section "genomic design for fiber quality improvement". I think the authors make their case based on a rather thin ice here. So basically they are saying that the expression of a gene X is known to correlated with a trait, and the cotton genome has an extra copy of it (let's call it X') in the subgenome. in some cases, X' is not expressed like X (biased expression). They argue that since the breeders are already selecting for the gene X to improve the trait, why don't we use the other copy gene X' as well? That really depends on whether the effect of X and X' on the trait are cumulative.

L438 they are "potential" regulatory variants. eQTL variant does not guaranty casual regulation.

L449. As mentioned above, I would be a lot more cautious about this.

L452. Which part of the result can lead to this "combinatorial complexity" hypothesis?

L454. With the ancient whole genome duplication, isn't that most of the angiosperms are polyploid at

some stage? Eventually the duplicated genes are lost or neofunctionalized, and most sub-genome regions disappeared or mutated, and eventually merged into one. Cotton just happen to be a recent one. So the results in this paper doesn't seem to be too related to "how polyploid contribute to plant diversification".

L455. I believe this is a very good paper for nature genetics. But I don't buy the idea that subgenome coordination can be used for breeding (for now), unless they can perform an experiment to demonstrate that they can improve fiber quality or whatever trait using this strategy. So if they do not have such experimental data to show it, I suggest them do not make such claim or write clearly about the limitation of such.

Author Rebuttal to Initial comments

Reviewers' Comments:

Reviewer #1:

Remarks to the Author:

In this study, to dissect the genetic regulome of dynamic gene expression in allotetraploid cotton enables fiber improvement through subgenome coordination, the authors first construct a genetic regulome of gene expression in 376 diverse *G. hirsutum* accessions using transcriptome from 6 fiber development timepoints, then they performed comprehensive genetic regulome analysis, including eQTL, TWAS, bias eQTL, and multiple combination analyses, which led them to uncover the genetic components that may optimize homoeologous gene expression for unlocking the potential for fiber improvement. I would say this work is one of the most comprehensive genetic regulome analysis, at least in plants, and provides valuable information and highlight the significance of subgenomic coordination in fiber improvement in allotetraploid cotton.

Response: Thank you for your positive comments and suggestions, which help us craft a better manuscript. We have revised the manuscript thoroughly based on your suggestions.

Here are some comments that may help to further improve the manuscript:

1. I believe there must be some artificial error or noise for the eQTLs. The fact "the authors found that stage-shared eQTLs showed larger effects than stage-specific eQTLs, and stage-shared eGenes had a higher proportion of cis-eQTLs than stage-specific eGenes" may be related to the artificial error or

noise. The authors may consider to do some further analyses to check if there are some artificial errors. This may help to remove some artificial errors or noise eQTLs.

Another solution might be to focus on the leading eQTLs to perform the important analysis.

Response: We thank the reviewer for these comments. For comparative analysis of eQTL across stages, we followed the following processes: 1) We combined the lead SNPs identified in each stage based on linkage disequilibrium to obtain regulatory linkage blocks. 2) An eQTL was considered a stage-shared eQTL if it regulated the same gene in at least two stages and its lead SNP was located in the same LD block.

In this analysis, each linkage block consisted of one or more lead SNPs. For the comparative analysis of the eQTL effect size, we indeed compared the difference in the effect size of the lead SNPs of each eQTL, and provide more detailed information in the Methods (line 948-952, line 968-970) of the revised manuscript. For comparative analysis of eGene across stages, we classified genes into stage-shared eGenes or stage-specific eGenes according to whether significant eQTL (cis-eQTL or trans-eQTL) was detected for the same gene at different stages. The definition of stage-shared eGenes may cause ambiguity as it may be interpreted as the stage-shared eQTL regulating the gene in different stages. It is well known that genetic regulation often has tissue or spatiotemporal specificity, and the comparison of eQTL sharing in different tissues or stages can provide in-depth understanding of the differences in genetic regulation between tissues. For genes with eQTLs detected at multiple stages, they contained a higher proportion of cis-eQTLs. We think this may be related to the highly shared pattern of cis-eQTL among tissues (**Extended Data Fig. 2h**). We have provided a more detailed explanation of 'stage-shared eGenes' in the revised manuscript (line 134-137). Considering hidden batch effects and other global confounders, we estimated hidden covariates for gene expression using the peer method. The significant cis variant-gene associations were filtered by applying P value thresholds corresponding to $FDR < 0.05$. For trans-eQTLs, variant-gene pairs with P -value less than 3.76×10^{-7} were considered significant. The significant SNP with the strongest association signal was defined as the lead SNP for each association.

For this revision, to test the true association of these eQTLs, we randomly selected 5,000 eQTLs for permutation testing. The lead SNPs of 98.62% (4,931) eQTLs had significant differences (p -value < 0.05 , scaled value < -1.96 or > 1.96) in the expression differences between the two genotypes (**Fig. R1a**). For 69 non-significant eQTLs (scaled value > -1.96 and < 1.96), the reason may be that genes with very low expression levels may give rise to low statistical power when calculating the expression difference

between two genotypes (**Fig. R1b**). These results should demonstrate the robustness of our eQTL analysis.

Figure R1. Significance testing of eQTLs.

a: Comparison of the distribution of observed and permuted values for 5000 randomly selected eQTLs. The scaled values are calculated by measuring the difference in expression level for each lead SNP between two genotype samples.

b: Sample expression distribution of genes corresponding to two eQTLs.

2. The authors found that “Most GWAS QTLs overlapped with TWAS peaks, indicating that our data have good repeatability across different methods”. This part is very interesting. In fact, there are some specific QTLs for each method. For example, for the TWAS, there is a significant TWAS QTL in D03. These

kinds of specific TWAS QTLs may need to pay more attention, which also closely match to the character of the datasets from this study.

In addition, I am not quite understand why the TWAS show such high overlap pattern with GWAS: the QTLs are overlapped; the TWAS QTLs also showed association peak. Does that mean, most of the genetic variations of this region result in gene expression changes, and also related to the phenotype? Or one genetic variation resulted in the expression change of multiple genes? This is very interesting to me. The authors may try more detailed analyses for this part.

Response: Thank you very much for your suggestions. In this study, we identified 18 GWAS QTLs and 1,255 TWAS genes, and 17 of 18 QTLs overlapped with 43 of 1255 TWAS genes. Considering this, we have rewritten this sentence as “Specifically, 43 genes were prioritized as candidate genes for 17 GWAS QTLs using TWAS (**Supplementary Table 7**)” in the revised manuscript.

As Transcriptome-wide association study (TWAS) was “a powerful strategy that integrates gene expression measurements with summary association statistics from large-scale genome-wide association studies (GWAS) to identify genes whose *cis*-regulated expression is associated with complex traits” (Alexander et al., *Nat Genet*, 2016), we applied TWAS as an approach to assist GWAS to prioritize candidate genes. Actually, TWAS can also identify gene-trait associations that do not overlap with significant loci in the corresponding GWAS. As the developers of TWAS mentioned in their study (Alexander et al., *Nat Genet*, 2016), “Averaging over the new genes, the z^2 statistics from TWAS were 1.5× higher than the strongest eQTL SNP for the same gene (although this may be slightly inflated because of winner’s curse). Our previous simulations suggest that the substantial gain over testing the *cis*-eQTL is an indication of pervasive allelic heterogeneity at these loci, and analyses of expression showed strong evidence for allelic heterogeneity at the TWAS genes”. Considering this, we agree that these TWAS specific gene-trait associations may be due to the presence of allelic heterozygous loci.

As both GWAS and TWAS were developed based on association analysis but not fine-mapping, we cannot identify causal variances. To investigate whether one or multiple genetic variances lead to expression change of multiple genes, we used primary *cis*-eQTLs instead of causal variances to categorize 12 GWAS QTLs (that were overlapped with more than one TWAS genes) into two classes: class1, one *cis*-eQTL resulted in the expression change of multiple genes; class2, multiple *cis*-eQTLs resulted in the expression change of multiple genes. In these 12 GWAS QTLs, 1 QTL had one *cis*-eQTL that regulated the expression of multiple genes; 11 QTLs had multiple *cis*-eQTLs that regulated the expression of multiple genes. The detailed information was shown as **Table R1**:

Table R1. Classification of GWAS QTL containing multiple genes.

GWAS QTL ID	Cis -eQTL count	TWAS gene count	Class
qFE-A01	8	6	class2
qFE-D05	2	4	class2
qFE-D07	2	4	class2
qFL-A11	4	4	class2
qFL-D05	7	8	class2
qFS-A01	8	6	class2
qFS-A07	2	2	class2
qFS-A11	4	7	class2
qFS-D01	1	2	class1
qFS-D06	2	2	class2
qFS-D13	4	4	class2
qFU-D12	3	2	class2

3. For the “genetic regulatory network” part, it is very interesting and provides a lot of information. However, it sounds too complicated to me. If it can be simplified, it may give clearer clues. For example, for the 36 modules, most of them show tiny or even no contribution for the four traits. If these components were reduced, the major modules that dominantly regulate the fiber traits may be left, and may give more clear information.

Response: Thank you very much for your suggestions.

1) We performed this analysis in order to explore the regulatory unit (module) of fiber quality from a larger scale than single gene or single eQTL. Each module consists of a collection of genes related to fiber quality (TWAS. FDR < 0.05 or COLOC. PP. H4 > 0.8). During fiber development, we found multiple

genes were regulated by the same eQTL. We deduce that these co-regulated genes might have interconnected biological functions, and each module might exhibit special biological implications. So we presented the network that was clustered into modules, and analyzed the biological characters of the modules.

2) This genetic network was based on eQTL-eGene associations. The circular nodes and square nodes in the network represent genes and eQTLs, the hexagonal nodes represent eQTL hotspots, and the edges in the network represent the regulation of eQTL/hotspots on eGene.

3) We used the algorithm embedded in the Gephi software (v0.9.5) to cluster nodes into modules. The principle of modularization was to maximize the connectivity between nodes in the same module and minimize the connectivity between nodes in the different modules. The modularity value could reflect the quality of modularization. The closer the value to 1, the better the quality of the modularization was. The modularity of this network was 0.86, which suggested that the modules well described the characters of this network.

4) We quantified the heritability of eQTL for traits (FS, FL, FE, FU) in each module (**Fig. 3d**) and calculated the correlation coefficient (**Fig. 3e**) between the expression of genes and the phenotypic value in the module across timepoints. We identified 23 FL-related modules, 25 FS-related modules, 25 FU-related modules, and 26 FE-related modules with heritability ≥ 0.05 . Modules with heritability ≥ 0.05 and gene counts ≥ 5 were used for multiple regression of gene expression to phenotypic values across timepoints. For each module, the degree of correlation between gene expression and phenotype at a given timepoint was indicated by the magnitude of the normalized R-squared value. Higher normalized R-square value suggested a stronger association between gene expression and phenotype at that timepoint. The details were shown in **Fig. 3e**, **Supplementary Fig. 11a**, **Supplementary Fig. 12a**, and **Supplementary Fig. 13a**.

To make the content of network clearer to readers, we provided the details above in the **Supplementary Results** (please see Supplementary Information file page 19) and **Supplementary Table 9** in the revised manuscript.

Minor comments:

1. Line 98: in the text, it is described that "Among these, 74.2% were expressed at all timepoints". However, in the Extended Data Fig. 1d, no data shows the genes expressed at all timepoints.

Response: Thank you for your comments. For 49,860 expressed genes, 3,712 genes were expressed at only one timepoint (red label), 9,163 genes were expressed at 2-5 timepoints (green label, orange label, pink label and blue label), and 36,985 (74.2% of 49,860) genes were expressed at all 6 timepoints (black label). As the number of genes expressed at all 6 timepoints were far more than that of other genes, we just annotated gene counts with a label to make the **Extended Data Fig. 1d** clearer. We rewrote this sentence in the revised manuscript as “of which 12,875 were expressed at 1–5 timepoints (**Extended Data Fig. 1d**)”.

2. The authors found that “23,811 trans-eQTLs for 23,012 genes”, and then “a higher proportion of trans-eGenes were detected with multiple eQTL signals”. The two datasets are little confused. May need to further check or to revise the text a little bit to make it clear.

Response: Thanks very much for your suggestions. Considering that there may be multiple eQTLs for a single gene, we performed a comparative analysis of *cis*-eGenes and *trans*-eGenes in each timepoint, and pooled the statistical results of all timepoints. Compared with *cis*-eGenes, a higher proportion of *trans*-Genes were detected with multiple eQTL signals (**Table R2**). In total, 53,854 *cis*-eQTLs were identified for 18,637 genes and 23,811 *trans*-eQTLs were identified for 10,391 genes. The *cis*-eGenes contained a higher average number of eQTLs compared with *trans*-eGenes (2.88 vs 2.29). It is possible that the more shared pattern of *cis*-eQTL (**Extended Data Fig. 2h**) could introduce bias in the comparison of the number of genes with multiple eQTLs, which might be identified consistently for the same gene at different timepoints. To avoid confusion, we have removed this part “Compared with *cis*-eGenes, a higher proportion of *trans*-eGenes were detected with multiple eQTL signals” from the result and have rewritten this sentence to make it clearer (line 117-122).

Table R2. The number of genes with one or multiple eQTLs.

Stage	Cis -eGenes		Trans -eGenes	
	One eQTL	Multiple eQTLs	One eQTL	Multiple eQTLs
0 DPA	8,129	1,261	1,579	836
4 DPA	7,766	1,418	1,350	636
8 DPA	8,442	1,363	1,181	560
12 DPA	7,938	1,303	2,643	2,624
16 DPA	7,254	1,073	1,928	513

20 DPA	7,091	816	1,183	500
Total	46,620	7,234	9,864	5,669

3. It is better to give some explanation of the abbreviations in the text. For example, for Fig 4b, it is better to term the bias and biasN in the text. It is hard to get the meaning of “BiasN” without careful consider.

Response: We thank the reviewer for pointing out this issue. We have provided this information in the revised manuscript (line 101; line 260; line 498-501; line 673-674).

4. For the example of BB2, it is found that, from group 2 to group 5, the expression pattern of BB2 was gradually delayed, and the average FL of the corresponding accessions gradually increased. I am not sure if this is related to the maturity/flowering time which may affect the development of fiber development. If so, it might be better to give a discussion.

Response: Thank you for your suggestion.

1) Cotton fiber is epidermal hair differentiated from ovule epidermal cells, and each cotton fiber was a single cell. The development of cotton fiber includes 5 stages: initiation (0 to 3 DPA), elongation (3 to 16 DPA), transition (16 to 20 DPA), secondary cell wall synthesis (20 to 40 DPA), maturation (40 to 50 DPA) (Haigler et al., *Front Plant Sci.*, 2012). The length of cotton fiber is mostly determined by the “elongation” stage (3 to 16 DPA).

2) In *Arabidopsis*, *BB2* encodes an E3 ubiquitin ligase that regulates organ size, like leaf and root (Vanhaeren et al., *Plant Physiol.*, 2017; Cattaneo et al., *Plant Cell Physiol.*, 2017). The *bb* loss-of-function mutations prolong the cell proliferation phase in *Arabidopsis* root, and can influence cell elongation in genetic backgrounds in which the cell proliferation phase is severely shortened and cell differentiation is accelerated (Cattaneo et al., *Plant Cell Physiol.*, 2017).

3) We provide functional implication for how *BB2* participates in fiber development in the revised manuscript (line 179-183).

Reference:

1. Haigler, C. H. et al. Cotton fiber: a powerful single-cell model for cell wall and cellulose research. *Front Plant Sci.* **3**, 104 (2012).
2. Vanhaeren, H. et al. Forever Young: The Role of Ubiquitin Receptor DA1 and E3 Ligase BIG BROTHER in Controlling Leaf Growth and Development. *Plant Physiol.* **173**, 1269-1282 (2017).
3. Cattaneo, P. et al. BIG BROTHER Uncouples Cell Proliferation from Elongation in the Arabidopsis Primary Root. *Plant Cell Physiol.* **58**, 1519-1527(2017)
5. Line 253-254: It was found that expression variance of genes in each subgenome was mainly explained by intra-subgenomic variants. Based on Extended Fig 5a, the At and Dt for At eGene are close although the statistic has some difference.

Response: Thanks very much for pointing out this concern. We have checked these eQTL data. According to our statistics, 37.68% (4,087) of At-subgenomic genes had at least one inter-subgenomic eQTL, which showed a higher proportion than Dt-subgenomic genes (17.0%, 1,910). For the genes in the At subgenome, their genetic regulations between the At and Dt subgenomes had a significant difference, but the extent of difference was smaller than those for genes in the Dt subgenome. For the reasons outlined above, we have modified the related description (line 256-259).

6. For the part of “Definition of biased expression pattern in fiber development”, all cotton accessions were classified into 8 classes according to the biased expression patterns. There should be more complicated classes, such as 3 stage showed the one direction and the other 3 stages showed another direction. Are they belong to the ‘bias direction change’? Whereas, the definition of it looks like is not the case: direction of expression bias is different in any two timepoints.

In addition, what is the relationship between the 8 classes and the 3 groups (switched, time-dependent and dominant)?

Response: We appreciate these helpful comments. For each gene pair, the 340 accessions (with RNA-Seq data at all 6 timepoints) were classified into 8 classes according to their biased expression patterns in fiber development. The definition of class 8 may be confusing, which means the directions of expression bias from at least two timepoints are different. For each gene pair, the number of accessions in each class was counted. According to the number of accessions in each class, 2,658 gene pairs were sorted using unsupervised hierarchical clustering (**Extended Data Fig. 5g**). For the gene-pairs in the switched cluster, most of cotton accessions are presented in the bias expression of class 8. For the gene-pairs in the dominant cluster, most cotton accessions show the same bias direction across 6 timepoints. For the gene-pairs in the time-dependent cluster, most cotton accessions have no expression bias in at least one timepoint. We have modified the definition of class 8 and the other classes in the Methods of the revised manuscript to make this clearer (line 1051-1073).

7. As to the comprehensive analyses, the discussion is too simple. The authors may add more discussion, for example for the subgenome coordination, the cis/trans eQTL, and the candidate genes for the fiber development.

Response: Thank you very much for your suggestion. We have added the descriptions “These findings suggest that the genetic effect of genes on fiber quality should be evaluated at a specific developmental stage”, “This study highlights that the dissection of genetic basis of agronomic traits or identification of functional genes should consider the subgenomic counterparts in polyploid crops. We note that although we do not provide experimental evidence demonstrating that subgenome optimization in breeding programs actually has improved fiber quality, we point to this important consideration here.” to the revised manuscript (line 444-446; line 457-462).

Reviewer #2:

Remarks to the Author:

The work described in this paper uses extensive genotyping and transcriptome data from different stages of cotton ovule and fibre development in many diverse cotton accessions to identify eQTL that are associated with variation in gene expression. They identified 18 QTL for different fibre traits, of which 9 were new.

The main strengths of the work are the large datasets that permit identification of new genetic variants that can be used in breeding of this key crop. The current study aims to be comprehensive, such as defining the roles of genetic variation in homoeologous genes that influence gene expression in developing fibres and their influence on traits. The Introduction and Methods are succinctly and clearly laid out, giving hope that the general reader will be able to follow the work. However, the Discussion does little to relate the current work to the field and is a mere brief recitation of the Results.

Response: Thank you for these your comments and suggestions, which help us improve this manuscript. We have revised the manuscript thoroughly based on your suggestions and those of the other reviewers, and have particularly noted your comments about improving the Results and Discussion sections.

The major problem is the Results section, which is densely opaque and extremely difficult to follow and to interpret. I found the rationale for some approaches and analyses was not clearly explained, and the results obtained were sometimes over-interpreted and at times left quite vague. This is not a problem of the written English, as there are several native speakers as authors- some with the specific role of editing the manuscript. I think the problem lies in lack of experience in reducing complex observations into clear and easily understood messages. To do this the authors need, for each set of Results, to first define the objective and why it's important, then explain the choice of methods, and explain how significant the results are. As currently written, the reader (at least me) has no clear idea about why the set of analyses were carried out and could not work out its significance.

Response: Thank you very much for your suggestions to help us convey the results and relevance of the work more clearly. As suggested above, we tried to reorganize our manuscript to indicate: 1) what the question is. 2) what kind of data we generated. 3) how these data are analyzed. 4) the results. 5) possible biological interpretations. Especially in the Results for "Genetic modules underpinning fiber quality", we realized that two loosely connected parts were contained in this section, one part about "favorable expression patterns" and another about "genetic networks". In the revised manuscript, we moved the descriptions about "favorable expression patterns" to the section covering "Fine-mapping of fiber quality associations", and added descriptions related to eQTL hotspots, which were included in the "genetic networks". We also added the rationale about the reason why we think the section

“Subgenomic coordination of genetic effect on fiber quality” is important (Lines 320-321). In addition, to avoid over-interpretation, we try to tone down claims if we do not have enough experimental evidence, such as in lines 457-462. In addition, we provided more details in **Supplementary Results** (please see Supplementary Information file page 19), **Supplementary Methods** (please see Supplementary Information file page 20) and **Supplementary Table 9** to make this manuscript clearer to readers.

It may be that some experts in the field can follow the work clearly, but for publication in a general journal, the Results and Discussion sections need to be comprehensively re-written with the objective of helping the reader.

Response: Thank you very much for your suggestion. We have carefully rewritten paragraphs in the Results and Discussion, and provided more details in **Supplementary Results** (please see Supplementary Information file page 19), **Supplementary Methods** (please see Supplementary Information file page 20) and **Supplementary Table 9** to make this manuscript clearer to readers. Thank you again.

Reviewer #3:

Remarks to the Author:

You et al. have generated an impressive collection of cotton genome data. I can summarize their key results/findings as below:

- Re-sequence 376 cotton accessions. RNA-seq for 6 tissues for each accession.
- Compute eQTL separately for each tissue.
- Performed a fiber quality GWAS and found 18 QTL that overlaps with eQTL. Also did a TWAS and found 1255 genes associated with fiber quality.
- Combine eQTL derived from 5 fiber tissues to create a “regulatory network”
- They found gene expression differences of homologous genes in At and Dt subgenome.

- They also examined the expression pattern of some homologous gene pairs that are associated with fiber quality. Then they proposed that one can further improve fiber quality by targeting one of the “less favourably expressed” gene in the homologous gene pairs.

To me, this is an important genomic resource for cotton and did deserve a spot in a top journal. But there are lots of small problems in writing, analysis, missing statistics, and how to properly interpret the result. I list them below and hope they can correct those in the revised manuscript.

Response: Thank you for all your suggestions that help us improve this manuscript. We have revised the manuscript thoroughly based on your suggestions.

L101. If there is no replicate for the RNA-seq, how to call expression bias for the homologous gene pairs? And how to correct for multiple test? It appears to me that they only use FPKM fold change. At least one can use the raw count for a binominal test and then correct for a 5% FDR.

Response: We thank the reviewer for the suggestions.

Question: how to call expression bias for homoeologous gene pairs?

In this study, we defined expression bias of homoeologous gene pairs at the population level, but not at a single accession level. We have provided more details in the Methods (line 876-892) of the revised manuscript. We defined expression bias of homoeologous gene pairs as follows:

1) Calculation of expression bias score of homoeologous gene pairs in each accession (described in the Methods). For each expressed homoeologous gene pair (at least one gene of a pair with FPKM > 0.1) in each accession, a 2-fold expression change threshold is commonly used to defined differential expression (Love et al., *Genome Biol.*, 2014). Bias-At: the expression level of At-subgenomic gene is at least 2-fold that of the Dt-subgenomic gene; Bias-Dt: the expression level of Dt-subgenomic gene is at least 2-fold that of the At-subgenomic gene; Bias-N: no difference between homoeologous genes.

2) Definition of expression bias of homoeologous gene pairs at the population level. For each expressed homoeologous gene pair in all accessions, if the number of accessions with expression bias (at the single accession level) was greater than 5% of all accessions, this gene pair was identified as exhibiting biased homoeologous gene expression. Bias-At: The number of Bias-At (single accession level) was greater than 5% of all accessions; Bias-Dt: The number of Bias-Dt counts (single accession level) was greater than 5%

of all accessions; Bidirectional-bias: both the number of Bias-At (single accession level) and the number of Bias-Dt (single accession level) were greater than 5% of all accessions. Bias-N: other situations.

Question: how to correct for multiple test?

Actually, we can't correct or adjust the significance of gene pairs with differential expression in each accession based on the 2-fold FPKM change. However, our analysis was performed for a large population. In this study, we used a criterion similar to Bonferroni correction to limit the number of accessions with the same direction of expression bias and significant differential expression to at least 5% of accessions. We believe that this ensures that the expression bias of a gene pair is not by chance, as it is differentially expressed (2-fold change) in a sufficient number of accessions. To ensure that the expression bias of each homoeologous gene pair is not artifactual, we performed multiple testing for all homoeologous pairs with expression bias (2-fold change) in at least 5% accessions. For each homoeologous gene pair with biased expression, we compared the expression level at each timepoint between the At and the Dt in all accessions using two-sided Wilcoxon rank sum test and corrected the *P*-value with Benjamini-Hochberg procedure. Homoeologous gene pairs with expression bias in at least 5% accessions and false discovery rate (FDR) ≤ 0.05 were considered as a biased homoeologous gene pair at the population level. After FDR correction, the number of biased homoeologous gene pairs decreased slightly (**Table R3**). Based on this analysis, the related results and Figures have been modified in the revised manuscript (**Fig. 1d; Extended Data Fig. 1f; Fig. 4b-h; Extended Data Fig. 5c-h**).

Question: Is it possible to correct expression bias for homoeologous pairs by using raw reads counts?

Quantification of RNA-seq read counts at heterozygous sites or for the same genes between tissues is widely used to analyze allele-specific expression and discover genes with differential expression (Knowles et al., *Nat. Methods.*, 2017; Mohammadi et al., *Genome Res.*, 2017). The significance obtained through this method is based on the similarity of the gene structure. However, the comparison of homoeologous gene pairs in tetraploid cotton based on the read counts is challenging due to the difference in gene structure and gene length. In this study, we performed binomial tests for each gene pair based on the count of reads mapped to exon regions and adjust the *P*-value to obtain differentially expressed homoeologous gene pairs in each accession. The results were then compared with those based on 2-fold FPKM change. Here are the methods used to conduct this analysis:

- 1) The htseq-count (v2.0.2) was used to count the number of raw reads aligned to the exon region.

2) For each accession, gene pairs with more than 10 aligned reads (At reads plus Dt reads) were used for the binomial distribution test. Only gene pairs with the corrected p -value ≤ 0.05 (Bonferroni correction) were identified as differentially expressed gene pairs.

3) The expression bias of gene pairs in each accession was coded into three categories: -1 (significant bias-Dt), 0 (non-significant biased expression) and 1 (significant bias-At). We clustered the accessions based on the biased expression of all genes in each accession. The similarity of clustering for the two methods (based on 2-fold FPKM change and raw reads count) was calculated using the Rand index.

In total, 6.8% (1,644) of the gene pairs had differences in the number of exons, and 44.8% (10,786) of the gene pairs had at least 200bp difference of exon length (Fig. R2a). Therefore, when counting the difference in the number of homoeologous gene reads, bias caused by the difference in sequence length cannot be ruled out. We found that the similarity (Rand index) of the results based on the two strategies reached 79.95% (range from 78.4% to 81.2%) (Fig. R2b, c). The difference in sequence length may be responsible for more biased gene pairs that were identified in the raw read counting method (Table R4). These results show that the method for analyzing the differential expression of homoeologous gene pairs based on 2-fold FPKM changes is robust.

Table R3. The number of biased expression gene pairs across 6 timepoints.

Stage	2-fold-changes	FDR ≤ 0.05
0 DPA	9,382	9,286
4 DPA	10,214	10,100
8 DPA	10,659	10,538
12 DPA	10,943	10,807
16 DPA	11,519	11,335
20 DPA	11,077	10,895
Unique	16,124	16,081

Table R4. Comparison of the similarities between the two strategies.

Stage	Biased expression gene (FPKM)	Biased expression gene (read count)	Commonly biased expression gene	Rand index (%)
0 DPA	9,149	16,190	7,383	78.4
4 DPA	9,939	15,152	8,247	80.8

8 DPA	10,359	15,138	8,569	81.2
12 DPA	10,569	15,414	8,775	79.9
16 DPA	11,110	15,636	9,183	78.7
20 DPA	10,658	15,849	8,918	80.7

Figure R2. Comparison of sample clustering results based on two methods.

a: Comparison of exon length for homoeologous gene pairs.

b: Accessions were sorted based on 2-fold FPKM change method using unsupervised hierarchical clustering. This value represents the Pearson correlation of gene pair expression bias between accessions.

c: Accessions were sorted based on clustering (b). This value represents the Pearson correlation of gene pair expression bias (based raw reads count) between accessions.

Reference:

1. Love, M. I. et al. Moderated estimation of fold change and dispersion for RNA-seq data with DESeq2. *Genome Biol.* **15**, 550 (2014).
2. Knowles, D. A. et al. Allele-specific expression reveals interactions between genetic variation and environment. *Nat. Methods.* **14**, 699-702 (2017).
3. Mohammadi, P. et al. Quantifying the regulatory effect size of cis-acting genetic variation using allelic fold change. *Genome Res.* **27**, 1872-1884 (2017).

L103, a proper statistical test must be performed to call the At and Dt bias pairs in all tissue (time point), and also need multiple test correction. We can't just use FPKM fold change to infer differential expression.

Response: We thank the reviewer for this point. We agree that a proper statistical test or replicates should be performed when conducting expression analysis of individual accessions across stages. However, our analysis was performed for a large population. As explained above, we used a criterion similar to Bonferroni correction to limit the number of accessions with the same direction of expression bias and differential expression to at least 5% of accessions, which ensures that the expression bias of gene pair is not by chance at each timepoint. To reduce the false discovery rate for identifying expression bias in multiple homoeologous gene pairs, we performed multiple testing in homoeologous pairs with expression bias (at least 5% accessions with 2-fold changes). We determined the classification (Bias-At expression, Bias-Dt expression and Balanced expression) of expression bias for gene pair at each timepoint, and then took the intersection of these classification across 6 timepoints. We found 3,256 homoeologous gene pairs had a stable bias direction across 6 timepoints.

L119. Explain (or hypothesize) why the 12 DPA sample has the largest number of eGene.

Response: We have checked the eQTL data at each stage and observed the highest number of trans-eQTLs (8,548, 35.89%) identified at 12 DPA. These trans-eQTLs regulated the expression of 41.56% (5,267) eGenes at 12 DPA, which may be responsible for the presence of most eGenes at 12 DPA. Fiber development is associated with dynamic process of cell elongation, cell-wall biosynthesis, loosening, and expansion (Qin et al., *Curr. Opin. Plant Biol.*, 2011; Ruan et al., *Plant Cell.*, 2001; Cao et al., *Mol. Plant.*, 2020). Primary cell wall extension has been proposed to drive cotton fiber elongation (3 to 16 DPA), which was subject to termination with the onset of secondary cell wall deposition (20 to 40 DPA) (Haigler et al., *Frontiers Plant Sci.*, 2012). This complex development process involves multiple genes, which can be regulated by specific eQTL at different stages. We identified several stage-specific regulatory hotspots (> 100 trans-eQTLs) by scanning statistics of all lead SNPs at each stage (**Table R5**). It is possible that potential regulatory hubs containing transcription factors and lncRNAs, for example, may be activated at specific developmental stages (such as at 12 DPA) to help direct the stage-specific transcriptional program during cotton fiber development. To be sure, identification and characterization of such regulators represents an exciting future direction, one we hope to explore in future analyses.

Table R5. Genomic regions with multiple eQTLs.

Region	eGenes	Stages
Ghir_D11:24319170-24667186	1,546	12 DPA
Ghir_D10:12401557-12402806	999	12 DPA
Ghir_D05:62620512-62642506	156	0 DPA
Ghir_A12:93814122-93912132	113	16 DPA
Ghir_D01:10891800-11007616	573	16 DPA
Ghir_D11:24319170-24667186	125	16 DPA

Reference:

1. Qin, Y. M. et al. How cotton fibers elongate: a tale of linear cell-growth mode. *Curr. Opin. Plant Biol.* **14**, 106-111 (2011).
2. Ruan, Y. L. et al. The control of single-celled cotton fiber elongation by developmentally reversible gating of plasmodesmata and coordinated expression of sucrose and K⁺ transporters and expansin. *Plant Cell.* **13**, 47-60 (2011).

3. Cao, J. F. et al. The miR319-Targeted GhTCP4 Promotes the Transition from Cell Elongation to Wall Thickening in Cotton Fiber. *Mol. Plant.* **13**, 1063-1077 (2020).

4. Haigler, C. H. et al. Cotton fiber: a powerful single-cell model for cell wall and cellulose research. *Frontiers Plant Sci.* **3**, 104 (2012).

L121, which the distance cutoff used for cis-eQTL? 1Mb? Better write it in front at L112.

Response: We used 1 Mb for cis-eQTL identification, and provided this information in the revised manuscript.

L126. What is the overlap of cis-eQTL of open chromatin regions?

Response: In this study, we did not focus on open chromatin analysis in fiber development, so we did not perform DNase-seq or ATAC-seq experiments. During the revision, the published DNase-seq data of fibers at 10 DPA and 20 DPA from upland cotton TM-1 were downloaded from our previous study (Wang et al., *Nat. Genet.*, 2017) to identify open chromatin regions. In total, 153,295 and 124,294 open chromatin regions at 10 DPA and 20 DPA fibers were identified, respectively. There are ~24% or ~27% of cis-eQTLs overlapping with open chromatin regions at each timepoint (**Table R6**), as summarized below. We have also provided this information in the revised manuscript (Line 117-122).

Table R6. Cis-eQTLs overlapped with open chromatin regions.

	0 DPA eQTL	4 DPA eQTL	8 DPA eQTL	12 DPA eQTL	16 DPA eQTL	20 DPA eQTL
10 DPA	2330	2,214	2,358	2,248	2,027	1,884
DHS	(24.8%)	(24.1%)	(24.0%)	(24.3%)	(24.3%)	(23.8%)
20 DPA	2665	2,500	2,686	2,550	2,331	2,168
DHS	(28.4%)	(27.2%)	(27.4%)	(27.6%)	(28.0%)	(27.4%)

L133 “stage-shared eGene had a higher proportion of cis-eQTL than stage-specific eGenes” Why? And How come? Stage-share or -specific apply to eQTL, not the gene. Do you mean stage-shared eQTLs have a higher proportion of cis-eQTL than stage-specific eQTLs?

Response: We thank the reviewer for these comments. The definition of “stage-shared eGene” here may be misleading. For comparative analysis of eGenes across stages, we classified genes into stage-shared eGenes or stage-specific eGenes according to whether eQTL (cis-eQTL or trans-eQTL) were detected for the same gene at different stages. The definition of stage-shared eGenes may cause ambiguity as it may be interpreted as the stage-shared eQTL regulating the gene at different stages. In this regard, why do genes with eQTLs detected at multiple stages have a higher proportion of cis-eQTLs? We speculate that this may be attributed to the stronger regulatory effect of cis-eQTL than trans-eQTL, which leads to a high degree of sharing of cis-eQTLs among different tissues (**Extended Data Fig. 2h**). We have rewritten this sentence in the revised manuscript (line 134-137).

L150, whenever we say “significantly higher” there should be a statistic test to support that claim.

Response: Thank you for your suggestion. We have added the corresponding statistical method and significant *P*-values in the revised manuscript (line 154).

L151. This is such a meaningless conclusion. Isn't it common sense that genes regulated by the same mechanism have the same expression pattern.

Response: If a gene is regulated by shared eQTLs at different timepoints, it may lead to similar expression patterns. However, gene expression is regulated by multiple variants with varying degrees of complex and interacting regulatory effects, and there may be other stage-specific regulators in addition to the shared cis regulation. Considering this, we performed this analysis and speculated that the high similarity in gene expression with shared cis-eQTLs may suggest the consistency of their genetic regulation across stages. In the revised manuscript, we have modified this sentence (line 152-155).

L165. “Most GWAS QTLs overlap”. Please be precise, how many GWAS QTL are there? And how many of them overlap with TWAS. Also please explain how the GWAS QTL (eg how big is the region?) can overlap with TWAS peaks (that is individual gene's transcribed region?).

Response: Thank you for these suggestions.

1) We identified a total of 18 QTLs related to fiber quality, 5/5 FL-related QTLs, 6/6 FS-related QTLs, 4/4 FE-related QTLs, and 2/3 FU-related QTLs overlapped with TWAS genes.

2) In this analysis, we mainly wished to show that most GWAS QTL regions included TWAS genes. We realized that this sentence would mislead readers, so we replaced “Most GWAS QTLs overlap” with “Specifically, 43 genes were prioritized as candidate genes for 17 GWAS QTLs using TWAS (Supplementary Table 7)” in the revised manuscript. As for the length of QTL regions, we performed local LD analysis for each QTL and found the average length was 0.9 Mb for all QTLs.

L207-209. Define what is favorable expression before using this term. If it is based on association of traits, then what is the cutoff and what association (pearson?), please write with clarity.

Response: The expression pattern of genes in accessions with favorable traits was defined as a “favorable expression pattern”. This analysis was performed as follows:

1) Filtering genes. For candidate gene identified through TWAS or co-localization analysis, only genes with more than one expression pattern in 340 accessions (sampled in all 5 timepoints) were retained (**Extended Data Fig. 3d**). 2) Filtering expression pattern. For each retained gene, the expression patterns were retained only if they corresponded to at least 3 accessions. 3) Significance test. For each retained gene, a significance analysis (wilcox.test) of phenotypic values was performed between accessions corresponding to different expression patterns. A P -value < 0.05 indicates that there were significant differences between the two expression patterns. 4) The expression patterns corresponding to the accessions with significantly higher phenotypic values than other patterns were identified as favorable expression patterns.

To make this clear to readers, we have defined “favorable expression” in the revised manuscript (line 188-191). We also provide more detailed methods in the revision, in the hopes of making this clearer (please see Supplementary Information file page 20).

L211. “Aggregation of gene with favorable expression pattern was helpful to improve fiber quality” We shouldn’t draw a conclusion like this. You define favorable expression pattern based on whether the expression correlate with fiber quality. It is like you first define rich people as the one has more money and then conclude that having more money can make people rich.

Response: Thank you for this comment! We thought you used the example of "rich people" to show that we have a mistake of "given A=B, then B=A". Here A was "rich people" and B was "having more money". In fact, we did things more like "given A=B, verify A=C". Here A was "good fiber quality", B was "a single gene with favorable expression pattern", and C was "multiple genes with favorable expression pattern". In this analysis, each gene with favorable expression pattern may improve fiber quality, but aggregation of different genes may yield fiber quality that is not improved. This may be because there are complex interactions between genes. Considering this, the accumulation of favorable genes does not necessarily lead to better fiber quality, so we performed this analysis and made this speculation. We realized that this sentence might mislead readers, so we removed this sentence in the revised manuscript.

L216. "construct a regulatory network"? This is wrong. It is not a regulatory network. It is just merging eQTL identified from different tissues. I remember the Genotype-Tissue Expression Consortium (GTEx) would refrain from calling their multi-tissue eQTL data a network. They won't even say it is a "regulome". I suggest the authors to be more modest in writing and there is no need to use those fancy and not realistic words. The data is good enough, just describe it as it is.

Response: Thank you these helpful comments. In this analysis, we focused on eQTL–eGene associations during fiber development. As we know that some eQTLs (especially eQTL hotspots) can regulate multiple eGenes and some eGenes can also be regulated by multiple eQTLs, such eQTL–eGene associations constitute a form of complex genetic network. We realize now that this sentence may have misled readers, so based on your comment we rewrote this to indicate that "The eQTL/hotspot–eGene relationships from five timepoints constituted a comprehensive genetic network" in the revised manuscript (line 217-218).

L201-247. I personally found this whole result section "genetic regulatory modules underpinning fiber quality" rather empty, vague and meaningless. What did we actually learn from this analysis? If it achieved anything, haven't we already known it from the eQTL analysis? The most funny part is this. When the module show enrichment for particular trait, they say the module is "relatively functionally differentiated with respect to phenotypic effects (L226). When they didn't see a clear enrichment, they said "both modules may play a role in ... (L246). Then what is the point of having a fake network and analyze its module? If they really want to keep this section, they can use some softwares that can handle multiple tissue RNA-seq data for eQTL calling. I remember the GTEx consortium has done something like that.

Response: Thank you for this perspective, which we have incorporated as follows:

1) We performed this analysis to explore the regulatory unit (module) of fiber quality on a larger scale than single genes or single eQTL. The genetic network showed complex eQTL-eGene associations (one eQTL may be associated with multiple eGenes, one eGene may be associated with multiple eQTLs). We are sorry that our inadequate description caused this misunderstanding.

2) The algorithm built in the Gephi (v0.9.5), a frequently-used software in wheat and maize networks (He et al., *Nat. Commun.*, 2022; Tu et al., *Nat. Commun.*, 2020), was used to cluster nodes (eQTLs/hotspots and eGenes) into modules. The principle of modularization was to maximize the connectivity between nodes in the same module and minimize the connectivity between nodes in the different modules. The modularity value could reflect the quality of modularization. The closer the value is to 1, the better is the quality of the modularization. The modularity of this network was 0.86, which suggests that the modules well-describe the characters of this genetic network.

3) To explore the biological function of each module, we quantified the heritability of eQTL for traits (FS, FL, FE, FU) in each module (**Fig. 3d**). In this analysis, 23 FL-related modules, 25 FS-related modules, 25 FU-related modules, and 26 FE-related modules were found with heritability ≥ 0.05 .

4) To explore the key timepoint of each trait-related module, we calculated the correlation coefficient (**Fig. 3e**) between the expression of genes and the phenotypic value in the module across timepoints. Modules with heritability ≥ 0.05 and gene counts ≥ 5 were used for multiple regression of gene expression to phenotypic values across timepoints. For each module, the degree of correlation between gene expression and phenotype at a given timepoint was indicated by the magnitude of the normalized R-squared value. Higher normalized R-square values suggest a stronger association between gene expression and phenotype at that timepoint.

5) We are sorry for the ambiguity caused by the descriptions in **L226 and 246**. In fact, when filtering the modules with heritability > 0.05 , we identified 2 trait-specific modules associated with single trait, 32 pleiotropic modules associated with 2-4 traits, and 2 modules without trait association (module heritability < 0.05) (**Supplementary Table 9**). That is evidence to support our summary “relatively functionally differentiated with respect to phenotypic effects (L226)”, and that “both modules may play a role in ... (L246)”.

To make the content of this analysis clearer to readers, in this revision we provide more details in **Supplementary Results** (please see Supplementary Information file page 19) and **Supplementary Table 9**. Thank you again for stimulating these important changes.

Reference:

1. He, F. et al. Genomic variants affecting homoeologous gene expression dosage contribute to agronomic trait variation in allopolyploid wheat. *Nat. Commun.* **13**, 826 (2022).
2. Tu, X. et al. Reconstructing the maize leaf regulatory network using CHIP-seq data of 104 transcription factors. *Nat. Commun.* **11**, 5089 (2020).

L250 “the genetics of gene expression maybe regulated by variants..” this doesn’t make sense.

Response: Thank you for this suggestion. This was a mistake in writing. We reorganized this sentence into "gene expression may be regulated by eQTLs from both the At and Dt subgenomes or either of them" in the revised manuscript (line 253-254) to make the sentence meaningful.

L253. poor interpretation. Firstly, shouldn’t most gene expression variance are mainly explained by cis-eQTL? Then why single out sub-genome genes? If they want to test something meaningful, they can look at the trans-eQTL, and ask whether the trans- and their target eGene are located in the same subgenome.

Response: Thank you this comment. We agree with you that most gene expression variance is explained by cis-eQTLs. But in this section, we want to distinguish the contribution of different subgenomes, of individual homoeologs, rather than focusing on the analysis of cis- and trans-eQTL, as the co-existing subgenomes in allotetraploid cotton undoubtedly is important and complicates the regulation of gene expression. To partition of genetic variation for gene expression, we accounted for the cumulative effect of all SNPs (SNP-based heritability) from the distinct subgenomes, and identified more ‘inter-subgenomic’ regulation from the At subgenome compared with the Dt subgenome (**Extended Data Fig. 5a, b**). As you mention, we counted the number of regulations representing that trans-eQTLs and their target eGenes located in the different subgenomes (**Fig. 4a**), which exhibited more ‘inter-subgenomic’ trans-eQTLs in the At subgenome than Dt subgenome. We have made necessary modifications in this sentence to make it clearer to readers.

L260. Again, a lazy/meaningless conclusion. When we don’t know how to explain, we can’t just say “indicative of the complex genetic control”. I think they should first get a reliable list of gene pairs with

expression bias. Also can check the direction of the bias, and the expression level of these gene groups. Or check their expression profile to come up with a proper conclusion or hypothesis.

Response: We thank the reviewer for sharing this reaction, which causes us to write more precisely. All of the gene pairs (17,878 in total) were expressed in at least 5% of accessions and were used to detect eQTLs. This ensures that there is no bias introduced in the comparison of the two groups of gene pairs (Bias and BiasN). For each stage, we checked the proportion of homoeologous gene pairs that had at least one homoeologous gene identified as an eGene. Among these gene pairs with expression bias, a higher proportion of genes had eQTLs (**Table R7**). This suggested genetic variation in the regulatory regions of gene pairs having biased expression, which prompted us to conduct a bias-GWAS analysis to identify variants that regulate the expression of homoeologous gene pairs. We have rewritten this as “suggesting genetic variation that might lead to expression bias of homoeologous genes” (line 259-262).

Table R7. Comparison of the number of eGenes between Bias- and BiasN-gene pairs.

Stage	Bias gene pairs (eGene/non-eGene)	BiasN gene pairs (eGene/non-eGene)	Fisher' exact test P -value
0 DPA	2,584/3,090	1,570/3,836	$< 2.2 \times 10^{-16}$
4 DPA	2,846/3,151	991/2,539	$< 2.2 \times 10^{-16}$
8 DPA	2,987/3,054	1,010/2,391	$< 2.2 \times 10^{-16}$
12 DPA	3,705/2,668	1,171/2,077	$< 2.2 \times 10^{-16}$
16 DPA	2,966/3,776	793/2,169	$< 2.2 \times 10^{-16}$
20 DPA	2,740/3,481	690/1,962	$< 2.2 \times 10^{-16}$
Total	17,828/19,220	6,225/14,974	$< 2.2 \times 10^{-16}$

L262, no such thing as “genetic architecture”

Response: In the revised manuscript, "genetic architecture" was changed to "genetic regulation" (line 263).

L263. Need to call the bias expression gene with proper statistical test.

Response: As mentioned above, gene pairs with biased counts (single accession level) greater than 5% of all accessions were identified as biased homeologous gene pairs. To identify genetic variances associated with homeologous gene expression bias, we performed bias-eQTL analysis. Different from eQTL analysis, we used bias score (line 957-958) instead of gene expression as the phenotype to perform genome-wide association analysis. The genetic variance with P -value less than 3.76×10^{-7} ($1/n$, where n is the total number of genomic SNPs) was considered significant with homeologous gene expression bias.

L268. Define clearly how “highly correlated”, and what test?

Response: We evaluated the effect of the lead SNP for each significant bias-eQTL on the regulation of gene pairs with expression bias and also on the expression of individual genes. To test the correlation between these two effect values, we performed Pearson correlation analysis on all gene pairs with significant bias-eQTLs. We have provided the corresponding correlation coefficient and significant P -values in the revised manuscript (line 269-272).

L277, we don't know how the variants are gained or lost. So how can we know it is “unequal accumulation of variants .. lead to asymmetric regulation..”. in maize, subgenome experienced unequal gene loss and have biased expression pattern. Is it possible what they saw in cotton is just like the maize? One of the duplicated genes is no longer needed or the plant can't tolerate a doubled expression level. So it either turned it off or express it in another tissue for another biological function.

Response: Thank you for these interesting comments. In maize, the two genomes are differentiated by ongoing fractionation, which refers to the bias in gene loss and retention between duplicated genome segments (Schnable et al., *PNAS.*, 2011; Hufford et al., *Science*, 2021). Additionally, there is a pattern of overexpression of genes in the genome that has experienced less gene loss. In cotton, there is also bias in the loss of genes and the retention of duplicated segments. Compared to the diploid A genome (A_1 -genome, 1,556 Mb; A_2 -genome, 1,637 Mb), the At subgenome of upland cotton (1,413 Mb) is significantly reduced, while the Dt subgenome has expanded from 738 Mb in the putative D genome donor (D_5 -genome) to the current 820 Mb (Wang et al., *Nat. Genet.*, 2019). We also found that more gene pairs (At vs Dt, 1,533 vs 1,723) exhibited the pattern of biased expression from the Dt genome (**Extended Data Fig. 1f**). The reasons and mechanisms underlying genome fractionation after polyploidization are still unclear in cotton, but may be related to the activity of transposable elements (TEs) and epigenetic modifications near genes. In this study, we were indeed unable to assess whether

the regulatory sequences were lost or gained, but we observed differences in the frequency of variation of these sequences across subgenomes. We have modified the corresponding description in the revised manuscript (line 278-279) to make this clearer.

Reference:

1. Schnable, J. C. et al. Differentiation of the maize subgenomes by genome dominance and both ancient and ongoing gene loss. *PNAS*. **108**, 4069-4074 (2011).
2. Wang, M. et al. Reference genome sequences of two cultivated allotetraploid cottons, *Gossypium hirsutum* and *Gossypium barbadense*. *Nat. Genet.* **51**, 224–229 (2019).

L296. Define the 3 terms before using them.

Response: We now describe these three groups in more detail (line 1051-1073) in the Methods section.

L300. Define “highly correlate”

Response: We calculated expression bias levels for each gene pair within the three dynamic expression bias groups (switched, time-dependent, and dominant) and calculated the Spearman correlation coefficient of their eQTLs across stages (**Fig. 4f**). Compared with other groups (time-dependent and dominant), the switched group exhibited changes for the direction of expression bias at different stages and had relatively low correlation of genetic effect of eQTLs across stages. We have rewritten this sentence to make it clearer in the revised manuscript (line 299-301).

L301. If the dominant group is defined as pairs with consistent expression bias in all 5 tissues’ RNA-seq data, isn’t it common sense that we can call the same eQTL using these 5 RNA-seq data separately?

Response: Thanks for this comment. For a gene pair X exhibiting consistent expression bias towards the At subgenome (Bias-At) in all 6 timepoints, the Bias-At may be caused by regulatory variation in the At subgenome that leads to up-regulation of At expression at 4 DPA. Alternatively, Bias-At may be caused by a regulatory variation in the Dt subgenome that leads to down-regulation of Dt expression at another

timepoint such as 8 DPA. Therefore, the same direction of bias across timepoints may be regulated by different genetic variations. Of course, some gene pairs of the dominant group may be regulated by the same eQTLs. Actually, we want to investigate the dynamics of genetic regulation on gene expression, so we identified eQTLs at each of the 6 RNA-Seq datasets.

L304. This is common sense again. Trans-regulator = transcription factors. expression change of TF during plant development of coz will correlate with their target genes.

Response: Thank you for this perspective. As we know, variation in cis regulatory regions accounts for most of expression variation. Differences in cis regulatory sequences between homoeologous gene pairs may lead to expression bias. During development, however, the expression level of transcription factors (trans-regulator) typically changes at specific stage, which may result in a change in the direction of the homoeologous expression bias. Of course, the expression change of many TFs during plant development will correlate with their target genes. The target genes may also be regulated by multiple TFs, which will increase the complexity of gene regulation, so it is also likely that at some special conditions, the expression change of a specific TF may not be correlated with the target genes. We have removed this sentence in the revised manuscript.

L314. How to test this is not random? Even if it is not random, how can we tell that these genes are just clustered together because they have lower connectivity? Then the whole argumemt following it will collapse.

Response: Thank you for your comments. In this analysis, weighted correlation network analysis (WGCNA) was used to find highly correlated gene clusters according to the expression correlation patterns of genes among different accessions. In this method, genes with high connectivity are usually clustered in the same gene cluster, while genes with low connectivity are clustered in different clusters or even isolated. In our result, we found gene clusters (1, 7 and 8) were enriched with genes from the switched groups (**Extended Data Fig. 5 i-j, l**), and the genes in clusters (1,7 and 8) showed lower network connectivity than genes in other clusters (**Fig. 4j**). Therefore, we speculate that homoeologous genes with switched expression bias direction during fiber development may have simpler regulatory relationships. Before the clustering analysis, we did not know homoeologous genes with switched expression bias had lower connectivity, and we just made this speculation after this analysis. This phenomenon might be explained that simpler regulatory relationships, so expression bias might

switched direction. We do not have additional evidence supporting this speculation at present, but we do see this as a promising avenue for future studies. Thank you again for pointing out this.

L327. Define “flanking” clearly. How many bp up and downstream?

Response: Thank you for your suggestion. We define “flanking” region as 2 Mb up and downstream of the homoeologous gene. We have provided more details information (line 327) in the revised manuscript.

L328. Define “rarely mutated”

Response: Thank you for your suggestion. Rarely mutated means that the site exhibits few variations (frequency <0.01) among the samples. We have provided this detail (line 328) in the revised manuscript.

L330. Better provide some evidence showing that it is not due to alignment artefacts. Or the homologous gene are “dead” (no promoter open chromatin? Not expressed?)

Response: Thank you very much for your suggestion. In this analysis, we combined sequence alignment and gene synteny analysis to identify subgenomic homoeologous sequences (pseudo-regulatory loci). The alignment results were filtered by setting a minimum sequence identity threshold of 80% and a minimum sequence coverage of 85%. Meanwhile, we retained the alignment regions locating in gene syntenic regions (2 Mb up and downstream of the homoeologous gene). In order to verify the accuracy of the alignment results, we identified 62,208 subgenome syntenic regions with a median length of 3,540 bp using MUMmer (v 4.0.0) and Assembly (v1.2.1). For 934 loci without detected conserved pseudo loci in homoeologous gene region (2 Mb up and downstream), 88.9% (831) of these loci were in the noncollinear region between the subgenome. However, 48.5% (524) loci with conserved pseudo loci in homoeologous gene region were located in the subgenome syntenic regions (**Fig. R3a**). High ratio of conserved pseudo loci and low ratio of unconserved loci in the subgenome syntenic region suggested the robustness of our results in the two methods. Significant association signals may not be identified due to low gene expression (dead). We checked the expression of the candidate genes and its homoeologs in the accessions. More than 71.7% of the genes (candidate genes: 1,316; homoeologs: 1,195) had FPKM greater than 1 in at least 350 samples, and the median expression level (FPKM) in all samples was approximately 20 (**Fig. R3b, c**), suggesting that these homoeologous genes are not dead.

Figure R3. Expression profile of homoeologous gene pairs.

- a. The distribution of regulatory loci in the subgenome syntenic regions.
- b. The count of gene pairs with different numbers of samples.
- c. The comparison of median expression of homoeologous gene pairs.

L348. L367 plus the whole final result section “genomic design for fiber quality improvement”. I think the authors make their case based on a rather thin ice here. So basically they are saying that the expression of a gene X is known to correlated with a trait, and the cotton genome has an extra copy of it (let’s call it X’) in the subgenome. in some cases, X’ is not expressed like X (biased expression). They argue that since the breeders are already selecting for the gene X to improve the trait, why don’t we use the other copy gene X’ as well? That really depends on whether the effect of X and X’ on the trait are cumulative.

Response: Thank you for this thoughtful comment. We agree that improving phenotypes by controlling the expression of another homoeologous gene depends on whether the effects of these genes are cumulative and without negative pleiotropic effects. In this manuscript, we hold the view that if a gene can make the phenotype better, then we can use this gene and its functionally similar homoeologous gene for genetic improvement. If the expression of the homoeologous gene is significantly lower than

the other gene, we speculate that improving the expression of the homoeologous gene can make the phenotype better. Actually, this idea was mentioned in many previous publications (Ramírez-González et al., *Science*, 2018; Osborn et al., *Trends Genet.*, 2003; Li et al., *Plant Cell.*, 2016; Yan et al., *Theor. Appl. Genet.*, 2004; Zhao et al., *New Phytol.*, 2018). In this study, we highlight the necessity of the prebreeding strategy using genomic data based on this idea. Although manipulation of the expression of homoeologous genes seems to be challenging, we believe that with advances in biotechnology, this approach may soon become feasible. In view of the fact that this assumption has no experimental evidence at present, we include comments on limitations of this strategy in the Discussion (line 459-462).

Reference:

1. Ramírez-González, R. H. et al. The transcriptional landscape of polyploid wheat. *Science* **361**, (2018).
2. Osborn, T. C. et al. Understanding mechanisms of novel gene expression in polyploids. *Trends Genet.* **19**, 141-147 (2003).
3. Li, Z. et al. Gene Duplicability of Core Genes Is Highly Consistent across All Angiosperms. *Plant Cell.* **28**, 326-344 (2016)
4. Yan, L. et al. Allelic variation at the VRN-1 promoter region in polyploid wheat. *Theor. Appl. Genet.* **109**, 1677-1686 (2004).
5. Zhao, B. et al. Core cis-element variation confers subgenome-biased expression of a transcription factor that functions in cotton fiber elongation. *New Phytol.* **218**, 1061-1075 (2018).

L438 they are “potential” regulatory variants. eQTL variant does not guaranty causal regulation.

Response: Thank you for your suggestion. In the revised manuscript, "regulatory variants" was changed to "potential regulatory variants" (line 438-440).

L449. As mentioned above, I would be a lot more cautious about this.

Response: Thank you for your suggestion. We now write “We note that although we do not provide experimental evidence demonstrating that subgenome optimization in breeding programs actually has

improved fiber quality, we point to this important consideration here.” in the revised manuscript (line 459-462).

L452. Which part of the result can lead to this “combinatorial complexity” hypothesis?

Response: Thank you for this question. Combinatorial complexity arises from the interplay of various genetic elements, including homoeologous genes, which often exhibit biased expression patterns during fiber development (**Fig. 4f**). In addition, the coexistence of two subgenomes within the same nucleus leads to more than additivity in terms of regulatory possibilities, because of the added dimension of interactions between the myriad genes and regulatory regions contributed by the two different genomes (**Fig. 4a**). Thus, polyploid genomes exhibit greater than additive combinatorial complexity. We have modified this sentence to make this speculation clearer to readers in the revised manuscript.

L454. With the ancient whole genome duplication, isn't that most of the angiosperms are polyploid at some stage? Eventually the duplicated genes are lost or neofunctionalized, and most sub-genome regions disappeared or mutated, and eventually merged into one. Cotton just happen to be a recent one. So the results in this paper doesn't seem to be too related to “how polyploid contribute to plant diversification”.

Response: Thank you for your suggestions. We agree with you that “most of the angiosperms are polyploid at some stage” and “Cotton just happen to be a recent one”. We have removed the speculation that “how polyploid contributes to plant diversification” in the revised manuscript.

L455. I believe this is a very good paper for nature genetics. But I don't buy the idea that subgenome coordination can be used for breeding (for now), unless they can perform an experiment to demonstrate that they can improve fiber quality or whatever trait using this strategy. So if they do not have such experimental data to show it, I suggest them do not make such claim or write clearly about the limitation of such.

Response: Thank you for your suggestions. While we agree that it is true that we did not provide relevant experimental data to prove that “subgenome coordination can be used for breeding”, we hold the view that it will be very important for breeding in polyploid plants as we learn more about complex regulatory relationships and intergenomic interactions. We have tried to be clearer in this revision on the limitations of our current capabilities and state of knowledge (line 459-462). Considering this, we

modified the manuscript title as “Genetic regulome of dynamic gene expression in allotetraploid cotton informs fiber improvement through subgenome coordination” to replace “Genetic regulome of dynamic gene expression in allotetraploid cotton enables fiber improvement through subgenome coordination”. Meanwhile, we modified the last sentence in the Abstract as “...highlights the potential of subgenomic coordination underpinning phenotypes in polyploid plants”. Thank you again for all your nice comments to help improve this manuscript

Decision Letter, first revision:

5th May 2023

Dear Professor Wang,

Your Article, "Genetic regulome of dynamic gene expression in allotetraploid cotton informs fiber improvement through subgenome coordination" has now been seen by 3 referees. You will see from their comments below that while they find your work of interest, some important points are raised. We are interested in the possibility of publishing your study in Nature Genetics, but would like to consider your response to these concerns in the form of a revised manuscript before we make a final decision on publication.

To guide the scope of the revisions, the editors discuss the referee reports in detail within the team with a view to identifying key priorities that should be addressed in revision. **In this case, the referees have serious concerns about the manuscript structure and writing. We ask that you substantially improve the clarity of the writing in the main text and the title, and address their comments as thoroughly as possible with appropriate revisions. You might want to consider seeking professional help on the English writing (for example: <https://authorservices.springernature.com/>).** We hope that you will find the prioritized set of referee points to be useful when revising your study.

We therefore invite you to revise your manuscript taking into account all reviewer and editor comments. Please highlight all changes in the manuscript text file. At this stage we will need you to upload a copy of the manuscript in MS Word .docx or similar editable format.

*1) Include a “Response to referees” document detailing, point-by-point, how you addressed each referee comment. If no action was taken to address a point, you must provide a compelling argument. This response will be sent back to the referees along with the revised manuscript.

*2) If you have not done so already please begin to revise your manuscript so that it conforms to our

Article format instructions, available

[here](http://www.nature.com/ng/authors/article_types/index.html).

*3) Include a revised version of any required Reporting Summary:

[redacted]

We hope to receive your revised manuscript within about eight weeks. If you cannot send it within this time, please let us know.

Sincerely,
Wei

Wei Li, PhD
Senior Editor
Nature Genetics
New York, NY 10004, USA
www.nature.com/ng

Reviewers' Comments:

Reviewer #1:

Remarks to the Author:

The revised manuscript has addressed all my questions. I have no further comments.

Reviewer #2:

Remarks to the Author:

although the authors have attempted to describe their results in as clear way as possible for the general reader, the use of jargon such as "regulome" obscures whatever value the analyses may have.

The Discussion is also still problematic. It does not concisely describe the main findings except for (I think) that during breeding useful variation in homoeologs has not been exploited to its full extent. There are empty statements eg lines 454-457 that the increased complexity of homoeolog gene expression accounts for transgressive phenotypes. There are no insights there.

Reviewer #3:

Remarks to the Author:

I agree with reviewer #2's previous comments that the result was poorly written. Seems the revised version is no good either. It might not be possible for a non-expert to comprehend. The authors do not seem to understand how to present a story in a logical, concise and meaningful way. For each section, they seldom follow a regular flow such as: what is the goal, why we want to analyze this, why we use this method instead of others, what are the key findings, then come up with a hypothesis leading to the next result section. Currently, it is like a collection of all the results coming out from the analysis and all the figures. I suggest the authors to think about which part needs to be kept, which part can be removed without hurting your conclusion? currently, a lot of them do not seem to be related to the main story line, and they can go to supplementary. Do we need all these figures? It is not a show off of work load. Just pick the ones that can support the hypothesis. On the meantime, it can free up some space to write more about those findings that matter, which are currently under-explained making it very hard to read.

As for the discussion, it is like a recap of the result. But it is a common problem for this type of paper. Unlike the GWAS and eQTL studies in medical science, which can find causal SNP for disease diagnostic or treatment, plant population genetic projects like this often come up with lots of data without immediate application. The only core value is the genomic data itself and the authors have to come up with a fancy story. Like this, they tried to claim that for allotetraploid crops, people can use homoeologous genes in the sub-genome for breeding. As I commented previously, other plants have whole-genome duplication and have ortholog gene pairs within the genome, and there is no need to prioritize the homoeologs. I agree that without content, writing a meaningful and relevant discussion is indeed challenging. Maybe the authors can trim down their discussion to a short paragraph and tell

the reader what is their data, how it can be used for their research, and how to access them?

Also, it would be helpful if the author highlights the changes in the text.

Here is my suggestions for them to improve the result section:

L93. There is no need to draw a vague conclusion like "These data represent a rich resource for characterizing gene expression profiles in fiber development."

L95. "FPKM >0.1" is used to defined expressed genes. Why choose this? We should perform the analysis using multiple cut off to test whether the conclusion is robust. For example, how about raising FPKM cut off to 1, 10 and 100? that will reduce the number of detected transcribed gene and affect the % of transcribed genes in Fig 1c. How about the number of bias expressed genes in Ext Fig 1f, will that change? How about the expression variance? All those can be in a sup table. If the result stays the same when the cut off changes, then they can write something like "and it has been tested under different conditions".

L101. The author introduced a term here called "gene pair expression bias". The definition is actually buried in the method L878: Pair's expression level foldchange ≥ 2 , and occurred in $> 5\%$ accessions. Since they would used bias gene expression in many subsequent analysis, it is much better to put these to the main text and the reader understand what you have selected. Also, they need to tell us why they have chosen this. I understand the space is limited, but can they at least write in the method/supplementary and explain to readers why they choose $FC \geq 2$ and $> 5\%$? Whether the result is robust? Will changing the FPKM cut off and this FC cut off affect their result?

L104, Please remove this meaningless conclusion "These data provide genome-wide overview of dynamic regulation of homoeologous expression during fiber development."

L121. They used a new term "eQTL hotspot" without defining it.

L130, "25% of eGenes at each time point had more than 1 eQTL (Fig 1f)". This can have different interpretation. Please rephrase it to make it clear. Same for L127. Are they trying to say "a lot of genes have different eQTL in different developmental stage"?

L134, Please defined what is "stage-shared" and "stage-specific" before using these terms.

L136. Please explain or at least give a hypothesis for the observation that genes with stage-shared eQTL have more cis- than trans-eQTL. How are they distributed in the population, similar? What are these genes? Do they have specific functions, expression pattern etc? Does it mean that there are 2 groups of genes, whose expression change (diversity in ur population) are due to promoter evolution (cis-eQTL) or TF evolution (more trans-eQTL)?

L148. "sharing pattern of eQTLs was consistent with the correlation of gene expression (Fig. 1g)". Please explain. I can't tell what Fig 1g is from its legend. I guess it is a pair-wise comparison, and the correlation of gene expression is among accessions. If that is the case, the authors are basically saying if 2 developmental stage have similar gene expression pattern, the eQTL analysis will assign the similar eQTL to those genes. That is because we got the eQTL from gene expression, if the 2 stage have similar gene expression, they will surely have similar eQTL. If my understanding is correct, then

this part has little meaning.

L150 "different patterns between cis- and trans-eQTLs were observed in estimates of stage similarity (Ext Fig 2g)." Same as above.

L153: "Genes with shared cis-eQTL showed significantly higher expression correlation (Fig 1h)". Same as above.

L190: This is an interesting finding. But I guess most people can't understand what it mean. Adding a careful and detailed expiation of "favorable expression" and "favorable trait" might help. Also, why all these figures have no title? It kind of forced me to check the figure legend each time.

L212, "identified 406 eQTL hotspot". Please write about what is hot spot first. I am guessing that the authors merged the eQTL within a region to construct their network.

L222, the term "genetic module" is not accurate here. Just "module" is ok.

L229, The author found that the network modules are enriched of different features. But we can't say the module has "functionally differentiated". Because that implies the modules used to have the same feature and gained different ones during evolution/domestication/breeding. We only know that they are different. I suggest the authors to discuss why they are different. For example, if we consider some of those eQTL or eQTL hotspots are transcription factors genes, genes in the same module means genes regulated by similar TFs. So they have similar function. That has been shown in the maize 104 TF – target gene network.

L259- 262: This is an interesting finding (a lot of Dt subgenome trans-eQTL are regulating genes in the At subgenome), but what is the conclusion/hypothesis? The author said "suggesting genetic variation might lead to expression bias of homoeologous genes (Fig. 4b)". Of course genetic variation lead to expression bias. What else can? I think one could discuss why there are a lot of Dt subgenome eQTL that regulate At genes. Maybe it is the selection pressure and mutation of a couple TFs in Dt subgenome during domestication/breeding/selection? Are they the hotspot u find in Fig 3? I think this is an important finding that might help us to understand the domestication/breeding history of cotton.

L263-270. I don't understand why they emphasize this bias-eQTL, and why we need this analysis at all. They found the bias-eQTL is 90.5% overlap with eQTL. Which means I can just find the bias-expressed gene.

L278: "This suggests that the existence of genetic variants for homoeologous genes may lead to asymmetric regulation of expression." Doesn't make sense. For a pair of homologous gene, a variant (I guess they meant cis-eQTL) can only be assigned to one of them in say in At subgenome. So the other gene in Dt can't have the same variant.

L282, "suggesting the complexity of ... regulation" is a horrible conclusion. What is not complex?

L294. Same bad concluding sentence here.

L297. I think a lot needs to be explained here before going into "we have 3 groups (switched, time-dependent and dominant) of genes". What is the rationale of this analysis. How they get these 3

groups, by clustering? What is the biology meaning of these groups? Most of them are buried in the method and are hard to understand.

L317. Another meaningless concluding sentence. What do we actually learn from this bias expression analysis from L295 to L317? Just trans-eQTL enriched in the biased expressed genes? How does it help breeding or does it align with the theme of this paper? If not (at least I can't see), how about just use 1-2 sentence to tell the readers that there are expression bias, and put all these to the supplementary info and move on?

L328, "This analysis showed that these pseudo-regulatory sites are rarely mutated in the population (frequency < 0.01)" what is the control? Do you mean 1% SNP? That is very high for other species. And this is a horrible way to compare regulatory sites. Since the open chromatin is ~1kb, and most of the time, it is the TF binding sites ~10bp that cause the gene expression change. Comparing 2Mb up and down-stream has little biological/regulatory meaning. So the conclusion in L330 is at risk.

L354 "we counted ... candiate genes and their homoeologs in each accession". This doesn't seem to be what they actually did. I think they examined the homoe gene pairs' expression pattern in the population and linked them to the traits. They then counted the number of genes with a particular expression pattern in different groups with different traits.

Sup Fig 2a, please define the region promoter, how many kb upstream of the TSS?

Sup Fig2b, shouldn't one first compare the lead trans-variant vs non-lead trans-variants? then compare the lead trans-variant with the random variants?

Author Rebuttal, first revision:

Reviewer #1:

Remarks to the Author:

The revised manuscript has addressed all my questions. I have no further comments.

Response: Thank you very much for all previous suggestions that helped us craft a better manuscript.

Reviewer #2:

Remarks to the Author:

although the authors have attempted to describe their results in as clear way as possible for the general reader, the use of jargon such as "regulome" obscures whatever value the analyses may have.

The Discussion is also still problematic. It does not concisely describe the main findings except for (I think) that during breeding useful variation in homoeologs has not been exploited to its full extent. There are empty statements eg lines 454-457 that the increased complexity of homoeolog gene expression accounts for transgressive phenotypes. There are no insights there.

Response: Thank you very much for suggestions that help us improve the Results and Discussion.

In response, we changed “regulome” to “regulation” throughout the manuscript and reorganized some paragraphs in the Results to make them as clear as possible for the readers. We have also provided new thoughts in the Discussion and removed the description of “the increased complexity of homoeolog gene expression accounts for transgressive phenotypes”. We hope that the current Discussion is clearer.

Reviewer #3:

Remarks to the Author:

I agree with reviewer #2’s previous comments that the result was poorly written. Seems the revised version is no good either. It might not be possible for a non-expert to comprehend. The authors do not seem to understand how to present a story in a logical, concise and meaningful way. For each section, they seldom follow a regular flow such as: what is the goal, why we want to analyze this, why we use this method instead of others, what are the key findings, then come up with a hypothesis leading to the next result section. Currently, it is like a collection of all the results coming out from the analysis and all the figures. I suggest the authors to think about which part needs to be kept, which part can be removed without hurting your conclusion? currently, a lot of them do not seem to be related to the main story line, and they can go to supplementary. Do we need all these figures? It is not a show off of work load. Just pick the ones that can support the hypothesis. On the meantime, it can free up some space to write more about those findings that matter, which are currently under-explained making it very hard to read.

Response: Thank you very much for your comments and suggestions. We have tried our best to present the Results in a logical, concise and meaningful way. In the revised manuscript, we removed many results in the sections of “Genetic regulation of dynamic gene expression” and “Genetic effects on homoeologous expression partition”, and reorganized Figure 1 and Figure 4. We also added more descriptions related to some of the results showing why we performed that analysis, what we found and the implications or hypothesis. We hope the revised manuscript presents our findings clearly.

As for the discussion, it is like a recap of the result. But it is a common problem for this type of paper. Unlike the GWAS and eQTL studies in medical science, which can find casual SNP for disease diagnostic or treatment, plant population genetic projects like this often come up with lots

of data without immediate application. The only core value is the genomic data itself and the authors have to come up with a fancy story. Like this, they tried to claim that for allotetraploid crops, people can use homoeologous genes in the sub-genome for breeding. As I commented previously, other plants have whole-genome duplication and have ortholog gene pairs within the genome, and there is no need to prioritize the homoeologs. I agree that without content, writing a meaningful and relevant discussion is indeed challenging. Maybe the authors can trim down their discussion to a short paragraph and tell the reader what is their data, how it can be used for their research, and how to access them?

Response: Thank you for your comments and suggestions. Based on the suggestions from you and reviewer #2, we have rewritten the Discussion. Now, we mainly present and discuss three aspects: 1) The eQTL data in this study will promote the study of cis- and trans-regulation for fiber quality related genes, and future studies may explore the functional roles of cis-eQTLs in transcriptional factor binding sites, and also identify whether some key transcriptional factors are mutated in trans-eQTLs. 2) Transcriptional regulation is highly dynamic, and the eQTL analysis in different developmental stages will help understand the regulatory complexities of fiber development. 3) The TWAS genes and their homoeologs may be important candidates for uncovering the molecular mechanisms for fiber quality formation, and may promote genome-enabled breeding in polyploid crops. You will see that we have shown the value of the new data in this study, and make perspective on how to use these data in the future research.

Also, it would be helpful if the author highlights the changes in the text.

Response: We actually submitted the revised manuscript with the revision mode in Word. All the make-ups can be seen in the document.

Here is my suggestions for them to improve the result section:

L93. There is no need to draw a vague conclusion like “These data represent a rich resource for characterizing gene expression profiles in fiber development.”.

Response: Thank you for your suggestion. We have removed this sentence in the revised manuscript.

L95. “FPKM >0.1” is used to defined expressed genes. Why choose this? We should perform the analysis using multiple cut off to test whether the conclusion is robust. For example, how about raising FPKM cut off to 1, 10 and 100? that will reduce the number of detected transcribed gene and affect the % of transcribed genes in Fig 1c. How about the number of bias expressed genes in Ext Fig 1f, will that change? How about the expression variance? All those can be in a sup table.

If the result stays the same when the cut off changes, then they can write something like “and it has been tested under different conditions”.

Response: Thank you for your suggestion. We performed the analysis using multiple thresholds (range 0.1-20 FPKM) to test whether the conclusion is robust. Undoubtedly, the number of transcribed genes will decrease as the FPKM threshold increases. However, there is a consistent trend that more genes are transcribed in the Dt than in the At subgenome, and more homoeologous gene pairs show a steady expression bias towards the Dt than At (**Table R1**). As the number of expressed genes decreases, there is also a decrease in the number of genes showing expression variation (fold change ≥ 2) among accessions (**Table R2**). Because the conclusion remains robust under different conditions, we have chosen a threshold of FPKM > 0.1 based on our experiences in previous cotton RNA-Seq analysis. We have added a corresponding description in the Methods for the threshold of “FPKM > 0.1 ” (line 796-799) and added a supplementary table of statistics under different thresholds (**Supplementary Table 3**).

Table R1. The number of expressed genes and homoeologous genes with stable bias at various FPKM thresholds.

Threshold of FPKM	Expressed genes	Expressed At gene	Expressed Dt gene	Steady expression bias toward At	Steady expression bias toward Dt	Steady without expression bias
0.1	49,860	24,486	25,238	1,533	1,723	4,108
0.5	44,744	21,785	22,474	1,476	1,649	2,289
1	40,890	19,937	20,524	1,275	1,426	1,990
5	28,645	13,943	14,422	596	671	2,733
10	21,005	10,242	10,558	325	379	2,618
20	13,320	6,464	6,730	172	182	1,893

Table R2. Expression variation for the expressed genes.

Threshold of FPKM	Expressed genes	Expressed genes (6 stages)	Expressed genes (6 stages) under a range of fold change				
			1-2	2-4	4-8	8-16	≥ 16
0.1	49,860	258,443	99,226	111,712	26,961	11,307	9,237
0.5	44,744	223,827	64,610	111,712	26,961	11,307	9,237
1	40,890	196,461	64,297	94,010	21,635	8,643	7,876
5	28,645	114,480	55,618	45,086	7,368	3,032	3,376
10	21,005	76,273	41,573	27,031	4,033	1,636	2,000
20	13,320	45,082	25,621	15,281	2,198	835	1,147

*Note: Fold change between 5% and 95% quantile of expression levels across accessions was calculated and divided into five bins.

L101. The author introduced a term here called “gene pair expression bias”. The definition is actually buried in the method L878: Pair’s expression level foldchange ≥ 2 , and occurred in $> 5\%$ accessions. Since they would use biased gene expression in many subsequent analyses, it is much better to put these in the main text and the reader understand what you have selected. Also, they need to tell us why they have chosen this. I understand the space is limited, but can they at least write in the method/supplementary and explain to readers why they choose $FC \geq 2$ and $> 5\%$? Whether the result is robust? Will changing the FPKM cut off and this FC cut off affect their result?

Response: Thank you for your suggestion. We have added the definition of homoeologous expression bias to the main text (line 96-101) in the revised manuscript. In the method (line 796-799), we explain why we used the criteria of $FC \geq 2$ and percentage of accessions with biased expression $\geq 5\%$. Undoubtedly, the number of homoeologous genes with expression bias will decrease as the percentage of accessions increases. However, a 2-fold expression change threshold is commonly used to define expression change, and we used this criterion similar to Bonferroni correction to limit the number of accessions with the same direction of expression bias and differential expression to at least 5% of accessions, which ensures that the expression bias of gene pair is not by chance at each timepoint. We tested the robustness of results under different conditions. There are consistent trends that more homoeologous gene pairs show a steady expression bias towards the Dt than At, and eGenes were significantly enriched in gene pairs with biased expression (**Table R3**). We have added a supplementary table of statistics under different thresholds (**Supplementary Table 3**).

Table R3. The count of biased expression gene pairs at various thresholds.

Threshold of fold change	Threshold for percentage of accessions	Count of biased expression gene pairs	Steady expression bias toward At	Steady expression bias toward Dt	Percentage of eGene for biased expression gene pairs	Percentage of eGene for equal expression gene pairs
2	5%	16,081	1,533	1,723	48.1%	29.3%
2	10%	15,181	1,299	1,491	43.4%	27.7%
2	15%	14,449	1,150	1,312	44.0%	28.5%
3	5%	13,648	934	1,025	42.9%	30.1%
3	10%	12,710	796	890	43.9%	30.7%
3	15%	11,948	696	781	44.6%	31.2%
4	5%	12,602	746	830	42.4%	31.5%
4	10%	11,698	631	708	43.5%	31.9%
4	15%	10,946	552	621	44.2%	32.2%

L104, Please remove this meaningless conclusion “These data provide genome-wide overview of dynamic regulation of homoeologous expression during fiber development.”

Response: Thank you for your suggestion. We have removed this sentence in the revised manuscript.

L121. They used a new term “eQTL hotspot” without defining it.

Response: Thank you for your suggestion. We have changed the sentence to “eQTL hotspots (i.e. local chromosomal regions that were associated with transcriptional regulation of more than 3 genes) at this timepoint” (line 115-121). The method of eQTL hotspot identification was described in line 1005-1007.

L130, “25% of eGenes at each time point had more than 1 eQTL (Fig 1f)”. This can have different interpretation. Please rephrase it to make it clear. Same for L127. Are they trying to say “a lot of genes have different eQTL in different developmental stage”?

Response: Thank you for your suggestion. In this paragraph, we explored the eQTL profile of eGenes at each timepoint and did not compare the difference across stages. It emphasizes that a lot of genes were regulated by multiple eQTLs simultaneously, and an average of 25% eGenes had more than 1 eQTL. We have moved this part of the results to the Supplementary Information file (**Supplementary Results; Supplementary Figs. 3a-c**).

L134, Please defined what is “stage-shared” and “stage-specific” before using these terms.

Response: Thank you for your suggestion. We have added a corresponding description in the manuscript (line 122-125).

L136. Please explain or at least give a hypothesis for the observation that genes with stage-shared eQTL have more cis- than trans-eQTL. How are they distributed in the population, similar? What are these genes? Do they have specific functions, expression pattern etc? Does it mean that there are 2 groups of genes, whose expression change (diversity in ur population) are due to promoter evolution (cis-eQTL) or TF evolution (more trans-eQTL)?

Response: Thank you for your suggestion. According to whether a gene has been identified as an eGene in multiple stages, all eGenes are divided into two groups. Genes in group 1 had eQTLs in at least 2 stages, while genes in group 2 had only detected eQTL at a specific stage.

There may be two hypotheses here. Hypothesis 1: Regulators (such TFs) that regulate gene expression have stage-specific expression, leading to stage-specific expression of genes under their regulation. If a lot of eGenes in group 2 show a stage-specific expression pattern, then these genes will have eQTLs at specific stages. Hypothesis 2: Genes with constitutive expression across multiple stages, but the regulator (such TF) that regulate gene expression changes in different stages. Therefore, genes in group 2 can only have trans-eQTLs at specific stage, while the widespread variation in promoter regions of group1 genes leads to the identification of the same or different cis-eQTL at multiple stages.

To verify the above hypotheses, we calculated the proportion of genes that were expressed (at least of 95% accessions with FPKM > 0.1) in multiple stages and counted the number of SNPs in the promoter regions in both groups. A total of 6,793 (75.9%) and 12,231 (86.9%) genes were expressed in all 6 stages (**Fig. R1a**), indicating a lack of stage-specific expression. We found that the promoter region of genes in group 1 had more variations (**Fig. R1b**), suggesting that variations in the cis-regulatory regions of genes can affect the regulation of their expression across multiple stages. We provided a hypothesis for the observation to make it easier to understand (line 125-130).

Figure R1. The comparison of gene expression pattern and the number of variants in the promoter regions.

a: Comparison of the stage count distribution for eGene without expression.

b: Distribution of the count of variants in the promoter regions.

L148. “sharing pattern of eQTLs was consistent with the correlation of gene expression (Fig. 1g)”. Please explain. I can’t tell what Fig 1g is from its legend. I guess it is a pair-wise comparison, and the correlation of gene expression is among accessions. If that is the case, the authors are basically saying if 2 developmental stage have similar gene expression pattern, the eQTL analysis will assign the similar eQTL to those genes. That is because we got the eQTL from gene expression, if the 2 stage have similar gene expression, they will surely have similar eQTL. If my understanding is correct, then this part has little meaning.

Response: Thank you for your comments. In the Fig 1g, the degree of eQTL sharing between timepoints is quantified as the proportion of cis-eQTLs that are shared between any two given timepoints. For the comparison of FPKM between timepoints, we calculated the median expression level for each gene across different accessions, and analyzed the correlation of median expression levels among genes. We agree with you that similar expression across stages will result in similar eQTL regulation. We have removed this sentence in the revised manuscript.

L150 “different patterns between cis- and trans-eQTLs were observed in estimates of stage similarity (Ext Fig 2g).” Same as above.

Response: Thank you for your comments. We have removed this sentence in the revised manuscript.

L153: “Genes with shared cis-eQTL showed significantly higher expression correlation (Fig 1h)”. Same as above.

Response: Thank you for your comments. We have removed this sentence in the revised manuscript.

L190: This is an interesting finding. But I guess most people can’t understand what it mean. Adding a careful and detailed expiation of “favorable expression” and “favorable trait” might help. Also, why all these figures have no title? It kind of forced me to check the figure legend each time.

Response: Thank you for your suggestion. To make it clear to readers, we used BB2 as an example to explain the definition of “favorable expression pattern” in the revised manuscript (line 172-175) and the detailed method could be found in Supplementary Information file.

The revised sentence is shown below: “Similar to the analysis for *BB2*, we investigated the expression patterns for all 1,258 genes from TWAS and colocalization analysis in all accessions, and defined “favorable expression” patterns as those in accessions with favorable fiber quality

traits, such as longer or stronger fiber, which represent the major breeding goal of high quality cotton”.

L212, “identified 406 eQTL hotspot”. Please write about what is hot spot first. I am guessing that the authors merged the eQTL within a region to construct their network.

Response: Thank you for your suggestion. We have defined eQTL hotspots as “local chromosomal regions that were associated with transcriptional regulation of more than 3 genes” in the revised manuscript (line 115-121). The detailed method of eQTL hotspot identification is now described in line 1005-1007.

L222, the term “genetic module” is not accurate here. Just “module” is ok.

Response: Thank you for your suggestion. We changed “genetic module” to “module” here.

L229, The author found that the network modules are enriched of different features. But we can't say the module has “functionally differentiated”. Because that implies the modules used to have the same feature and gained different ones during evolution/domestication/breeding. We only know that they are different. I suggest the authors to discuss why they are different. For example, if we consider some of those eQTL or eQTL hotspots are transcription factors genes, genes in the same module means genes regulated by similar TFs. So they have similar function. That has been shown in the maize 104 TF – target gene network.

Response: Thank you for your suggestion. We have rewritten this sentence as “This indicates that modules may differ with respect to phenotypic effects, possibly because different modules are controlled by different regulatory factors such as transcriptional factors”. We also cited the maize 104 TF paper to support the speculation here.

L259- 262: This is an interesting finding (a lot of Dt subgenome trans-eQTL are regulating genes in the At subgenome), but what is the conclusion/hypothesis? The author said “suggesting genetic variation might lead to expression bias of homoeologous genes (Fig. 4b).” Of course genetic variation lead to expression bias. What else can? I think one could discuss why there are a lot of Dt subgenome eQTL that regulate At genes. Maybe it is the selection pressure and mutation of a couple TFs in Dt subgenome during domestication/breeding/selection? Are they the hotspot u find in Fig 3? I think this is an important finding that might help us to understand the domestication/breeding history of cotton.

Response: Thank you for your suggestion. In the eQTL hotspots that we have identified, there are indeed several hotspots located on the Dt subgenome that regulate gene expression on the At subgenome (**Table R3**), which may explain why many genes on the At are regulated by eQTLs from the Dt. Since this part of the results is a bit distant from the theme of the homoeologous expression bias, we moved these results to the supplementary files. We have reorganized the results of the “biased gene pairs enriched with eQTLs” (**Fig. 4b**) and present them together with the results of bias-eQTL to highlight eQTLs as the driving factor for biased expression of homoeologous genes (line 248-265).

Table R3. The count of target eGenes for hotspots in the Dt subgenome.

Hotspot id	Target eGenes	At eGene	Stages
Hot369	1,546	760	12 DPA
Hot355	999	471	12 DPA
Hot281	156	78	4 DPA
Hot206	573	296	16 DPA
Hot369	125	61	16 DPA

L263-270. I don't understand why they emphasize this bias-eQTL, and why we need this analysis at all. They found the bias-eQTL is 90.5% overlap with eQTL. Which means I can just find the bias-expressed gene.

Response: Thank you for your comments. We want to know which loci in the genome regulate the biased expression of homoeologous gene pairs. We could not determine whether a locus significantly regulates the expression bias of homoeologous gene pairs from eQTL analysis alone. Therefore, we conducted GWAS analysis (bias-GWAS) for each gene pair with expression bias. A total of 14,133 (bias-QTL) were detected for 4,026 homoeologous gene pairs. Since we found significant enrichment of eQTL for gene pairs with expression bias (**Fig 4b**), we evaluated the regulatory effects of 14,133 bias-QTLs on expression of individual genes. The results demonstrated a high correlation between eQTL and bias-QTL (coefficient: 0.96) indicating that eQTL are drivers of biased expression of homoeologous gene pairs. We further analyzed 5,350 bias-eQTLs that were identified as significant QTL signals in both bias-GWAS and eQTL analysis. We found that 90.5% of bias-eQTLs regulated expression bias of homoeologous gene pairs by changing expression of a gene in the homoeologous gene pair. However, 8,841 QTL loci identified in bias-GWAS or eQTL analysis were not further analyzed due to their lower regulatory effects and non-repeatability between the two methods. In summary, bias-GWAS can help us identify which QTL loci are associated with biased expression of homoeologous gene pairs. By intersecting these loci with eQTLs, we can identify regulatory sites (bias-eQTLs) that are significant in both approaches.

L278: “This suggests that the existence of genetic variants for homoeologous genes may lead to asymmetric regulation of expression.” Doesn’t make sense. For a pair of homologous gene, a variant (I guess they meant cis-eQTL) can only be assigned to one of them in say in At subgenome. So the other gene in Dt can’t have the same variant.

Response: Thank you for your comments. In this part, we found that 4,846 (90.5%) of the bias-eQTLs regulated expression bias of homoeologous gene pairs by changing the expression of a gene in the homoeologous gene pair. Furthermore, the count of SNPs in the promoter regions of homoeologous gene pairs showed significant differences (**Extended Data Fig. 5c**). This suggests that there are different frequencies of genetic variation in the regulatory elements that control gene expression in the two subgenomes, which in turn lead to the occurrence of expression bias. For example, we identified a bias-eQTL in the At subgenome but not in the Dt subgenome. This suggests that the regulatory elements (such as TF binding sites) in the At subgenome have variations among the accessions, whereas the corresponding regulatory element in the Dt subgenome appears to be present but has no variation among accessions. We have rewritten the section to make it clear (line 256-261).

L282, “suggesting the complexity of ... regulation” is a horrible conclusion. What is not complex?

Response: Thank you for your suggestion. In **Fig. 4b**, we illustrated the relationship between homoeologous gene expression and phenotype through an example. Specifically, accessions with lower expression bias between homoeologous genes show longer fiber length. This suggests that further utilization of homoeologous genes may be possible to improve fiber phenotype. We have rewritten the result of this section and removed the conclusion (line 248-265).

L294. Same bad concluding sentence here.

Response: Thank you for your suggestion. As we mentioned before (line 256-261), we have observed that the majority of expression bias in homoeologs may result from variations in the regulatory regions of one subgenome while the other subgenome’s regulatory region has no variation across accessions. To test this hypothesis, we assessed the relationship between the difference of regulatory sequence in the two subgenomes and the expression bias in homoeologous genes. Theoretically, if regulatory elements (TFBSs) within a single accession vary across subgenomes, this could lead to differential expression of homoeologous genes in that accession. Indeed, differences in SNPs among subgenomes were accompanied by the existence of expression bias and positively correlated with the regulatory effect of bias-eQTL. This suggested that

differences in the expression regulatory region of the homoeologs are one of the factors contributing to expression bias. We have removed this section in the revised manuscript.

L297. I think a lot needs to be explained here before going into “we have 3 groups (switched, time-dependent and dominant) of genes”. What is the rationale of this analysis. How they get these 3 groups, by clustering? What is the biology meaning of these groups? Most of them are buried in the method and are hard to understand.

Response: Thank you for your suggestion. Please see below for our explanations.

Question: What is the rationale of this analysis?

Homoeologous genes in the two subgenomes exhibit high sequence similarity and can have redundancy or subfunctionalization in their functions. At a given timepoint, we can analyze the presence of expression bias and determine the direction of bias. By comparing changes in expression bias for the same gene pair at different timepoints, we can identify potential novel functional or subfunctionalized gene pairs.

Question: How to get these 3 groups, by clustering?

For each pair of homoeologous genes, we clustered their dynamics of expression bias in each accession into different classes. Among the 340 accessions with transcriptomes recorded across 6 timepoints, we counted the number of accessions in each class for each gene pair. For instance, let's consider gene pair X, which has consistent expression bias toward the At subgenome in 6 timepoints across 10 accessions. However, the direction of expression bias changes in at least 2 timepoints across 330 accessions. Then, we performed hierarchical clustering on the homoeologous gene pairs based on the count of accessions across different classes. For gene pairs that fall into the switched groups, they exhibit a change in the direction of expression in at least 2 timepoints across the majority of accessions.

Question: What is the biology meaning of these groups?

For gene pairs in the switched group, they exhibit expression bias towards the At in a specific timepoint while expression bias towards the Dt in other timepoints. This may reflect potential functional divergence among homoeologous genes and their collaborative regulation across fiber development. In the time-dependent group, gene pairs show biased expression towards the same subgenome in 1-5 time points. In the dominant group, gene pairs consistently exhibit biased expression towards the same subgenome in all six time points. These two groups of homoeologous gene pairs suggest that homoeologous genes may primarily rely on the regulation from each subgenome across fiber development.

L317. Another meaningless concluding sentence. What do we actually learn from this bias expression analysis from L295 to L317? Just trans-eQTL enriched in the biased expressed genes? How does it help breeding or does it align with the theme of this paper? If not (at least I can't see), how about just use 1-2 sentence to tell the readers that there are expression bias, and put all these to the supplementary info and move on?

Response: Thank you very much for your suggestion. In this study, we sought to identify the genetic basis for homoeologous expression bias using bias-QTL analysis, and also explored the situation in which bias-eQTLs contribute to homoeologous expression bias. Moreover, we interrogated the impact of genetic variants on the dynamics of expression bias during fiber development. We hold the view that previous studies in polyploid plants failed to perform the genetic analysis of homoeologous expression bias, so we think this section will provide new insights into the genetic regulation of homoeologous genes. During the revision, we reorganized this section "Genetic effects on homoeologous expression bias". You will see that we have shortened this section by removing some results and moving some to the supplementary files. We hope the revised manuscript will make our manuscript clearer and more readily understood with respect to significance of the findings.

L328, "This analysis showed that these pseudo-regulatory sites are rarely mutated in the population (frequency < 0.01)" what is the control? Do you mean 1% SNP? That is very high for other species. And this is a horrible way to compare regulatory sites. Since the open chromatin is ~1kb, and most of the time, it is the TF binding sites ~10bp that cause the gene expression change. Comparing 2Mb up and down-stream has little biological/regulatory meaning. So the conclusion in L330 is at risk.

Response: Thank you for your suggestion. In this study, we assumed that the regulatory sites of the homoeologous genes of TWAS genes would be located near the homoeologous genes, and the regions in cis-eQTL analysis are generally "1Mb up and down-stream". Since there is a difference in the size of the two subgenomes (At, 1.4 Gb; Dt, 0.8 Gb), we identified pseudo-regulatory sites in the region of "2Mb up and down-stream" of the homoeologous genes. We extracted the "200bp up and down-stream" of significant SNPs as query sequences and the "2Mb up and down-stream" of the homoeologous genes as subject sequences. Alignments were generated using LASTZ (v1.04.03). Using this approach, we identified pseudo-regulatory sites of homoeologous genes. We found that 0 of 323 pseudo sites for FL homoeologous genes were mutated in the population, 1 of 732 pseudo sites for FS homoeologous genes were mutated in the population, 0 of 316 pseudo sites for FU homoeologous genes were mutated in the population, and 1 of 1071 pseudo sites for FL homoeologous genes were mutated in the population. The total number of pseudo sites for all four traits is 2442. To make it clear to readers, we have rewritten this sentence as "This analysis showed that few pseudo-regulatory sites (2 of 2442) for the four fiber quality traits are mutated in the population" (line 295-297).

L354 “we counted ... candidate genes and their homoeologs in each accession”. This doesn’t seem to be what they actually did. I think they examined the homoe gene pairs’ expression pattern in the population and linked them to the traits. They then counted the number of genes with a particular expression pattern in different groups with different traits.

Response: Thank you very much for your comments. We are sorry for the ambiguity caused by the descriptions in L354. We have rewritten these sentences as “To assess the impact of aggregating favorable homoeologous pairs with favorable genotypes or expression on fiber quality traits, we counted the number of gene pairs represented by each of the four models above in each accession. For FL, we observed a larger number of gene pairs categorized as favorable homoeologous pairs in long fiber accessions than in short fiber accessions” (line 324-326).

Sup Fig 2a, please define the region promoter, how many kb upstream of the TSS?

Response: Thank you for your suggestion. We have added the definition of the promoter region (2 kb upstream of TSS) in the legend.

Sup Fig2b, shouldn’t one first compare the lead trans-variant vs non-lead trans-variants? then compare the lead trans-variant with the random variants?

Response: Thank you very much for your comments. In this analysis, we investigated whether a region associated with a trans-eQTL also acted as a cis-eQTL for another gene. We use lead trans-variant to represent the entire trans-eQTL and assessed its potential cis-regulatory effects on other genes. It is worth noting that, in the absence of fine-mapping evidence, the lead variant is often considered as a potential causal variation in QTL studies. Therefore, we compared the lead trans-variants with random variants.

Decision Letter, second revision:

30th Jun 2023

Dear Dr. Wang,

Thank you for submitting your revised manuscript "Regulatory controls of duplicated gene expression during fiber development in allotetraploid cotton" (NG-A61414R1). It has now been seen by the

original referees and their comments are below. The reviewers find that the paper has improved in revision, and therefore we'll be happy in principle to publish it in Nature Genetics, pending minor revisions to satisfy the referees' final requests and to comply with our editorial and formatting guidelines.

Sincerely,
Wei

Wei Li, PhD
Senior Editor
Nature Genetics
New York, NY 10004, USA
www.nature.com/ng

Reviewer #3 (Remarks to the Author):

The authors have responded to my questions, and the writing has ben improved a bit despite it is still quite hard to read. I have no further comments.

Final Decision Letter:

14th Sep 2023

Dear Dr. Wang,

I am delighted to say that your manuscript "Regulatory controls of duplicated gene expression during fiber development in allotetraploid cotton" has been accepted for publication in an upcoming issue of Nature Genetics.

Your paper will be published online after we receive your corrections and will appear in print in the next available issue. You can find out your date of online publication by contacting the Nature Press Office (press@nature.com) after sending your e-proof corrections. Now is the time to inform your Public Relations or Press Office about your paper, as they might be interested in promoting its publication. This will allow them time to prepare an accurate and satisfactory press release. Include your manuscript tracking number (NG-A61414R2) and the name of the journal, which they will need when they contact our Press Office.

Please note that *Nature Genetics* is a Transformative Journal (TJ). Authors may publish their research with us through the traditional subscription access route or make their paper immediately open access through payment of an article-processing charge (APC). Authors will not be required to make a final decision about access to their article until it has been accepted. [Find out more about Transformative Journals](https://www.springernature.com/gp/open-research/transformative-journals)

Authors may need to take specific actions to achieve [compliance with funder and institutional open access mandates](https://www.springernature.com/gp/open-research/funding/policy-compliance-faqs). If your research is supported by a funder that requires immediate open access (e.g. according to [Plan S principles](https://www.springernature.com/gp/open-research/plan-s-compliance)) then you should select the gold OA route, and we will direct you to the compliant route where possible. For authors selecting the subscription publication route, the journal's standard licensing terms will need to be accepted, including [self-archiving-and-license-to-publish](https://www.nature.com/nature-portfolio/editorial-policies/self-archiving-and-license-to-publish). Those licensing terms will supersede any other terms that the author or any third party may assert apply to any version of the manuscript.

If you have not already done so, we invite you to upload the step-by-step protocols used in this manuscript to the Protocols Exchange, part of our on-line web resource, natureprotocols.com. If you complete the upload by the time you receive your manuscript proofs, we can insert links in your article that lead directly to the protocol details. Your protocol will be made freely available upon publication of your paper. By participating in natureprotocols.com, you are enabling researchers to more readily reproduce or adapt the methodology you use. [Natureprotocols.com](http://natureprotocols.com) is fully searchable, providing your protocols and paper with increased utility and visibility. Please submit your protocol to <https://protocolexchange.researchsquare.com/>. After entering your nature.com username and password you will need to enter your manuscript number (NG-A61414R2). Further information can be found at <https://www.nature.com/nature-portfolio/editorial-policies/reporting-standards#protocols>

Sincerely,
Wei

Wei Li, PhD
Senior Editor
Nature Genetics
New York, NY 10004, USA
www.nature.com/ng